# PepTri: Tri-Guided All-Atom Diffusion for Peptide Design via Physics, Evolution, and Mutual Information

**Ngoc-Quang Nguyen**[1]   **Jaeyoon Jung**[1]   **Seijung Kim**[1]   **Sunkyu Kim**[1] *   **Jaewoo Kang**[1,2] *

[1] AIGEN Sciences   [2] Korea University

{quang.nguyen, jaeyoon.jung, seijung.kim, sunkyu.kim, jaewoo.kang}@aigensciences.com

## Abstract

Peptides, short chains of amino acids capable of high-specificity protein binding, represent a powerful class of therapeutics. While deep generative models have shown promise for peptide design, existing approaches are often structure-centric and therefore generate sequences and structures in a decoupled manner, failing to ensure that designs are simultaneously physically stable, evolutionarily plausible, and internally coherent. To overcome this limitation, we introduce **PepTri**, a novel diffusion framework that addresses this by jointly generating peptide sequences and 3D structures within a unified, SE(3)-equivariant latent space. Our proposed model integrates three complementary guidance signals during the generative process: (i) physics-informed guidance via differentiable molecular mechanics to ensure structural stability and realism; (ii) evolutionary guidance to bias sequences toward conserved, functional motifs; and (iii) mutual information guidance to explicitly maximize sequence-structure coherence. This tri-guided approach ensures the generative process is steered by biophysical laws, biological priors, and information-theoretic alignment in tandem. Extensive evaluations on challenging peptide-protein design benchmarks, cross-domain (PepBench, LNR) and in-domain (PepBDB), demonstrate that PepTri substantially outperforms strong baselines, achieving state-of-the-art results in binding affinity, structural accuracy, and design diversity. Our results establish that integrating these complementary signals directly into the denoising process is crucial for generating viable, high-quality peptide medicines. PepTri is available at: https://github.com/aigensciences/PepTri

## 1 Introduction

The therapeutic potential of peptides—short chains of amino acids—is rapidly being realized, evidenced by over 100 approved drugs and a robust pipeline of hundreds more in development. (Kaygisiz et al., 2025; Zhai et al., 2025). Their advantages over small molecules and biologics include high specificity, low toxicity, and the ability to target "undruggable" proteins (Craik et al., 2013; Lai et al., 2025). Yet rational design remains challenging: the sequence space is astronomical ($20^L$ possibilities), peptides are highly flexible and often lack stable tertiary structure, and candidate sequences must satisfy interdependent geometric, evolutionary, and physicochemical constraints (Muttenthaler et al., 2021).

Deep generative models have advanced peptide and protein design but remain incomplete. *Structure-aware* generators produce plausible geometries yet neglect evolutionary constraints; *evolutionary* sequence models capture conservation but ignore 3D stability; and *physics*-based checks are usually applied post hoc rather than during generation (Ho et al., 2020; Kong et al., 2024). No existing method ensures designs that are simultaneously physically stable, evolutionarily plausible, and sequence–structure coherent.

---

*Corresponding authors.

We introduce **PepTri**, a tri-guided diffusion framework integrating complementary signals during training and sampling: (i) **physics guidance** with SE(3) awareness to ensure molecular stability; (ii) **evolutionary guidance** via BLOSUM-derived embeddings and co-variation; and (iii) **mutual-information maximization** aligning sequence and structure representations. Our contributions include a parameter-efficient architecture with compact latents, a dynamic guidance schedule balancing stability and diversity, a unified diffusion objective combining physics, evolutionary, and information-theoretic terms, and a robust training pipeline with mixed-precision and EMA stabilization. Together, these enable PepTri to generate diverse peptides with physically plausible, energetically favorable structures.

## 2 Related Work

**Physics- and empirical design.** Traditional pipelines—mutagenesis, phage display, and Rosetta-based modeling—have succeeded in narrow settings but face limited sampling and costly exploration of vast sequence spaces (Smith & Petrenko, 1997; Leaver-Fay et al., 2011; Kuhlman & Bradley, 2019).

**Evolutionary sequence models.** Potts/MSA-based models bias sequences toward biological plausibility (Marks et al., 2011) but depend on homologs and do not enforce geometric or energetic realism during generation.

**Structure-aware generative modeling.** Diffusion and flow-based models learn backbone or scaffold distributions and generate diverse structures (Trippe et al., 2022; Watson et al., 2023; Abdin & Kim, 2024). While accurate predictors (e.g., AlphaFold) aid evaluation (Abramson et al., 2024), they are not generative. For peptides, coupling sequence, structure, and domain constraints inside the generative loop remains unresolved.

**Peptide-focused baselines.** **PepGLAD** uses latent diffusion with auxiliary geometry losses but leaves energetics post hoc (Kong et al., 2024). **PepFlow** factorizes modalities via flow matching but checks stability only after generation (Li et al., 2024). **UniMoMo** unifies binders and pockets but relies on heuristics (e.g., distance thresholds) that weaken fine-scale couplings (Kong et al., 2025). Across these, physics and evolutionary priors remain auxiliary rather than shaping denoising dynamics.

**Our position.** Most existing models prioritize generating plausible 3D backbones but treat peptide sequences as secondary, often decoupled from structure or checked only post hoc. This imbalance leads to geometries that appear stable but in fact correspond to unrealistic or biologically implausible sequences. **PepTri** addresses this by unifying sequence and structure in an SE(3)-equivariant latent space and applying tri-guidance—physics, evolution, and information-theoretic alignment—*during* training and denoising, as illustrated in Figure 1. By directly injecting physical guidance into generation and explicitly aligning sequence–structure latents, PepTri yields peptides that are not only geometrically sound but also evolutionarily plausible and sequence–structure coherent.

## 3 Methodology

We adopt a two-stage framework: first, a VAE that compresses sequence–structure inputs into a latent space while preserving full SE(3) geometry; second, a latent diffusion model augmented with tri-guidance (physics, evolution, and mutual information) to generate biologically plausible peptides.

### 3.1 VAE with SE(3)-Equivariant Encoding

The PepTri encoder employs SE(3)-equivariant graph neural networks, which enforce rotational and translational symmetries while encoding both local residue-level interactions and global structural dependencies. This design enables accurate modeling of protein conformations within a symmetry-aware latent space, thereby facilitating downstream generative diffusion.

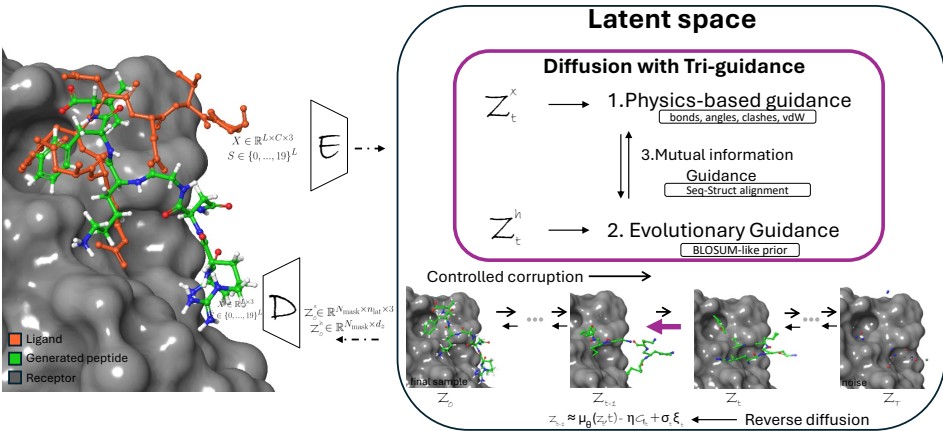

Figure 1: **PepTri architecture.** An SE(3)-equivariant encoder $E$ maps sequence–structure inputs $(S, X)$ to latents $(z_x, z_h)$; a decoder $D$ reconstructs $(\hat{S}, \hat{X})$. In latent space, sampling runs as *guided reverse diffusion* from $z_T$ (noise) to $z_0$ (sample), while training uses *controlled corruption* in the forward direction. At each step, a tri-guidance term $G_t$ steers denoising: (1) *physics-based* guidance acts on $z_x$ (bonds/angles/clashes/van der Waals), (2) *evolutionary* guidance biases $z_h$ (BLOSUM-like prior), and (3) *mutual-information* guidance aligns $z_x$ and $z_h$. The update follows $z_{t-1} \approx \mu_\theta(\mathbf{z}_t, t) - \eta_t G_t + \sigma_t \xi$. Gray surface: receptor; green: generated peptide; orange: ligand (receptor shown for context; guidance is intra-peptide).

**Graph construction.** From sequence $S \in \{0, \dots, 19\}^L$, coordinates $X \in \mathbb{R}^{L \times C \times 3}$, and mask $M$, we form a residue graph with (i) sequential edges $(i, i+1)$ and (ii) spatial edges within radius $r_{\text{cut}}$. Node features come from $S$, $X$; edge features are strictly SE(3)-invariant. The encoder outputs disentangled latents:

$$z_h \in \mathbb{R}^{L \times d_h} \quad \text{(sequence)}, \qquad z_x \in \mathbb{R}^{L \times n_{\text{lat}} \times 3} \quad \text{(structure)}.$$

$n_{\text{lat}} \in \mathbb{N}$ is the number of structural latent anchors (channels) per residue; each anchor carries 3D coordinates.

**SE(3)-equivariant message passing.** We developed an enhanced version of adaptive multi-channel EGNN (Kong et al., 2023). Invariant edge features include pairwise distances $d_{ij}$ and averaged triplet angles $\psi_{ij}$. Updates are:

$$\mathbf{x}_i' = \mathbf{x}_i + \sum_{j \in \mathcal{N}(i)} \phi(d_{ij}, \psi_{ij}, h_i, h_j)(\mathbf{x}_i - \mathbf{x}_j),$$

$$h_i' = \psi_h\left(h_i, \sum_{j \in \mathcal{N}(i)} \psi_m(h_i, h_j, d_{ij}, \psi_{ij})\right).$$

All coefficients are invariant, and updates use only relative vectors, guaranteeing SE(3)-equivariance.

**Training objective.** The VAE is trained to reconstruct both sequence and structure while enforcing geometric invariance:

$$\mathcal{L}_{\text{VAE}} = \text{CE}(S, \widehat{S}) + \|X - \widehat{X}\|_2^2 + \beta \, \mathcal{L}_{\text{KL}} + \lambda_{\text{geom}} \|D(\widehat{X}) - D(X)\|_{\text{Fr}}^2, \tag{1}$$

where $D(X)$ is the pairwise distance matrix. This enforces SE(3)-invariant structural consistency.

*Thus, all inputs, features, and updates respect equivariance, and the encoder–decoder is strictly SE(3)-equivariant. We use a VAE framework instead of a direct diffusion model to improve training stability and allow for a more compact, structured latent space.*

## 3.2 Tri-guidance diffusion

As shown in Figure 1, we augment latent diffusion with three complementary signals: (1) physics-informed structural guidance, (2) evolutionary sequence guidance, and (3) sequence–structure coherence via mutual information. These guide denoising to respect physical laws, evolutionary constraints, and sequence–structure alignment.

### 3.2.1 Physics-informed structural guidance

When relying only on data, we found that generated coordinates often exhibit broken bond lengths, unrealistic bond angles, or steric clashes that violate physical plausibility. To address this, we augment diffusion with *physics-informed regularization*. Our goal is to guide the model toward 3D structures that are both consistent with the data and stable under physical principles.

**Inputs and scope.** We represent peptides in an all-atom format, $X \in \mathbb{R}^{L \times C \times 3}$ with $C$ channels (backbone + sidechain), together with a binary design mask $M \in \{0, 1\}^L$ indicating which residues to optimize. For numerical stability, physics-guidance terms are computed on the $C_\alpha$ trace, while the full atom tensor is propagated and decoded.

**Physics parameterization.** We use a composite energy over the designed region:

$$E_{\text{phys}}(\widehat{X}, S; M) = \sum_j w_j \, E_j(\widehat{X}, S; M),\tag{2}$$

with active terms: bond-length, bond-angle, van der Waals, electrostatics, clash prevention, secondary-structure proxy, and diffusion smoothness. By averaging across all valid masked $C_\alpha$ interactions (excluding self-pairs and, for non-bonded terms, immediate neighbors), we obtain a well-scaled and numerically stable energy landscape.

**Gradient computation.** Inspired by (Guo et al., 2024), we treat $E_{\text{phys}}$ as an additional training-time regularizer. At each training step, we:

1. Obtain predicted coordinates $\widehat{X}$ for the designed region ($C_\alpha$ trace) from the encoder/decoder stack.
2. Evaluate $E_{\text{phys}}(\widehat{X}, S; M)$.
3. Backpropagate to obtain a masked gradient $\nabla_{\widehat{X}} E_{\text{phys}}$, which is propagated through the encoder and diffusion network to update the model parameters.

This regularizes the denoiser toward physically plausible structures on designed residues. Because the energy depends only on internal coordinates (distances, angles, and masked pairwise interactions) rather than absolute positions, the resulting gradients naturally preserve SE(3)-invariance.

**OpenMM coupling.** We additionally add a differentiable force field term:

$$\mathcal{L}_{\text{OpenMM}} = E_{\text{OpenMM}}(\widehat{X}, \, S; \, M),\tag{3}$$

computed via OpenMM. In our implementation, PepTri is coupled to the **Amber14** all-atom force field *(amber14-all.xml)* (Case et al., 2023; Eastman et al., 2023) for the last time step to evaluate $E_{\text{OpenMM}}$ and its forces on $C_\alpha$ atoms in the designed region, contributing an additional physics-based training loss that encourages realistic bond geometry, steric stability, and electrostatics.

In addition, we provide an *optional* test-time energy-guided sampler that performs energy-guided denoising,

$$\varepsilon_t \leftarrow \varepsilon_t - \gamma \, \nabla_{x_t} E_{\text{OpenMM}}(x_t, \, S)\big|_M,\tag{4}$$

where $\varepsilon_t$ is the predicted noise at step $t$ and the gradient is restricted to designed residues. This gently nudges the diffusion trajectory toward lower-energy conformations and acts as a physics-aware correction during generation.

**Physics loss.**

$$\mathcal{L}_{\text{phys}} = \lambda_{\text{phys}} E_{\text{phys}}(\widehat{X}, S; M) + \lambda_{\text{OpenMM}} \mathcal{L}_{\text{OpenMM}}. \tag{5}$$

*This module therefore enforces bond geometry, non-bonded interactions, and steric constraints while remaining SE(3)-consistent, ensuring generated structures remain physically plausible.*

Further details are provided in Appendix P.4

### 3.2.2 EVOLUTIONARY SEQUENCE GUIDANCE

Nature has already performed an enormous combinatorial search over protein space, leaving behind patterns of conservation and co-evolution that encode which residues "work". We inject this evolutionary signal so generated peptides remain biologically plausible.

**Inputs and scope.** We begin with residue-level embeddings $H_0 \in \mathbb{R}^{L \times d}$ from the VAE encoder and a mask $M$. Evolutionary guidance contributes training losses on these clean embeddings; its effect at sampling is implicit via the learned denoiser.

**BLOSUM-like embeddings.** We learn BLOSUM-inspired matrix $B \in \mathbb{R}^{20 \times 20}$ residue features

$$\tilde{H} = H_0 + \omega \, \phi\big(Y \, B \, F_1 + b_1\big) F_2 + b_2, \quad \phi = \text{ReLU}, \ \omega \in \mathbb{R}_{>0} \text{ learnable.} \tag{6}$$

$Y$ represents the one-hot encoded amino acid type, $Y \in \mathbb{R}^{L \times 20}$ and $F$ is a projection.

**Residue dependency attention via multi-head self-attention.** PepTri captures inter-positional dependencies with residual multi-head self-attention. Let $h$ denote the number of heads and $d_h = d/h$. For head $m$,

$$\text{head}_m = \text{Softmax}\Big(\frac{\tilde{H} F_m^Q (\tilde{H} F_m^K)^\top}{\sqrt{d_h}}\Big) \tilde{H} F_m^V, \quad \text{MHA}(\tilde{H}) = \big[\text{head}_1 \| \cdots \| \text{head}_h\big] F^O, \tag{7}$$

and we set

$$H_{\text{coevo}} = \tilde{H} + \alpha \, \text{MHA}(\tilde{H}), \quad \alpha > 0. \tag{8}$$

**Conservation and fitness heads.** Because $\text{Softmax}$ produces a probability vector over $K$ classes, the output lies on the $(K-1)$-simplex:

$$\text{Softmax} : \mathbb{R}^K \to \Delta^{K-1}, \qquad \Delta^{K-1} = \big\{ p \in \mathbb{R}^K \mid p_a \geq 0, \sum_{a=1}^{K} p_a = 1 \big\}. \tag{9}$$

With 20 amino acids ($K = 20$), the output lies on

$$\Delta^{19} = \big\{ p \in \mathbb{R}^{20} \mid p_a \geq 0, \sum_{a=1}^{20} p_a = 1 \big\}. \tag{10}$$

Hence, position-wise conservation preferences are produced by a small MLP. For each position $i$,

$$P_{\text{cons},i} = \text{Softmax}\big(V_c \, \phi(F_c \, H_{\text{coevo},i})\big) \in \Delta^{19}, \tag{11}$$

and the self-supervised fitness score pools the designed region

$$F(H_{\text{coevo}}) = \sigma\big(v_f^\top \, \phi(F_f \, \text{Pool}(H_{\text{coevo}}, M))\big) \in (0, 1). \tag{12}$$

**Losses.** In the current study, we *do not* use external MSA/PLM priors or MSA-depth gating. Instead, we combine a self-supervised evolutionary fitness target with entropy regularization of the learned conservation distribution:

$$\mathcal{L}_{\text{fit}} = \text{SmoothL1}\big(F_g(H_{\text{coevo}}), \tau_{\text{fit}}\big), \quad \tau_{\text{fit}} = 0.8, \tag{13}$$

$$\mathcal{L}_{\text{ent}} = -\frac{1}{\sum_i M_i} \sum_i M_i \Big[ \sum_a P_{\text{cons},i,a} \log P_{\text{cons},i,a} \Big], \tag{14}$$

so minimizing $\mathcal{L}_{\text{ent}}$ maximizes the entropy of $P_{\text{cons}}$ and encourages diversity. When decoder logits $q_{\text{pred},i}$ are available, we additionally add a local alignment term

$$\mathcal{L}_{\text{KL-local}} = \frac{1}{\sum_i M_i} \sum_i M_i \, \text{KL}\big(q_{\text{pred},i} \,\|\, P_{\text{cons},i}\big), \tag{15}$$

The KL divergence term encourages the latent space to remain compact and well-structured, preventing posterior drift and collapse. The total evolutionary objective is

$$\mathcal{L}_{\text{evo}} = \lambda_{\text{fit}}\mathcal{L}_{\text{fit}} + \lambda_{\text{ent}}\mathcal{L}_{\text{ent}} + \lambda_{\text{KL}}\mathcal{L}_{\text{KL-local}} \, . \tag{16}$$

*Taken together, evolutionary guidance biases peptide design toward conserved motifs and globally fit sequences, narrowing the search space to biologically plausible candidates while still encouraging diversity.*

Further details are provided in Appendix P.5

### 3.2.3 MUTUAL INFORMATION REGULARIZATION

A functional peptide is not just a plausible sequence or a plausible structure — the two must be aligned. Inspired by (Belghazi et al., 2018), to ensure coherence, we maximize the mutual information (MI) between sequence and structure embeddings. This encourages our model to generate sequences that "make sense" in the structural context.

**MINE objective and physics validity.** We pool embeddings from both sequence $H_{\text{coevo}}$ and structure embedding $z_{\text{struct}}$, then compute summaries $s = f_s(\text{Pool}(H_{\text{coevo}}, M))$, $z = f_x(\text{Pool}(z_{\text{struct}}, M))$. We train a critic $T_\theta$:

$$\widehat{I}_\theta = \mathbb{E}[T_\theta(s, z)] - \log \mathbb{E}[e^{T_\theta(s, z')}], \qquad \mathcal{L}_{\text{MI}} = -\widehat{I}_\theta. \tag{17}$$

Additionally, we include an auxiliary head $p_{\text{phys}}$ that predicts whether a structure is physically valid from its latent embedding $z$. This gives an extra push toward physically sensible outputs.

**MI loss.**

$$\mathcal{L}_{\text{MI-total}} = \lambda_{\text{MI}} \, \mathcal{L}_{\text{MI}} + \lambda_{\text{MI-phys}} \, \text{MSE}(p_{\text{phys}}, 1). \tag{18}$$

*Thus, MI regularization aligns sequence semantics with structural intent, reducing incoherent designs and promoting functional alignment.*

Further details are provided in Appendix P.6 and Appendix M

### 3.2.4 LATENT INPAINTING DIFFUSION

We represent the latent at step $t$ as $\mathbf{z}_t = (\mathbf{z}_{H,t}, \mathbf{z}_{X,t})$, stacking sequence and structure components. Redesign is localized by a binary residue mask $M \in \{0, 1\}^L$: noise and supervision are applied only where $M = 1$. The denoiser conditions on positional encodings, atom features, and $M$, and predicts $\hat{\varepsilon}_t = (\hat{\varepsilon}_{H,t}, \hat{\varepsilon}_{X,t})$.

**Forward noising (masked inpainting).** We use a variance-preserving cosine schedule with $\alpha_t = 1 - \beta_t$ and $\bar{\alpha}_t = \prod_{s=1}^t \alpha_s$. The forward process is

$$q(\mathbf{z}_t \mid \mathbf{z}_0) = \sqrt{\bar{\alpha}_t}\,\mathbf{z}_0 + \sqrt{1 - \bar{\alpha}_t}\,\varepsilon_t, \qquad \varepsilon_t \sim \mathcal{N}(\mathbf{0}, \mathbf{I}), \tag{19}$$

realized as latent *inpainting* by adding noise only where $M = 1$:

$$\mathbf{z}_t = M \odot \left(\sqrt{\bar{\alpha}_t}\,\mathbf{z}_0 + \sqrt{1 - \bar{\alpha}_t}\,\varepsilon_t\right) + (1 - M) \odot \mathbf{z}_0. \tag{20}$$

**Masked diffusion loss.** To supervise only redesigned residues, we broadcast $M$ over latent dimensions:

$$M_X := M \otimes \mathbf{1}_{n_{\text{lat}} \times 3}, \qquad M_H := M \otimes \mathbf{1}_{d_h}. \tag{21}$$

For $t \sim \mathcal{U}\{1, \ldots, T\}$ and $\varepsilon_t \sim \mathcal{N}(0, I)$, the masked noise-prediction loss is

$$\mathcal{L}_{\text{diff}}(t) = \frac{1}{\sigma_t^2}\left(\lambda_H \frac{\left\|(\varepsilon_{H,t} - \hat{\varepsilon}_{H,t}) \odot M_H\right\|_2^2}{\|M_H\|_1} + \lambda_X \frac{\left\|(\varepsilon_{X,t} - \hat{\varepsilon}_{X,t}) \odot M_X\right\|_2^2}{\|M_X\|_1}\right), \tag{22}$$

where $\sigma_t^2 = \frac{1 - \bar{\alpha}_{t-1}}{1 - \bar{\alpha}_t}\beta_t$ for the cosine schedule.

**Training the denoiser.** We learn parameters by minimizing the diffusion loss together with physics, evolutionary, and MI objectives.

$$\theta^\star = \arg \min_\theta \mathbb{E}_{t, \mathbf{z}_0, \boldsymbol{\varepsilon}_t}\Big[\mathcal{L}_{\text{diff}}(t) \ + \ \underbrace{\mathcal{L}_{\text{phys}}}_{\text{structure quality}} \ + \ \underbrace{\mathcal{L}_{\text{evo}}}_{\text{sequence viability}} \ + \ \underbrace{\mathcal{L}_{\text{MI-total}}}_{\text{sequence–structure consistency}}\Big]. \quad (23)$$

Here, $\mathcal{L}_{\text{diff}}(t)$ depends explicitly on the sampled diffusion timestep $t$, whereas $\mathcal{L}_{\text{phys}}, \mathcal{L}_{\text{evo}}, \mathcal{L}_{\text{MI-total}}$ and are time-independent regularizers computed on decoded or pooled representations for each sample. .At inference, parameters are frozen, and we write

$$\hat{\boldsymbol{\varepsilon}}_t := \varepsilon_{\theta^\star}(\mathbf{z}_t, t) = \big(\varepsilon_{H,\theta^\star}(\mathbf{z}_t, t), \ \varepsilon_{X,\theta^\star}(\mathbf{z}_t, t)\big),$$

so evolutionary and MI guidance act via $\theta^\star$. The $\lambda$ ablation can be found in Appendix I

**Reverse diffusion with explicit physics guidance.** Starting from $z_T \sim \mathcal{N}(\mathbf{0}, \mathbf{I})$, we iteratively update to $\mathbf{z}_0$. We apply a physics correction only to the structural component:

$$\tilde{\varepsilon}_{H,t} = \varepsilon_{H,\theta^\star}(\mathbf{z}_t, t), \quad (24)$$

$$\tilde{\varepsilon}_{X,t} = \varepsilon_{X,\theta^\star}(\mathbf{z}_t, t) - \sqrt{1 - \bar{\alpha}_t}\, G_t^{\text{phys}}, \qquad G_t^{\text{phys}} = -\lambda_{\text{phys}} \nabla_{\mathbf{z}_{X,t}} E_{\text{phys}}(\widehat{X}_t, S; M), \quad (25)$$

where $\widehat{X}_t$ is a partial decode of $\mathbf{z}_{X,t}$ (coordinates only). Because $E_{\text{phys}}$ depends only on internal distances/angles, this guidance is SE(3)-consistent. We anneal $\lambda_{\text{phys}}$ to strengthen physics late in the trajectory. The Gaussian reverse transition is

$$\mathbf{z}_{t-1} = \frac{1}{\sqrt{\alpha_t}}\bigg(\mathbf{z}_t - \frac{1 - \alpha_t}{\sqrt{1 - \bar{\alpha}_t}}\, \tilde{\varepsilon}_t\bigg) + \sigma_t\,\boldsymbol{\xi}, \qquad \boldsymbol{\xi} \sim \mathcal{N}(\mathbf{0}, \mathbf{I}), \quad (26)$$

where $\boldsymbol{\xi}$ is an independent standard normal (resampled at each step) with the same shape as $\mathbf{z}_t$.

**Stochasticity and context control.** To confine randomness to redesigned residues, we optionally mask the noise:

$$\boldsymbol{\xi} \leftarrow M \odot \boldsymbol{\xi} \quad \Rightarrow \quad \sigma_t\,\boldsymbol{\xi} \leftarrow \sigma_t\,(M \odot \boldsymbol{\xi}).$$

To keep the context fixed, we clamp unmasked entries after each step:

$$\mathbf{z}_{t-1} \leftarrow M \odot \mathbf{z}_{t-1} \ + \ (1 - M) \odot \mathbf{z}_0. \quad (27)$$

Running equation 26–equation 27 for $T$ steps yields $(H_0, X_0)$ while preserving the unmasked structural context. Evolutionary and MI guidance influence sampling through $\theta^\star$, whereas the explicit physics term stabilizes local geometry and reduces clashes during generation.

*Together, the proposed tri-guidance objective integrates diffusion with physical, evolutionary, and MI regularization, producing peptides that are physically stable, biologically grounded, and sequence–structure coherent.*

Further details are provided in Appendix P.2

## 4 EXPERIMENTS

### 4.1 DATASETS

In our experiments, the dataset primarily consists of short peptides, with a substantial proportion shorter than 30 amino acids. This characteristic highlights the model's strength in generating such sequences (Wei et al., 2024). Following the recommendations of (Kong et al., 2024), we employed two experimental setups: **Cross-domain**: To assess the model's generalization capability, we trained on the PepBench dataset, which contains 6,105 non-redundant protein–peptide complexes, and evaluated on the non-redundant dataset (LNR) from Tsaban (Tsaban et al., 2022), comprising 93 protein–peptide complexes with canonical amino acids curated by domain experts. **In-domain**: Using PepBDB (Wen et al., 2019), which includes 7,014 complexes. We ensured that no protein target was duplicated between the training and test sets, thereby preventing data leakage and enabling a fair evaluation. To achieve this, we applied the MMseqs2 clustering technique.

Further details of the datasets are provided in Appendix O and Appendix O.1.

## 4.2 EVALUATION METRICS

To rigorously assess the performance of the proposed methods against state-of-the-art (SOTA) models, we employed a comprehensive set of evaluation metrics capturing both structural quality and functional relevance. These include: **Success Rate** (fraction of generated peptides with thermodynamically stable binding, defined as $\Delta G < -5$ REU from Rosetta (Alford et al., 2017)), **Binding Free Energy** ($\Delta G$), **DockQ** (interface quality), **GDT_TS** (global structural similarity), **Contact_F1** (local interaction accuracy), **Local RMSD** (local structural precision), **Clash_in** (C$\alpha$ internal clashes), **Clash_out** (C$\alpha$ interface clashes), **Bond-outlier rate** (fraction of backbone bonds deviating from ideal geometry), **Sequence Diversity** (BLOSUM62 clustering), **Sequence Validity** (fraction of generated sequences passing biochemical criteria), **Structure Diversity** (RMSD clustering), and **Consistency** (Cramér's V across multiple generations). To capture variability across targets, we additionally report per-target standard deviations for six primary metrics.

Further details of evaluation metrics are provided in Appendix Q.

## 4.3 SEQUENCE-STRUCTURE PEPTIDE CO-DESIGN TASK

During training and evaluation, peptide design was carried out *in situ* within the receptor's binding pocket. The receptor was treated as a rigid scaffold, and all diffusion steps for the peptide sequence and structure were conditioned on its local environment. This setup ensures that generated peptides are evaluated in the same geometric and energetic context in which binding occurs. We report each metric for before relaxation and after relaxation (shown in italics). Geometry refinement was carried out using OpenMM energy minimization with the Amber14 force field, applied consistently across our models and baseline models. To preserve backbone geometry, positional restraints with a stiffness constant of 10.0 were applied to all non-hydrogen atoms. The minimizer was run with no iteration cap (maximum iterations = 0), allowing convergence to a local minimum.

### 4.3.1 BINDING QUALITY, INTERFACE ASSESSMENT, AND STRUCTURAL ACCURACY

Table 1: Binding quality metrics on PepBench and PepBDB. Higher success rate and DockQ, lower $\Delta G$, are better. (without relaxation / *with relaxation*)

| Dataset | Method | Success Rate (↑) | $\Delta G$ (REU) ↓ | DockQ (↑) |
|---|---|---|---|---|
| PepBench | PepGLAD | $0.29_{\pm 0.19}$ / *$0.79_{\pm 0.17}$* | $-15.63_{\pm 8.51}$ / *$-34.48_{\pm 12.44}$* | $\underline{0.60}_{\pm 0.15}$ / *$\underline{0.59}_{\pm 0.14}$* |
| | PepFlow | $0.31_{\pm 0.19}$ / *$0.74_{\pm 0.13}$* | $-17.05_{\pm 8.25}$ / *$\underline{-35.98}_{\pm 18.81}$* | $0.53_{\pm 0.11}$ / *$0.42_{\pm 0.09}$* |
| | UniMoMo$_{single}$ | $\underline{0.34}_{\pm 0.19}$ / *$\underline{0.79}_{\pm 0.13}$* | $\underline{-19.04}_{\pm 7.17}$ / *$-30.19_{\pm 9.55}$* | $0.57_{\pm 0.23}$ / *$0.54_{\pm 0.19}$* |
| | **PepTri(Ours)** | $\mathbf{0.40}_{\pm 0.19}$ / *$\mathbf{0.83}_{\pm 0.16}$* | $\mathbf{-19.39}_{\pm 7.08}$ / *$\mathbf{-36.36}_{\pm 14.27}$* | $\mathbf{0.63}_{\pm 0.16}$ / *$\mathbf{0.62}_{\pm 0.15}$* |
| PepBDB | PepGLAD | $0.15_{\pm 0.12}$ / *$0.67_{\pm 0.26}$* | $-14.48_{\pm 9.91}$ / *$-31.22_{\pm 13.28}$* | $0.43_{\pm 0.20}$ / *$0.43_{\pm 0.19}$* |
| | PepFlow | $0.30_{\pm 0.15}$ / *$0.66_{\pm 0.31}$* | $-17.44_{\pm 9.52}$ / *$\mathbf{-34.98}_{\pm 25.97}$* | $0.41_{\pm 0.21}$ / *$0.31_{\pm 0.15}$* |
| | UniMoMo$_{single}$ | $\underline{0.30}_{\pm 0.22}$ / *$\underline{0.74}_{\pm 0.25}$* | $\mathbf{-18.89}_{\pm 12.59}$ / *$-34.05_{\pm 18.89}$* | $\underline{0.44}_{\pm 0.20}$ / *$\underline{0.43}_{\pm 0.18}$* |
| | **PepTri(Ours)** | $\mathbf{0.31}_{\pm 0.23}$ / *$\mathbf{0.74}_{\pm 0.27}$* | $\underline{-18.15}_{\pm 11.92}$ / *$\underline{-34.82}_{\pm 18.20}$* | $\mathbf{0.49}_{\pm 0.21}$ / *$\mathbf{0.49}_{\pm 0.19}$* |

Table 1 summarizes binding success, thermodynamic favorability, and interface quality across both cross-domain (PepBench) and in-domain (PepBDB) evaluations. PepTri achieves the strongest overall performance, outperforming baselines in success rate and interface quality and maintaining highly competitive $\Delta G$ values in both pre- and post-relaxation settings. This indicates robust generalization and stable peptide–receptor interfaces even under relaxation. Relaxation improves force-field energy but does not guarantee increased nativeness, often shifting poses toward lower-energy basins with slightly reduced DockQ. PepTri's guided denoising produces interfaces that remain native-like after refinement, suggesting that its tri-guidance mechanism shapes physically coherent structures rather than relying on post-hoc relaxation to correct them.

### 4.3.2 STRUCTURAL ACCURACY

Table 2 reports that PepTri consistently achieves the highest Contact_F1 and GDT_TS on both PepBench and PepBDB in both pre- and post-relaxation regimes. For local RMSD, PepFlow is slightly better before relaxation, whereas PepTri attains the best post-relaxation RMSD on both datasets, indicating that its conformations refine particularly well. Since GDT_TS > 0.5 indicates reasonable

Table 2: Structural accuracy metrics on PepBench and PepBDB (without relaxation / *with relaxation*).

| Dataset | Method | Contact_F1 (↑) | Local RMSD (Å) (↓) | GDT_TS (↑) |
|---|---|---|---|---|
| PepBench | PepGLAD | $0.80_{\pm0.26}$ / *$0.80_{\pm0.24}$* | $1.22_{\pm0.49}$ / *$1.21_{\pm0.43}$* | $0.72_{\pm0.20}$ / *$0.73_{\pm0.19}$* |
| | PepFlow | $\underline{0.80}_{\pm0.25}$ / *$\underline{0.82}_{\pm0.19}$* | $\mathbf{1.07}_{\pm0.40}$ / *$\mathbf{1.06}_{\pm0.48}$* | $\underline{0.74}_{\pm0.20}$ / *$\underline{0.74}_{\pm0.21}$* |
| | UniMoMo$_{single}$ | $0.61_{\pm0.35}$ / *$0.67_{\pm0.32}$* | $1.98_{\pm1.64}$ / *$1.37_{\pm1.27}$* | $0.62_{\pm0.27}$ / *$0.62_{\pm0.27}$* |
| | **PepTri(Ours)** | $\mathbf{0.83}_{\pm0.23}$ / *$\mathbf{0.84}_{\pm0.20}$* | $\underline{1.18}_{\pm0.42}$ / *$\underline{1.10}_{\pm0.46}$* | $\mathbf{0.75}_{\pm0.18}$ / *$\mathbf{0.76}_{\pm0.19}$* |
| PepBDB | PepGLAD | $0.52_{\pm0.36}$ / *$0.60_{\pm0.35}$* | $1.92_{\pm1.83}$ / *$1.46_{\pm0.48}$* | $0.52_{\pm0.26}$ / *$0.60_{\pm0.25}$* |
| | PepFlow | $\underline{0.71}_{\pm0.30}$ / *$\underline{0.72}_{\pm0.28}$* | $\mathbf{1.27}_{\pm0.45}$ / *$\underline{1.35}_{\pm0.44}$* | $\underline{0.65}_{\pm0.21}$ / *$\underline{0.63}_{\pm0.22}$* |
| | UniMoMo$_{single}$ | $0.48_{\pm0.36}$ / *$0.55_{\pm0.35}$* | $2.78_{\pm1.50}$ / *$1.52_{\pm0.65}$* | $0.48_{\pm0.24}$ / *$0.56_{\pm0.24}$* |
| | **PepTri(Ours)** | $\mathbf{0.75}_{\pm0.29}$ / *$\mathbf{0.77}_{\pm0.26}$* | $\underline{1.34}_{\pm1.43}$ / *$\mathbf{1.29}_{\pm0.41}$* | $\mathbf{0.66}_{\pm0.20}$ / *$\mathbf{0.67}_{\pm0.20}$* |

Table 3: Clash and geometry quality metrics on PepBench and PepBDB (without relaxation / *with relaxation*). Lower is better for all metrics.

| Dataset | Method | Clash_in ↓ (%) | Clash_out ↓ (%) | Bond Outliers ↓ (%) |
|---|---|---|---|---|
| PepBench | PepGLAD | $7.99_{\pm11.28}$ / *$1.60_{\pm5.91}$* | $6.08_{\pm12.36}$ / *$1.45_{\pm6.48}$* | $17.41_{\pm9.32}$ / *$6.53_{\pm3.08}$* |
| | PepFlow | $7.64_{\pm9.64}$ / *$0.70_{\pm2.36}$* | $7.82_{\pm14.05}$ / *$\underline{0.69}_{\pm3.74}$* | $19.64_{\pm12.77}$ / *$7.20_{\pm1.98}$* |
| | UniMoMo$_{single}$ | $\mathbf{5.99}_{\pm14.79}$ / *$1.08_{\pm4.47}$* | $5.55_{\pm11.35}$ / *$0.89_{\pm4.84}$* | $\mathbf{14.95}_{\pm11.22}$ / *$\underline{5.79}_{\pm2.39}$* |
| | **PepTri (Ours)** | $\underline{6.16}_{\pm13.65}$ / *$\mathbf{0.59}_{\pm4.59}$* | $\mathbf{4.73}_{\pm11.06}$ / *$\mathbf{0.54}_{\pm3.54}$* | $\underline{15.60}_{\pm10.48}$ / *$\mathbf{5.47}_{\pm1.73}$* |
| PepBDB | PepGLAD | $20.36_{\pm12.35}$ / *$5.44_{\pm6.07}$* | $8.52_{\pm16.64}$ / *$1.10_{\pm6.81}$* | $37.13_{\pm8.39}$ / *$7.28_{\pm2.44}$* |
| | PepFlow | $19.43_{\pm14.88}$ / *$2.48_{\pm2.72}$* | $12.45_{\pm13.35}$ / *$1.70_{\pm4.66}$* | $28.93_{\pm4.50}$ / *$5.17_{\pm1.60}$* |
| | UniMoMo$_{single}$ | $\mathbf{15.07}_{\pm11.18}$ / *$\mathbf{0.83}_{\pm6.45}$* | $\underline{7.82}_{\pm17.49}$ / *$\underline{1.45}_{\pm7.56}$* | $32.13_{\pm3.20}$ / *$\mathbf{4.46}_{\pm1.95}$* |
| | **PepTri (Ours)** | $\underline{16.72}_{\pm12.28}$ / *$1.25_{\pm7.31}$* | $\mathbf{6.22}_{\pm12.28}$ / *$\mathbf{0.71}_{\pm5.19}$* | $\mathbf{28.27}_{\pm8.54}$ / *$4.70_{\pm1.10}$* |

structural similarity, these results confirm that our generated peptides remain globally faithful to native folds, with the largest gains on PepBDB. Complementing these structural metrics, Table 3 summarizes clash and covalent-geometry outlier rates (lower is better for all metrics). Energy minimization substantially reduces clashes and bond-length violations for all methods, showing that the predicted backbones lie close to valid local minima. After relaxation, PepTri attains the best or near-best Clash_in, Clash_out, and bond-outlier rates across both datasets, yielding peptide structures that are not only accurate in terms of global fold but also highly clash-free and geometrically well-formed.

### 4.3.3 DIVERSITY ANALYSIS AND SEQUENCE VALIDITY RATE

Table 4: Design diversity metrics on PepBench and PepBDB (without relaxation / *with relaxation*).

| Dataset | Method | SeqDiv (↑) | SeqValid(↑) | StrDiv(↑) | Consistency(↑) |
|---|---|---|---|---|---|
| PepBench | PepGLAD | 0.92 | $\underline{0.27}$ | $\underline{0.54}$ / *$\underline{0.60}$* | **0.81** / *0.84* |
| | PepFlow | 0.83 | 0.25 | 0.44 / *0.58* | 0.67 / *0.80* |
| | UniMoMo$_{single}$ | $\underline{0.92}$ | 0.21 | **0.62** / *$\mathbf{0.65}$* | 0.79 / *$\underline{0.84}$* |
| | **PepTri (Ours)** | **0.93** | **0.27** | 0.44 / *0.63* | $\underline{0.80}$ / *$\mathbf{0.86}$* |
| PepBDB | PepGLAD | $\underline{0.92}$ | $\underline{0.24}$ | $\underline{0.81}$ / *$\mathbf{0.82}$* | **0.94** / *$\mathbf{0.96}$* |
| | PepFlow | 0.67 | 0.20 | 0.58 / *0.71* | 0.74 / *0.82* |
| | UniMoMo$_{single}$ | 0.90 | 0.20 | **0.81** / *$\underline{0.80}$* | $\underline{0.92}$ / *0.94* |
| | **PepTri (Ours)** | **0.93** | **0.28** | 0.63 / *0.70* | 0.89 / *$\underline{0.95}$* |

Table 4 reports sequence and structure diversity, sequence validity, and sequence–structure consistency. PepTri attains the strongest overall balance across these metrics, producing diverse yet biologically plausible sequences and coherent sequence–structure pairs. While its structural diversity is moderate, this reflects its tendency to generate low-energy, near-native conformational ensembles. PepTri also achieves the highest or near-highest consistency after relaxation, indicating that evolutionary guidance helps maintain coherence between sequence and fold while still supporting broad sequence exploration.

**Practical implications.** Results show a balanced, physics-constrained exploration that preserves sequence diversity while converging to stable structural basins, benefiting peptide design and drug discovery; **Cross-domain robustness**: our PepTri achieves state-of-the-art results on PepBench binding metrics, evidencing tri-guidance under distribution shift; **Refinement-friendly geometry**: best post-relax RMSD on both datasets indicates that SE(3)-aware latent modeling with physics guidance reaches physically consistent minima; **Diverse yet coherent designs**: highest sequence diversity alongside strong sequence–structure consistency validates mutual-information coupling of sequence and structure latents.

## 5 ABLATION STUDY ANALYSIS

*Setup.* We quantify the contributions of four components: physics guidance, evolutionary guidance, mutual information (MI) guidance, and all-atom modeling—using single-component removals (Table 5) and single-guidance variants (Table 7). Unless stated otherwise, higher values are better; for Local_RMSD_Mean and Sliding-AAR, lower is better. Our ablation study evaluates the high-level contribution of each guidance type (physics, evolutionary, and MI) because these components are designed to function as integrated modules. The full model, PepTri, achieves the strongest overall performance, with leading scores in mean success rate, $\Delta G$, DockQ, Contact F1, GDT-TS, sequence validity, and consistency. Removing any component degrades at least one core dimension of quality, underscoring the *synergy* between structural physics, evolutionary constraints, information-theoretic coupling, and atomistic detail.

Table 5: Ablation study results comparing different settings when removing single components .

|  | No_phys | No_evo | No_mi | PepTri$_{backbone}$ | **PepTri** |
|---|---|---|---|---|---|
| Physics guidance (phys) |  | ✓ | ✓ | ✓ | ✓ |
| Evolutionary guidance (evo) | ✓ |  | ✓ | ✓ | ✓ |
| Mutual information guidance (mi) | ✓ | ✓ |  | ✓ | ✓ |
| All atom | ✓ | ✓ | ✓ |  | ✓ |
| Mean success rate ($\Delta G < 0$) | 0.401 | 0.443 | 0.545 | 0.397 | **0.583** |
| $\Delta G$ (REU) ↓ | -15.485 | -16.501 | -18.949 | -16.961 | **-19.387** |
| DockQ | 0.621 | 0.618 | 0.633 | 0.578 | **0.633** |
| Contact_F1 | 0.750 | 0.769 | 0.804 | 0.760 | **0.829** |
| Local RMSD (Å) ↓ | 1.432 | 1.418 | **1.154** | 1.260 | 1.179 |
| GDT_TS | 0.704 | 0.709 | 0.745 | 0.726 | **0.747** |
| Sequence Diversity | 0.917 | 0.915 | 0.920 | 0.917 | **0.926** |
| Sequence Validity rate | 0.256 | 0.250 | 0.259 | 0.260 | **0.273** |
| Struct Diversity | 0.465 | 0.427 | 0.431 | **0.499** | 0.436 |
| Consistency | 0.783 | 0.771 | 0.779 | 0.744 | **0.799** |
| Sliding-AAR | 0.352 | **0.361** | 0.354 | 0.347 | 0.342 |
| TM-score | 0.221 | 0.220 | **0.250** | 0.242 | 0.244 |

## 6 CONCLUSION

We introduced PepTri, a latent-diffusion framework for peptide co-design that operates in an SE(3)-aware latent space with separate sequence and structure representations. Joint denoising preserves geometric equivariance and enables efficient control over both modalities. PepTri incorporates three complementary forms of guidance: physics-based gradients that enforce geometric plausibility, evolutionary priors that encourage realistic sequences, and mutual-information regularization that promotes coherent sequence–structure design. These components together produce stable sampling behavior and high-quality peptide candidates. Overall, PepTri provides a principled way to integrate physical and evolutionary priors into equivariant generative models, with clear opportunities for extension to broader conditioning and to more complex protein–peptide systems.

ACKNOWLEDGEMENTS

This research was supported by a grant of the Korea Machine Learning Ledger Orchestration for Drug Discovery Project(K-MELLODDY), funded by the Ministry of Health & Welfare and Ministry of Science and ICT, Republic of Korea (grant number: RS-2024-00462471). This work was supported by the Technology Development Program(RS-2024-00523644) funded by the Ministry of SMEs and Startups(MSS, Korea). This research was supported by a grant from the Korea Health Technology R & D Project through the Korea Health Industry Development Institute (KHIDI), funded by the Ministry of Health & Welfare, Republic of Korea (grant number: RS-2025-25462758). We also thank Soojung Lee [1] for her expertise, which significantly enhanced the quality of this work.

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

## USE OF LARGE LANGUAGE MODELS

We used a large language model to (i) polish grammar and wording across drafts,(ii) assist with high-level ideation and code, and (iii) find related works. All technical claims, equations, experiments, and analyses were conceived, implemented, and verified by the authors. The model was not used to generate data or run experiments. All citations and numerical results were manually checked by the authors.

## A  NOTATION

Table 6: Summary of notations used throughout the paper.

| Symbol | Meaning |
|---|---|
| $S \in \{0, \ldots, 19\}^L$ | Amino acid sequence of length $L$ (20 canonical residues) |
| $X \in \mathbb{R}^{L \times C \times 3}$ | All-atom 3D coordinates ($C = 14$ channels: backbone + sidechains) |
| $M \in \{0, 1\}^L$ | Binary mask for inpainting ($M_i = 1$ redesigned, $M_i = 0$ context) |
| $h_i, H$ | Residue-level embeddings; $H \in \mathbb{R}^{L \times d}$ |
| $z_h \in \mathbb{R}^{L \times d_h}$ | Latent sequence features (invariant) |
| $z_x \in \mathbb{R}^{L \times n_{\text{lat}} \times 3}$ | Latent structural anchors (equivariant) |
| $\mathbf{z}_t = (z_{H,t}, z_{X,t})$ | Latent variables at diffusion step $t$ |
| $\varepsilon_\theta$ | Denoiser network parameterized by $\theta$ |
| $\hat{\varepsilon}_t$ | Predicted noise at step $t$ |
| $\tilde{\varepsilon}_t$ | Guided noise with physics correction |
| $\alpha_t, \beta_t, \bar{\alpha}_t$ | Diffusion schedule coefficients (cosine schedule) |
| $\sigma_t$ | Variance term in reverse diffusion |
| $\boldsymbol{\xi}$ | Gaussian noise sampled at each diffusion step |
| $\zeta_{ijk}$ | Bond angle at residue $j$ between atoms $i$ and $k$ |
| $E_{\text{phys}}$ | Physics energy (bond, angle, vdW, electrostatics, etc.) |
| $G_t^{\text{phys}}$ | Physics gradient guidance at step $t$ |
| $P_{\text{cons},i}$ | Position-specific conservation distribution at residue $i$ |
| $F(H_{\text{coevo}})$ | Global self-supervised evolutionary fitness score |
| $\mathcal{L}_{\text{VAE}}$ | VAE training loss (reconstruction + KL + geometry) |
| $\mathcal{L}_{\text{diff}}$ | Diffusion noise prediction loss |
| $\mathcal{L}_{\text{phys}}$ | Physics-informed loss |
| $\mathcal{L}_{\text{evo}}$ | Evolutionary guidance loss |
| $\mathcal{L}_{\text{MI}}$ | Mutual-information loss |
| $\theta^\star$ | Trained PepTri parameters |

## B  THREE GUIDANCE DISCUSSIONS

### B.1  PHYSICS GUIDANCE: ENFORCING ENERGETIC FEASIBILITY

Imposes molecular mechanics constraints so sampled conformations are sterically and energetically plausible.
**Ablation Impact (No_phys).** This ablation causes the *largest* degradation in binding strength: mean success rate drops by $31.2\%$ ($0.583 \rightarrow 0.401$) and $\Delta G$ weakens ($-19.387 \rightarrow -15.485$). Contact-level accuracy also declines (Contact_F1: $0.829 \rightarrow 0.750$; DockQ: $0.633 \rightarrow 0.621$).
**Why indispensable.** Physics guidance anchors generation to biophysical reality; without it, designs drift toward *energetically unstable* states that are unlikely to bind.

### B.2  EVOLUTIONARY GUIDANCE: MAINTAINING BIOLOGICAL PLAUSIBILITY

Uses conservation signals to steer designs toward foldable, functionally plausible sequence distributions.

Table 7: Comparison of models with single guidance across evaluation metrics.

| | Noevo_Nomi | Nophys_Nomi | Noevo_Nophys | PepTri |
|---|---|---|---|---|
| Physics guidance (phys) | ✓ | | | ✓ |
| Evolutionary guidance (evo) | | ✓ | | ✓ |
| Mutual information guidance (mi) | | | ✓ | ✓ |
| All atom | ✓ | ✓ | ✓ | ✓ |
| Mean success rate ($\Delta G < 0$) | 0.534 | 0.363 | 0.369 | **0.583** |
| $\Delta G$ (REU) $\downarrow$ | -18.535 | -14.912 | -15.134 | **-19.387** |
| DockQ | 0.531 | 0.532 | 0.539 | **0.633** |
| Contact F1 | 0.803 | 0.738 | 0.722 | **0.829** |
| Local RMSD (Å) $\downarrow$ | 1.260 | 1.427 | 1.401 | **1.179** |
| GDT_TS | 0.745 | 0.704 | 0.701 | **0.747** |
| Sequence Diversity | 0.925 | 0.918 | 0.912 | **0.926** |
| Sequence Validity rate | 0.256 | 0.262 | 0.247 | **0.273** |
| Struct Diversity | 0.423 | 0.402 | 0.409 | **0.436** |
| Consistency | 0.797 | 0.742 | 0.716 | **0.799** |
| Sliding-AAR | 0.346 | **0.347** | 0.343 | 0.342 |
| TM-score | **0.256** | 0.214 | 0.208 | 0.244 |

**Ablation Impact (No_evo).** Removing evolutionary guidance harms interface and global fold metrics: Contact_F1 drops by 7.2% ($0.829 \rightarrow 0.769$), GDT-TS by 5.1% ($0.747 \rightarrow 0.709$), and consistency decreases ($0.799 \rightarrow 0.771$). Sequence validity decreases from 0.273 to 0.250.
**Why indispensable.** Evolutionary constraints curb exploration of physically permissible but *biologically irrelevant* sequences, improving validity and the reliability of the sequence→structure mapping.

### B.3    MUTUAL INFORMATION (MI) GUIDANCE: COUPLING SEQUENCE AND STRUCTURE

Maximizes MI between sequence and structure representations to promote coherent long-range dependencies and stable folding.
**Ablation Impact (No_mi).** A nuanced trade-off: removing MI slightly *improves* Local_RMSD_Mean ($1.179 \rightarrow 1.154$) and TM-score ($0.244 \rightarrow 0.250$), but *reduces* success rate ($0.583 \rightarrow 0.545$), Contact_F1 ($0.829 \rightarrow 0.804$), and consistency ($0.799 \rightarrow 0.779$).
**Why indispensable.** MI guidance strengthens global residue-residue coherence and contact fidelity. Even if local RMSD tightens marginally without MI, overall *stability and reliability* suffer.

### B.4    ALL-ATOM REPRESENTATION: SIDE-CHAIN RESOLUTION FOR PRECISION

Goes beyond backbone constraints to capture side-chain packing that determines interface quality.
**Ablation Impact (PepTri_backbone).** The backbone-only variant underperforms across contact-sensitive metrics (DockQ: $0.633 \rightarrow 0.578$, Contact_F1: $0.829 \rightarrow 0.760$) and consistency ($0.799 \rightarrow 0.744$), highlighting the need for atomistic detail.
**Why indispensable.** Binding is dictated by side-chain chemistry; atom-level modeling is required to evaluate and optimize interface specificity and packing.

### B.5    SINGLE-GUIDANCE VARIANTS: NO SINGLE SIGNAL IS SUFFICIENT

Observation. Physics-only, Evo-only, and MI-only models (Table 7) trail **PepTri** on nearly all metrics, especially DockQ, Contact F1, and consistency. For instance, the physics-only variant (*No-evo_Nomi*) attains Contact F1 = 0.803 and DockQ = 0.531, below PepTri (0.829 and 0.633). This confirms that *no single guidance signal* captures the multifaceted requirements of accurate, valid, and stable design.

### B.6 TAKEAWAY

Synergy is non-negotiable for high-performance peptide design. Our ablation study demonstrates that the integration of physical, evolutionary, and information-theoretic principles is paramount. For practitioners, this provides a clear hierarchy for model design:

1. **Anchor designs in physical reality.** Physics-based guidance is the most critical single component for predicting strong binding.

2. **Model at atomic resolution.** An all-atom representation is essential for achieving precise interface quality.

3. **Enforce global consistency.** Mutual information guidance is key to ensuring sequences reliably fold into their intended structures.

4. **Constrain for function.** Evolutionary guidance ensures designs are biologically plausible and evolutionarily informed.

While the physics+atom foundation is essential, the full integration of all four components in **PepTri** is required to simultaneously maximize binding affinity, structural accuracy, and biological validity.

## C DISCUSSION

Our physics-guided peptide design model demonstrates a compelling and biologically meaningful trade-off between structural quality and conformational diversity when compared to the state-of-the-art PepGLAD. While achieving substantial gains across most structural and binding quality metrics, our approach shows a moderate reduction in structural diversity—a result that offers valuable insight into how physics-based constraints influence generative peptide design. These improvements indicate that such constraints effectively guide the model toward energetically favorable and structurally realistic peptide conformations.

PepTri currently models canonical residues and peptides predominantly shorter than 30 aa. The MI estimator (MINE) can be high-variance; we mitigate this with EMA baselines and gradient clipping. Reported energies are computational proxies (Rosetta/OpenMM) and may not perfectly correlate with experimental affinities.

### C.1 QUALITY VS. DIVERSITY: A BENEFICIAL TRADE-OFF

The observed trade-off between structural quality and conformational diversity in our physics-guided peptide design model represents a beneficial optimization for therapeutic applications. The substantial improvements in binding prediction, structural accuracy, and biological consistency significantly outweigh the reduction in structural diversity. This finding supports the hypothesis that physics-based constraints serve as valuable inductive biases for peptide design, guiding models toward biologically relevant and therapeutically promising conformational space. (Ferruz et al., 2022; Dauparas et al., 2022).

**Biological Perspective** — Natural peptides do not explore the entirety of conformational space; instead, they preferentially adopt low-energy, functional conformations. Evolutionary pressures select for sequences that fold into stable, functional structures rather than maximizing conformational diversity. For therapeutic peptides, specific binding conformations are often required, making structural quality and binding accuracy more critical than maximizing diversity.

**Computational Perspective** — Physics-based constraints serve as strong inductive biases, reducing the exploration of unrealistic conformational space. By concentrating sampling efforts on physically plausible structures, the model improves computational efficiency while preserving functional relevance. In practice, improvements in structural quality and binding prediction directly translate into better drug design outcomes .

### C.2 ROLE OF EVOLUTIONARY GUIDANCE

Our ablation indicates that evolutionary guidance is a principal contributor to reliability and realism in PepTri. In Table 5, removing it (No_evo) reduces mean success rate ($0.583 \rightarrow 0.443$, -24%

rel.), weakens stability ( $\Delta G$ -19.387 $\rightarrow$ -16.501), and degrades contact accuracy and global fold quality (Contact_F1 0.829 $\rightarrow$ 0.769; GDT_TS 0.747 $\rightarrow$ 0.709), while slightly affecting diversity. In contrast, physics dominantly shapes stability and success, and MI adds modest but consistent gains in coherence; the full model combines these effects most effectively.

We hypothesize that three mechanisms underlie these gains. First, evolutionary embeddings blended via a learnable gate bias residue substitutions toward plausible regimes, reducing off-manifold proposals. Second, residue dependency attention captures residue–residue couplings that sharpen contact maps and local packing, improving Contact_F1 and GDT_TS. Third, a weak self-supervised evolutionary fitness regularizer calibrates broad biochemical profiles (e.g., charge, hydropathy), aiding foldability without collapsing diversity. A part of the Table 11 also indicates that maximizing evolutionary information benefits the model.

There are trade-offs and limits. No_evo shows a slight improvement in Sliding-AAR, suggesting that evolutionary priors may occasionally down-weight rare but compatible chemistries. Benefits can diminish when homologous information is sparse (shallow MSAs), and BLOSUM-like priors may bias novelty if over-weighted.

### C.3 LIMITATIONS AND FUTURE DIRECTIONS

#### C.3.1 LIMITATIONS

We highlight the main limitations of our current study:

- **Computational cost.** The all-atom representation combined with SE(3)-equivariant message passing is compute- and memory-intensive; physics-guided sampling further increases runtime relative to unguided diffusion.
- **Rigid receptor & missing environment.** Sampling assumes a fixed receptor. Induced-fit effects and explicit environment (solvent/water, ions, pH) are not modeled, which can limit realism at the interface. The known binding pocket information is required, which limits applicability to discovering novel binding sites or cryptic pockets on target receptors.
- **Length & multi-chain generalization.** Training focuses on peptide-scale systems; scalability to longer proteins and multi-chain assemblies remains untested.
- **Evaluation coverage.** Our docking/biophysics metrics (e.g., $\Delta G$, DockQ) are computational proxies for binding and fold quality. Prospective experimental validation of binding and function is outside the scope of this work.

#### C.3.2 OPTIMIZING THE QUALITY–DIVERSITY BALANCE

Future research could investigate strategies to retain quality improvements while recovering some degree of structural diversity:

1. **Adaptive Physics Weighting**: Dynamically adjust the physics loss weights during training to balance quality and diversity.
2. **Multi-Objective Optimization**: Simultaneously optimize for both quality metrics and diversity measures.
3. **Ensemble Methods**: Combine multiple physics-guided models with varying constraint strengths.
4. **Temperature Scaling**: Use temperature-controlled sampling to fine-tune the exploration–exploitation balance.

The success of our tri-guided approach highlights a clear pathway for further enhancement, particularly in enhancing the biological realism and applicability of the evolutionary guidance component:

- **Adaptive Evolutionary Guidance**: Implement a dynamic guidance schedule that explicitly manages the exploration-exploitation trade-off during denoising. By annealing the influence of evolutionary priors, the sampler could initially explore a diverse sequence landscape before converging to evolutionarily fit and conserved regions, potentially increasing the hit rate of functional designs.

- **Advanced Force Fields and Solvation Models**: Enhance the physical realism of generated structures by integrating more sophisticated force fields, polarizable charge models, and explicit solvation effects. This would provide a more accurate energetic landscape, particularly for designing peptides that function in specific cellular environments or require precise electrostatic interactions.

- **Allosteric and Long-Range Constraints**: Extend the physics-informed guidance to model allosteric mechanisms and long-range interactions explicitly. Incorporating constraints derived from molecular dynamics or elastic network models could capture the dynamic conformational changes essential for modulating protein function.

- **Integration of Experimental Data**: Incorporate experimental constraints (e.g., from NMR spectroscopy, X-ray crystallography, or cryo-EM densities) as structural restraints during the diffusion process. This would enable a closed-loop design pipeline where experimental data directly refines and validates generative proposals.

- **Richer evolutionary priors**: Explore integrating MSA- or PLM-based conservation signals to capture stronger position-specific constraints, while addressing their data availability and computational overhead. Developing lightweight or cached evolutionary features that remain applicable to short peptides and novel targets is an important direction for future work.

# D CORRELATIONS AMONG STRUCTURAL, ENERGETIC, AND SEQUENCE-LEVEL DESCRIPTORS

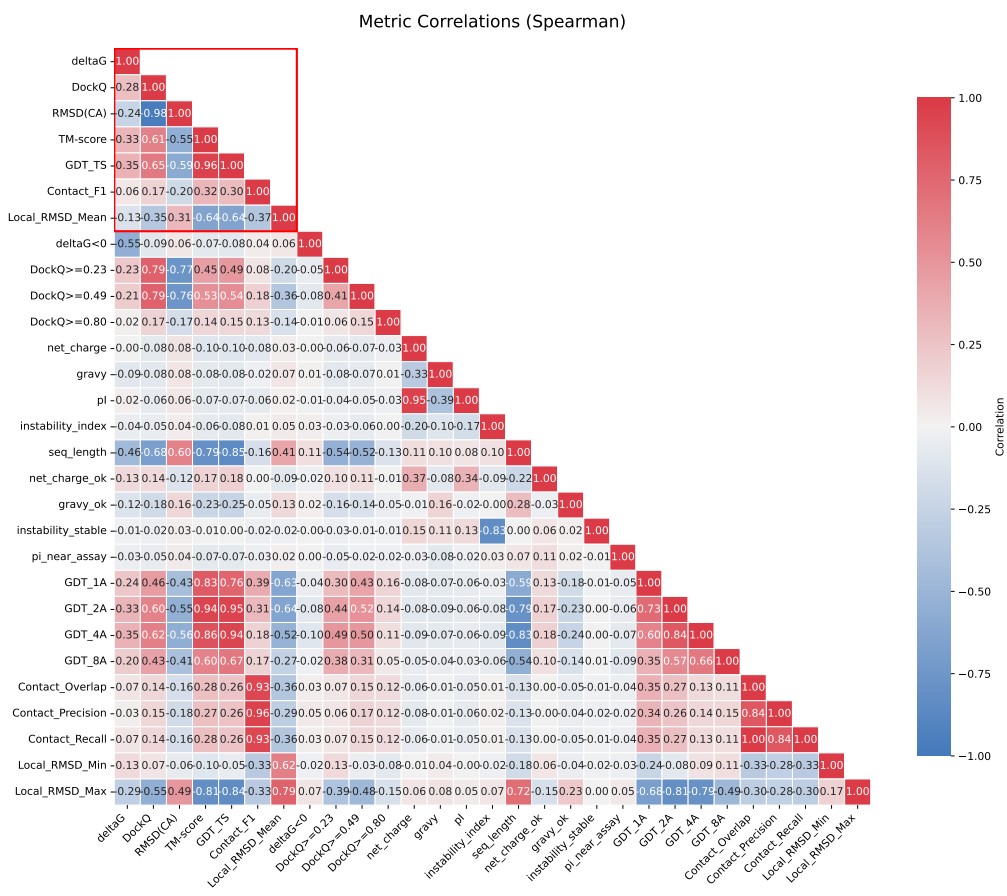

Figure 2: The paired metric heat map of results from PepTri for relaxed cross-domain experiment.

The pairwise correlation analysis shown in Fig. 2 reveals distinct patterns among geometric accuracy metrics, energetic scores, and sequence-derived descriptors.

**Structural quality metrics.** DockQ emerges as the most reliable indicator of structural correctness. It exhibits a nearly perfect inverse association with $C_\alpha$ RMSD ($\rho \approx -0.98$), indicating that lower backbone deviations directly translate to higher DockQ scores. DockQ also correlates strongly with TM-score ($\rho \approx +0.61$) and GDT ($\rho \approx +0.65$), confirming that models with high DockQ not only achieve accurate local geometry but also preserve global topology. Local RMSD statistics (mean, min, max) show moderate associations with TM-score and GDT ($\rho \approx +0.64$), but weaker alignment with DockQ, suggesting that localized backbone distortions contribute to global accuracy without directly determining interface quality. Contact-based metrics (precision, recall, F1, overlap) show only weak correlations with DockQ and RMSD ($\rho \leq 0.32$), indicating that recovering the correct set of contacts is insufficient to capture either precise geometry or overall structural fidelity.

**Energetics.** Rosetta interface energy ($\Delta G$) shows only a weak positive association with DockQ ($\rho \approx +0.28$). However, $\Delta G$ correlates moderately and negatively with RMSD ($\rho \approx -0.24$), TM-score ($\rho \approx -0.33$), and GDT ($\rho \approx -0.35$). Because more negative $\Delta G$ values represent more favorable binding energies, this pattern suggests that globally accurate structures tend to have somewhat better energetic profiles. Nevertheless, the magnitude of these correlations is insufficient for $\Delta G$ to serve as a primary ranking criterion.

**Sequence-level descriptors.** Most sequence-derived properties—including net charge, hydropathy (GRAVY), isoelectric point, and instability index—show negligible correlations with 3D accuracy metrics (DockQ, TM-score, GDT, RMSD, and contact F1). The one consistent exception is *sequence length*, which exhibits a negative correlation with DockQ ($\rho \approx -0.68$) and also with $\Delta G$ ($\rho \approx -0.46$), indicating that longer sequences tend to produce models with lower DockQ and less favorable interface energies. This effect likely arises because longer peptides form larger binding interfaces, which increase conformational flexibility and thereby pose greater sampling challenges. To avoid confounding effects, future analyses should explicitly control for sequence length, for example by computing partial correlations or by stratifying models into length-matched bins.

**Summary.** Together, these results establish DockQ as the most robust single metric for assessing structural quality, validated by its near-perfect correspondence with RMSD and strong agreement with TM-score and GDT_TS. While Rosetta $\Delta G$ provides complementary information about energetic plausibility, its weak correlation with DockQ limits its utility to secondary ranking. Sequence-level properties generally fail to predict structural quality, with the important exception of sequence length, which systematically influences both DockQ and $\Delta G$ and should be considered as a potential confounder in downstream evaluations.

# E  PEPTRI POSITIONING VS PEPGLAD/PEPFLOW/UNIMOMO

Compared to PepGLAD, which relies on sequential decoding with auxiliary geometry losses, and PepFlow, which factorizes sequence and structure into separate flows and applies energy evaluation only after generation, PepTri integrates physics, evolutionary priors, and a sequence–structure coupling term directly into the denoising process within a joint SE(3)-equivariant latent space. While UniMoMo offers a domain-agnostic abstraction that facilitates cross-domain transfer, our approach instead targets peptide-scale interactions within protein pockets, where in-loop physics and mutual information bring clear benefits. On our benchmarks, under consistent relaxation and success criteria, these design choices correlate with lower binding energies and higher success/DockQ scores, though we also observe instances where baseline methods remain competitive on specific metrics.

# F  CASE STUDY

As shown in Fig. 3 and Fig. 4, a key outcome of our case study is that PepTri does more than reproduce training data—it discovers energetically favorable conformations and outperforms baseline models. Because the diffusion process is guided by differentiable physics terms, the model

Table 8: Design dimensions for peptide co-design (✓ = native; ◯ = partial; — = not explicit).

| Dimension | GLAD | Flow | UniMoMo | **PepTri** |
|---|---|---|---|---|
| Joint SE(3) latent | ◯ | — | ◯ | ✓ |
| In-loop guidance | — | — | — | ✓ |
| Evolutionary prior | — | — | — | ✓ |
| Seq–struct coupling (MI) | — | — | — | ✓ |
| Mask-aware inpainting | ◯ | ◯ | ◯ | ✓ |
| Binder–pocket conditioning | ✓ | ✓ | ✓ | ✓ |

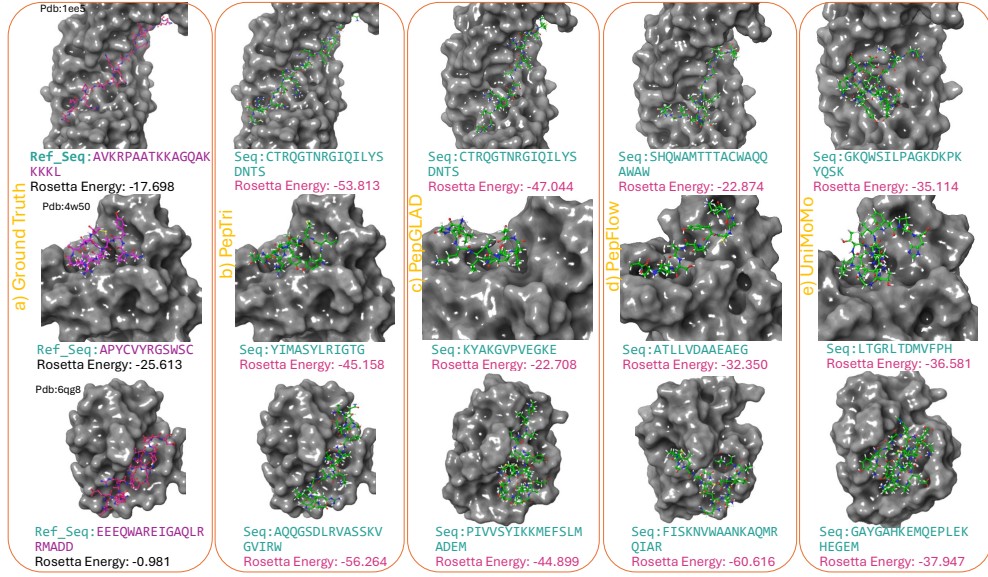

Figure 3: **PepTri case study.** PepTri is able to generate peptide conformations with lower computed physical energies than the experimentally resolved ground-truth structures. This suggests that Pep-Tri's physics-guided denoising does not merely replicate training data but can discover novel, energetically favorable conformations that remain sequence–structure coherent. While energy functions are approximations, this trend indicates that the model integrates physical principles in a meaningful way. In this experiment, we picked the best $\Delta G$ structures for each target.

consistently generates peptide structures with more favorable (lower) computed energies than the corresponding experimentally resolved conformations.

This observation highlights two important points:

- Physics-consistent learning — PepTri's generated conformations are lower in energy validates that our physics-guided denoising is not simply curve-fitting to observed structures. Instead, the model learns to navigate conformational space in a manner consistent with molecular mechanics, uncovering stable regions that even ground-truth datasets may not fully represent.

- Potential functional benefit — From a therapeutic design perspective, lower-energy conformations are often correlated with increased in vivo stability. By biasing generation toward such energetically favorable structures, PepTri improves the likelihood that designed peptides will remain folded and functional under physiological conditions, which is a critical property for drug-like candidates.

Taken together, these results indicate that PepTri embeds physical priors directly into its generative process, yielding designs that are not only statistically plausible but also biophysically robust. This positions PepTri as a practical framework for peptide drug discovery, where stability and functional relevance are equally important.

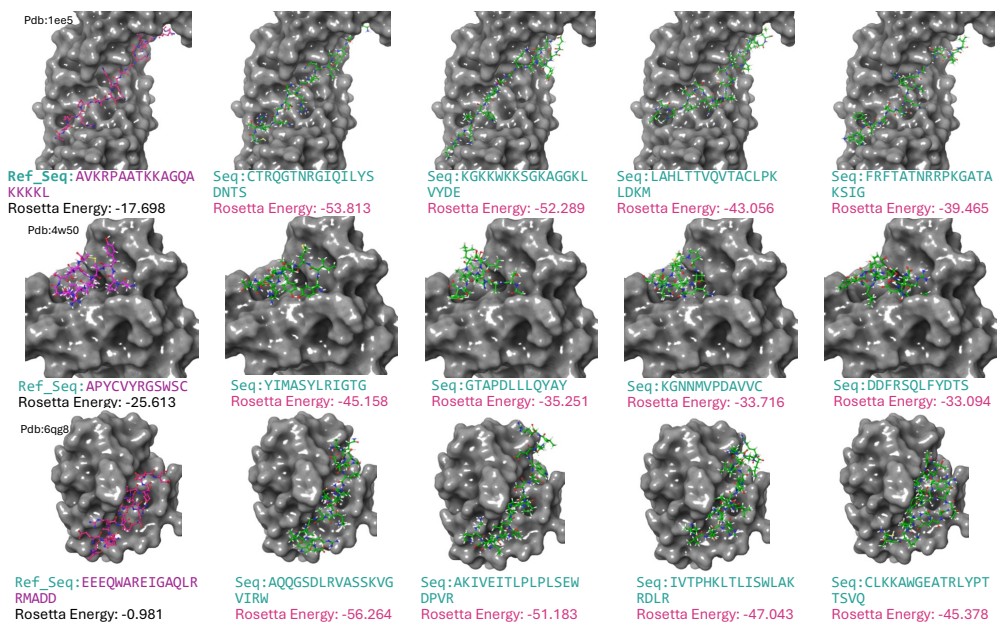

Figure 4: Comparative visualization of peptide–protein binding across ground truth and design methods. Three representative complexes (PDB IDs: 1ee5, 4w50, 6qg8) are shown with peptides bound to protein receptor surfaces (gray). (a) Native ground-truth peptides (pink). (b–e) Designed peptides from PepTri (ours), PepGLAD, PepFlow, and UniMoMo, respectively (green). Sequences and Rosetta binding energies are listed beneath each structure. PepTri consistently produces peptides with tighter binding poses and substantially lower (more favorable) Rosetta energies compared to baselines, demonstrating improved sequence–structure co-design and binding affinity.

# G  RESULTS FOR $\Delta G < 0(\, REU\,)$ THRESHOLD

Table 9: Comparison of success rates (without/with relaxation) in two thresholds of $\Delta G$.

| Dataset | Method | $\Delta G < -5(\, REU\,)\uparrow$ | $\Delta G < 0(\, REU\,)\uparrow$ |
|---|---|---|---|
| PepBench | PepGLAD | 0.294 / *0.790* | 0.459 / _0.862_ |
| | PepFlow | 0.308 / *0.742* | 0.437 / *0.815* |
| | UniMoMo$_{single}$ | _0.339_ / *0.791* | _0.532_ / *0.839* |
| | **PepTri**$_{backbone}$ | 0.254 / *0.784* | 0.397 / *0.849* |
| | **PepTri (Ours)** | **0.404** / *0.825* | **0.583** / ***0.885*** |
| PepBDB | PepGLAD | 0.154 / *0.670* | 0.441 / *0.753* |
| | PepFlow | 0.299 / *0.659* | 0.467 / *0.725* |
| | UniMoMo$_{single}$ | _0.302_ / *0.737* | _0.512_ / *0.782* |
| | **PepTri**$_{backbone}$ | 0.245 / _0.724_ | 0.407 / _0.793_ |
| | **PepTri (Ours)** | **0.313** / *0.742* | **0.530** / ***0.805*** |

In Table 9, PepTri consistently outperforms all baselines across both datasets, with especially strong gains under the stricter binding threshold ($\Delta G < -5$). While success rates are naturally higher at the looser threshold ($\Delta G < -0$), we report results at $\Delta G < -5$ as it reflects more meaningful binding affinity, making improvements more scientifically relevant and demonstrating the robustness of PepTri under stringent conditions.

## H  EFFECT OF RANDOM SEEDS ON PERFORMANCE METRICS

To evaluate the robustness of our protocol, we performed three independent runs using different random seeds. The resulting performance metrics are summarized in Table 10. In general, most of the structural quality indicators, including RMSD(C$\alpha$), TM-score, GDT_TS, Contact_F1, Local RMSD, and DockQ — exhibited very small standard deviations across the three runs. This indicates that the structural models generated are consistent and largely independent of the random seed. Similarly, amino acid recovery (Sliding-AAR), sequence diversity, and structural diversity values remained stable across replicates, suggesting that the design tendencies of the protocol are reproducible.

In contrast, the *mean success rate* showed noticeably higher variability among the three seeds (0.583, 0.559, and 0.563 with a standard deviation of $\approx 0.013$). This difference arises because the success rate is defined as a thresholded metric, where each model is categorized as either "success" or "failure" according to preset quality cutoffs. Small differences in sampling due to the stochastic nature of the Rosetta search process therefore lead to discrete changes in the number of models meeting the success criterion. As a result, the success rate is inherently more sensitive to random seeds than continuous measures such as RMSD or DockQ.

Overall, these results demonstrate that while most structural and energetic metrics are robust to the choice of random seed, the success rate can fluctuate appreciably. This highlights the importance of (i) performing multiple independent replicates, (ii) reporting averages together with standard deviations, and (iii) relying on continuous metrics when possible. Increasing the number of trajectories per seed would further reduce variance and yield a more reliable estimate of the true success probability.

Table 10: Performance variance across 3 independent runs of PepTri with different random seeds.

| Metrics | run1 | run2 | run3 | std |
|---|---|---|---|---|
| Mean success rate ($\Delta G < 0$) | 0.583 | 0.559 | 0.563 | 0.012858201 |
| $\Delta G$ (REU) $\downarrow$ | -19.387 | -19.365 | -19.803 | 0.246773851 |
| DockQ | 0.633 | 0.640 | 0.628 | 0.006027714 |
| Contact_F1 | 0.828 | 0.837 | 0.821 | 0.008020806 |
| Local RMSD (Å) $\downarrow$ | 1.179 | 1.176 | 1.157 | 0.011930353 |
| GDT_TS | 0.747 | 0.746 | 0.738 | 0.004932883 |
| Sequence Diversity | 0.926 | 0.918 | 0.920 | 0.004163332 |
| Sequence Validity | 0.273 | 0.269 | 0.262 | 0.005567764 |
| Struct Diversity | 0.436 | 0.407 | 0.459 | 0.026057628 |
| Consistency | 0.798 | 0.793 | 0.788 | 0.005000000 |
| Sliding-AAR | 0.342 | 0.357 | 0.355 | 0.008144528 |
| TM-score | 0.224 | 0.258 | 0.247 | 0.017349352 |

## I  GUIDANCE SCALE ABLATION STUDY

**Ablation on guidance weights.**  Table 11 shows that up-weighting any single guidance to $\lambda = 1$ degrades overall balance. Our default tri-guidance (PepTri) attains the strongest profile across binding and structure (mean success 0.583, $\Delta G = -19.39$, **DockQ** 0.633, Contact_F1 0.829, GDT_TS 0.747), while preserving diversity (seq 0.926; struct 0.436) and the highest consistency (0.799). Overweighting evolution ($\lambda_{\text{evo}}$=1) slightly raises success (0.587) and improves local RMSD (1.158), but lowers DockQ (0.530) and consistency (0.779); overweighting MI ($\lambda_{\text{MI}}$=1) tightens energy ( $\Delta G = -19.82$) yet reduces DockQ (0.530) and diversity. Overall, extreme single-term guidance ($\lambda$=1) harms docking and coherence, whereas moderate tri-guidance delivers the best joint gains in binding (success, $\Delta G$), docking (DockQ), contacts, and fold (GDT_TS) without sacrificing diversity.

## J  OUT-OF-DISTRIBUTION STRESS TEST

To further assess the robustness and generalization capability of our model, we conducted an out-of-distribution (OOD) stress test. For this experiment, we constructed an OOD benchmark by combin-

Table 11: Guidance-weight ablations where a single component is upweighted to 1.0. Higher is better for all metrics except Local_RMSD_Mean and Sliding-AAR (lower is better).

|  | $\lambda_{\text{phys}}$=1 | $\lambda_{\text{evo}}$=1 | $\lambda_{\text{MI}}$=1 | $\lambda_{\text{Opmm}}$=1 | **PepTri** |
|---|---|---|---|---|---|
| Mean success rate ($\Delta G$<0) | 0.559 | 0.587 | 0.564 | 0.565 | **0.583** |
| $\Delta G$ (REU) $\downarrow$ | -18.122 | -19.363 | **-19.815** | -18.238 | -19.387 |
| DockQ | 0.525 | 0.530 | 0.530 | 0.535 | **0.633** |
| Contact_F1 | 0.801 | **0.836** | 0.829 | 0.819 | 0.829 |
| Local RMSD (Å) $\downarrow$ | 1.211 | **1.158** | 1.197 | 1.201 | 1.179 |
| GDT_TS | 0.745 | 0.745 | 0.746 | 0.736 | **0.747** |
| Sequence Diversity | 0.919 | 0.924 | 0.916 | 0.921 | **0.926** |
| Sequence Validity | 0.256 | **0.278** | 0.274 | 0.276 | 0.273 |
| Struct Diversity | 0.427 | 0.418 | 0.401 | 0.434 | **0.436** |
| Consistency | 0.780 | 0.779 | 0.746 | 0.769 | **0.799** |
| Sliding-AAR | 0.347 | **0.353** | 0.344 | 0.346 | 0.342 |
| TM-score | 0.224 | **0.255** | 0.253 | 0.247 | 0.244 |

ing PepBench and PepBDB and removing duplicated receptors to ensure non-overlapping receptor contexts. From PepBDB, we specifically selected complexes containing peptides longer than 45 amino acids. We additionally removed 15 complexes that overlapped with our training set, resulting in an OOD test set of 114 complexes with substantially longer peptides (46–49 amino acids). The model was retrained using only complexes with peptides shorter than 30 amino acids, allowing us to directly evaluate how well it extrapolates to peptide lengths not observed during training.

Table 12 summarizes the peptide length statistics for the training and OOD test sets. As expected, the test set exhibits a markedly different length distribution, with a mean nearly four times larger than that of the training data and significantly reduced variance. This deliberate mismatch establishes a stringent challenge for sequence and structure generation.

| Statistic | Train set | Test set |
|---|---|---|
| Complex count | 12,823 | 99 |
| Minimum length | 1 | 46 |
| Maximum length | 30 | 49 |
| Mean length | 12.68 | 46.93 |
| Standard deviation | 7.33 | 0.94 |

Table 12: Peptide length statistics for the training and out-of-distribution test sets.

We report generation and structural evaluation metrics for the OOD test set in Table 13 after relaxation. Despite the substantial distribution shift, the model maintains reasonable performance across both sequence-level and structure-level metrics. Notably, the mean success rate ($\Delta G < 0$) remains above 0.6, indicating that a majority of generated peptides still achieve favorable binding energies even when extrapolating to much longer sequences. Structural quality metrics (DockQ, Contact F1, GDT_TS, and Local RMSD) decrease compared to in-distribution performance, as expected under this challenging regime, but remain within a meaningful predictive range. Diversity and consistency metrics remain high, suggesting that the model continues to generate varied yet coherent peptide candidates under OOD conditions.

## K COMPARISON WITH RFDIFFUSION-GENERATED 3D PEPTIDE–BINDER STRUCTURES

PepTri is preferable when accuracy and energetics are the main priorities, whereas RFDiffusion offers major advantages in speed and global topology preservation. The two methods therefore complement one another: PepTri as a precision tool for accurate designs, and RFDiffusion as a high-throughput generator suitable for broad screening.

| Metric | PepTri |
|---|---|
| Mean success rate ($\Delta G < 0$) | 0.624 ($\pm$ 0.208) |
| $\Delta G$ (REU) | -45.878 ($\pm$ 51.937) |
| DockQ | 0.152 ($\pm$ 0.026) |
| Contact F1 | 0.296 ($\pm$ 0.048) |
| Local RMSD | 1.938 ($\pm$ 0.988) |
| GDT_TS | 0.150 ($\pm$ 0.037) |
| Sequence Diversity | 0.999 |
| Sequence Validity rate | 0.128 |
| Struct Diversity | 1.0 |
| Consistency | 0.999 |
| Sliding-AAR | 0.189 |

Table 13: Performance metrics of the model on the out-of-distribution test set of long peptide–receptor complexes.

Table 14: Comparison of 3D peptide–binder structure generation performance metrics between **RFDiffusion** and **PepTri** (40 samples per target without relaxation).

| Metric | RFDiffusion | PepTri |
|---|---|---|
| $\Delta G$ (REU) $\downarrow$ | -16.479 | **-19.387** |
| DockQ | 0.286 | **0.633** |
| Contact F1 | 0.808 | **0.828** |
| Local RMSD (Å)) $\downarrow$ | **0.482** | 1.179 |
| GDT_TS | **0.812** | 0.749 |
| Struct Diversity | 0.401 | 0.436 |
| Running Time (seconds per complex) $\downarrow$ | 60.224 | **5.821** |
| Failed targets (Success rate = 0.0) $\downarrow$ | 33 | 1 |

**Notes on RFDiffusion usage.** Because the RFDiffusion model cannot be retrained for our specific tasks, we directly applied its released checkpoints for inference on our LNR test sets. In practice, the method failed on a substantial fraction of targets, particularly when attempting to generate very short peptides. We found that RFDiffusion has difficulty producing peptides shorter than 8 amino acids, which further restricts its applicability. For this reason, we do not report success rates for RFDiffusion, as such values would be misleading; instead, we focus on metrics that provide a clearer intuition for its performance relative to PepTri.

In Table 14, PepTri provides more accurate and energetically favorable models, while also being significantly faster. RFDiffusion, despite achieving better global fold preservation and lower local distortions, is slower, inference-only, and limited in handling short peptides. In practice, PepTri is better suited for accuracy- and efficiency-driven applications, whereas RFDiffusion may remain useful for generating diverse scaffolds or when global topology is prioritized.

## L  OPENMM FORCE-FIELD SENSITIVITY

Table 15: Comparison of CHARMM36 and Amber14 force fields. Amber14 yields slightly more favorable interface energies and a higher success rate, while DockQ, Contact_F1, and RMSD remain nearly unchanged. This shows that performance is consistent across force fields.

| Metric | CHARMM36 | Amber14 | Amb$-$Cha |
|---|---|---|---|
| Mean success rate ($\Delta G < 0$) | 0.842 | **0.884** | +0.042 |
| $\Delta G$ (REU) $\downarrow$ | -35.019 | **-36.364** | -1.345 |
| DockQ | **0.618** | 0.610 | -0.008 |
| Contact_F1 | **0.845** | 0.837 | -0.008 |
| Local RMSD (Å) $\downarrow$ | **1.047** | 1.101 | +0.054 |

We evaluated the effect of force-field choice by comparing CHARMM36 and Amber14 under identical protocols (Table 15). Switching from CHARMM36 to Amber14 produced more favorable interface energies: the mean success rate increased from 0.842 to 0.884, and the binding energy decreased from $-35.0$ to $-36.4$ REU. At the same time, structure-based measures of interface nativeness were essentially unchanged, with DockQ and Contact_F1 differing by only $-0.008$ each. Local backbone accuracy also remained stable, with a modest RMSD increase of just 0.054 Å. Taken together, these results demonstrate that our conclusions are robust across force fields: Amber14 tends to yield slightly stronger energetic scores, while geometry-based metrics remain consistent, indicating that performance does not hinge on the specific choice of force field. Amber14 yields slightly more favorable binding energies because of differences in parameterization. In particular, its torsional and nonbonded terms were tuned to reproduce peptide–protein interaction energies more closely, and its solvation model tends to give stronger stabilization of side chains at interfaces. As a result, Amber14 generally reports more attractive peptide–protein energies and a higher success rate, even though the underlying structural metrics (DockQ, Contact_F1, RMSD) remain nearly unchanged.

## M    CHOOSING THE MUTUAL-INFORMATION OBJECTIVE

We compare a contrastive objective (InfoNCE; (Oord et al., 2018)) with our *mutual-information* objective (MI/MINE) under identical training and evaluation settings. Table 16 reports aggregate metrics and their differences **MI−InfoNCE**. For RMSD and Sliding-AAR, lower is better; for all other metrics, higher is better. The MI objective directly strengthens global dependence between sequence and structure latents—the factor that drives binding success, energetic stability, and pose quality in peptides. By contrast, InfoNCE's instance-discrimination with finite negatives tends to emphasize local contact patterns and is sensitive to small batches and false negatives. In particular,

$$I(X;Y) \geq \log N - \mathcal{L}_{\text{NCE}}, \tag{28}$$

so the bound tightens only slowly as batch size $N$ increases, while false negatives (biophysically related pairs) corrupt the gradients. In peptide space, "negatives" are often related (shared motifs, fold families, local geometry), causing InfoNCE to underestimate $I(X;Y)$ and to over-penalize useful similarities. Interestingly, PepTri trained with InfoNCE yields a slightly higher fraction of valid sequences. Overall, for peptide co-design—short, flexible chains with tight sequence–structure coupling—the MI objective provides a more suitable inductive bias. By reinforcing cross-modal alignment, it consistently improves binding success, energetic favorability, and pose accuracy.

Table 16: Aggregate comparison of InfoNCE vs. MI (MINE) for cross-domain experiment.

| Metric | InfoNCE | MI (MINE) | MI−InfoNCE |
|---|---|---|---|
| Mean success rate ($\Delta G < 0$) | 0.5634 | **0.5831** | +0.0197 |
| $\Delta G$ (REU) ↓ | -18.8915 | **-19.387** | **-0.4958** |
| DockQ | 0.6171 | **0.6331** | **+0.0161** |
| Contact_F1 | **0.8567** | 0.8287 | -0.0280 |
| Local RMSD (Å) ↓ | 1.1841 | **1.1798** | **-0.0043** |
| GDT_TS | 0.7465 | **0.7474** | +0.0009 |
| Sequence Diversity | 0.9199 | **0.9261** | **+0.0062** |
| Sequence Validity | **0.2891** | 0.2734 | -0.0157 |
| Struct Diversity | **0.4651** | 0.4363 | -0.0288 |
| Consistency | 0.7794 | **0.7988** | **+0.0194** |
| Sliding-AAR | **0.3483** | 0.3428 | -0.0055 |
| TM-score | 0.2430 | **0.2446** | +0.0015 |

**Handling Variance in MINE**    It is well established that the Mutual Information Neural Estimator (MINE) exhibits substantial variance during optimization, frequently resulting in unstable convergence and performance deterioration in later stages of training. To address these challenges, we employed two complementary stabilization strategies:

- Learning rate decay – progressively reducing the step size helped dampen oscillations and avoid divergence in later iterations.

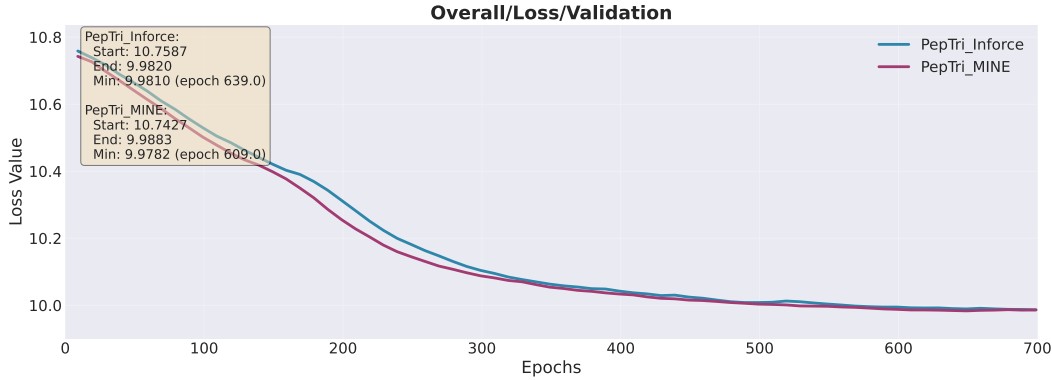

Figure 5: The smoothed validation comparison curve on the Cross-domain experiment between InfoNCE and MINE on validation epochs.

- Exponential Moving Average (EMA) – smoothing the parameter updates improved robustness against variance spikes.

Together, as shown in Figure 6 and Figure 5, these techniques not only stabilized training but also accelerated convergence, allowing us to reach the early minima observed in our curves and sustain better performance across sequence and structure losses.

## N  REPRODUCIBILITY

We trained the model using PyTorch 2.6 with CUDA 12.4 on 8 × A100 GPUs.

The code for data processing, model definition, training, testing, evaluation, and trained weights will be made available at: `https://github.com/aigensciences/PepTri`.

To compute the interface energy of generated peptides, we used PyRosetta. Please refer to the official installation instructions: `https://www.pyrosetta.org/`

For structural quality assessment, we employed DockQ: `https://github.com/wallnerlab/DockQ`

### N.1  HYPERPARAMETERS

**Encoder–Decoder Hyperparameters.**  We trained the encoder–decoder with the AdamW optimizer (learning rate $1.0 \times 10^{-4}$) and applied a ReduceLROnPlateau scheduler (factor $0.8$, patience $5$ epochs, mode `min`, evaluated at validation epochs, minimum learning rate $5.0 \times 10^{-6}$, max epochs $100$).

**Model configuration:** embedding size 128, hidden size 128, latent size 8, latent channels 1, number of layers 3, channels 14. **Regularization:** hierarchical KL weight 0.3, latent KL weight 0.5. **Loss weighting:** coordinate loss ratio 0.5, with subcomponents:

- $\mathcal{L}_{\text{X}}$: 1.0,   $\mathcal{L}_{\text{C}\alpha\text{-X}}$: 1.0

- $\mathcal{L}_{\text{bb-bond}}$: 1.0,   $\mathcal{L}_{\text{sc-bond}}$: 1.0

- $\mathcal{L}_{\text{bb-dihedral}}$: 0.0,   $\mathcal{L}_{\text{sc-chi}}$: 0.5

**Additional settings:** relative position encoding disabled; anchor at $C\alpha$ enabled; masking ratio 0.25; additive noise scale 0.1. **Stability controls:** spectral normalization disabled; 1 residual block used; gradient clipping at 1.0; exponential moving average enabled with decay 0.999.

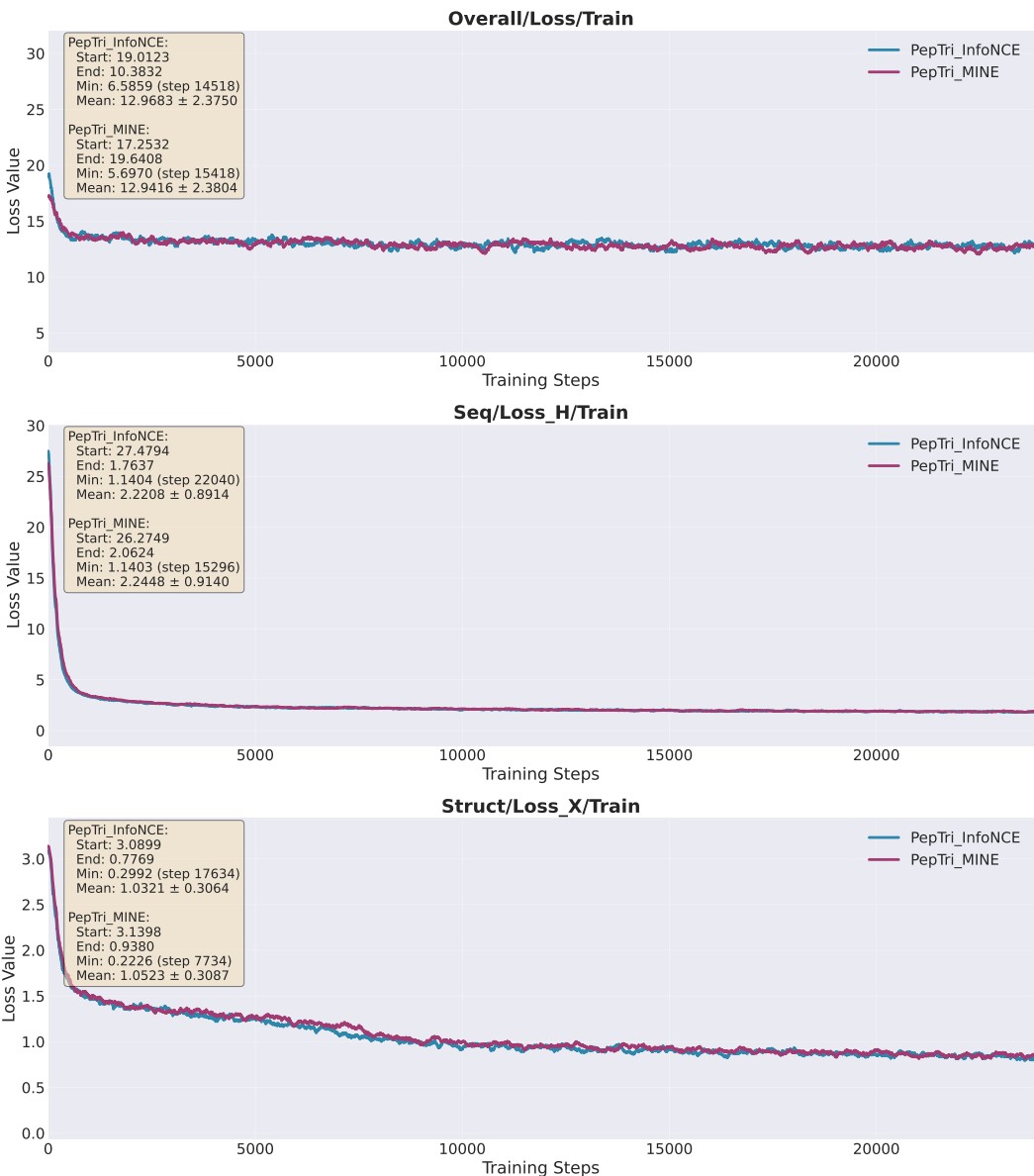

Figure 6: The smoothed validation comparison curve on the Cross-domain experiment between InfoNCE and MINE on training steps.

## N.2 DIFFUSION HYPERPARAMETERS

**Optimization.** We trained using AdamW (learning rate $1.0 \times 10^{-4}$) with a `ReduceLROnPlateau` scheduler (factor $0.6$, patience $3$ epochs, mode = `min`, monitored at validation epochs, minimum learning rate $5.0 \times 10^{-6}$).

**Model configuration.** We used the `LDMPepDesign` backbone with hidden size 128, 3 layers, 100 denoising steps, 32 RBF kernels (cutoff 3.0), and distance-based RBF encoding (32 channels, cutoff 7.0). Both sequence and positional transformations were modeled via diffusion.

**Physics-guided loss.** Physics loss was enabled with weight $0.15$. The physics configuration included:

- **Bond constraints:** bond length weight 1.2; ideal C$\alpha$–C$\alpha$ distance 3.8Å with tolerance 0.25Å.
- **Angles:** bond angle weight 0.8; ideal 109.5° with tolerance 18°.
- **Torsions:** torsion weight 0.4; Ramachandran prior weight 0.6.
- **Non-bonded:** van der Waals weight 0.6 ($\sigma = 3.4$Å, $\epsilon = 0.12$ kcal/mol); electrostatics weight 0.5 with dielectric constant 78.0 and cutoff 10.0Å.
- **Hydrogen bonding:** weight 0.7, distance cutoff 3.4Å, angle cutoff 25°.
- **Secondary structure:** total weight 0.4 (helix 0.3, sheet 0.3).
- **Sterics:** clash prevention weight 1.8, minimum distance 2.1Å, soft clash threshold 2.7Å.
- **Solvent/hydrophobicity:** SASA weight 0.3, hydrophobic weight 0.4.
- **Diffusion regularizers:** smoothness weight 0.25, temporal consistency weight 0.3.

**Evolutionary guidance.** Evolutionary priors were incorporated with fitness weight 0.05 and bias weight 0.02, with conservation bias and coevolution enabled.

**Information-theoretic regularization.** Mutual information (MI) guidance was applied with weight 0.1, and additional coupling to physics with weight 0.05.

**OpenMM guidance.** Force-field–based corrections were included with loss weight 0.01, guidance scale 0.001, and force-guidance scale 0.0001.

**SE(3)-aware diffusion.** We enabled SE(3)-aware latent diffusion with enhanced latent dimension 17 (8 global + 9 local). Integration with base latent features was via concatenation. SE(3) regularization weights were: rotation 0.1, translation 0.1, geometric consistency 0.1, scale invariance 0.05, local frame consistency 0.08, invariant feature weight 0.05.

## O   DATA ANALYSIS

Short peptides—typically fewer than 30 amino acids—offer notable advantages in efficiency, reproducibility, and overall experimental success. In this study, we focus on short peptides due to their ease of synthesis, higher yields, and lower error rates compared to longer sequences. Their smaller size also contributes to superior solubility and compatibility with high-purity purification techniques such as High-Performance Liquid Chromatography (HPLC). Moreover, in mass spectrometry workflows, short peptides exhibit more reliable fragmentation and ionization, leading to improved detection accuracy. These attributes make short peptides particularly well-suited for a variety of applications, including MHC binding assays, epitope mapping, and peptide-based screening.

While long peptides may be necessary for certain complex immunological applications, their production and handling present significant technical challenges. Notably, a review of FDA-approved peptide drugs shows that the vast majority fall within the short peptide range: drugs like leuprorelin (10 amino acids), ziconotide (25 aa), and difelikefalin (9 aa) are examples of clinically successful therapies with short sequences. Comprehensive databases such as THPdb2 and analyses published in peer-reviewed literature confirm that most approved peptide drugs fall well below the 30-amino-acid threshold[2]. Therefore, prioritizing short peptide generation aligns with both technical feasibility and biological relevance.

Following the suggestion from PepGLAD (Kong et al., 2024), we used MMseqs2 to cluster the entire dataset, enabling us to split it into training, validation, and test sets. As shown in Figures 7 and 8, there is no duplication between the targets in the training and test sets. More specifically, to assess cross-target generalization, we adopt the large non-redundant (LNR) dataset introduced by Tsaban et al. (Tsaban et al., 2022) as the test set. The LNR, curated by domain experts, originally comprised 96 protein–peptide complexes; after excluding entries with non-canonical amino acids, 93 complexes remained. These complexes were then clustered together with PDB data by receptor,

---

[2]Data source: `https://peptidesguide.com`, `https://www.sciencedirect.com/science/article/pii/S1359644624001727`

using a sequence identity threshold of 40%. In PepBDB experiment, we applied MMseqs2 clustering and then randomly partitioned the data into training, validation, and test sets based on the resulting clusters. To construct the test set, one protein–peptide complex was randomly chosen from each cluster, ensuring non-redundancy across samples.

Table 17: Peptide statistics of PepBench by train/val/test split

| Split | Count | Mean | Median | Max | Unique Proteins | Unique Peptides |
|---|---|---|---|---|---|---|
| Train | 4157 | 11.17 | 10.0 | 25 | 2783 | 3504 |
| Validation | 114 | 13.26 | 13.0 | 25 | 94 | 111 |
| Test (LNR) | 93 | 10.15 | 10.0 | 26 | 93 | 93 |

Table 18: Peptide statistics of PepBDB by train/val/test split

| Split | Count | Mean | Median | Max | Unique Proteins | Unique Peptides |
|---|---|---|---|---|---|---|
| Train | 7014 | 13.13 | 11 | 30 | 4394 | 4349 |
| Validation | 323 | 13.61 | 13 | 30 | 226 | 248 |
| Test | 142 | 12.58 | 10 | 30 | 142 | 142 |

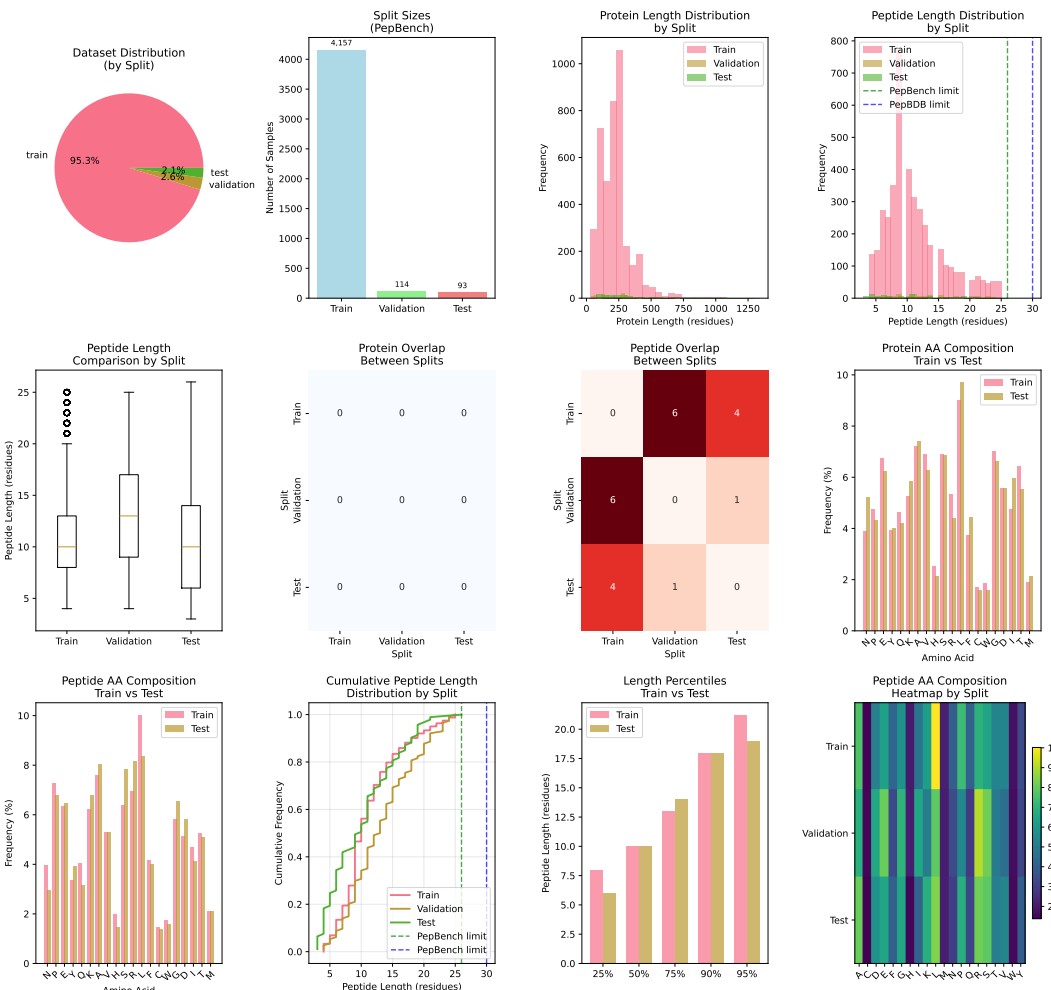

Figure 7: The comprehensive analysis examines the PepBench dataset (train from PepBench + validation from PepBench + test from LNR).

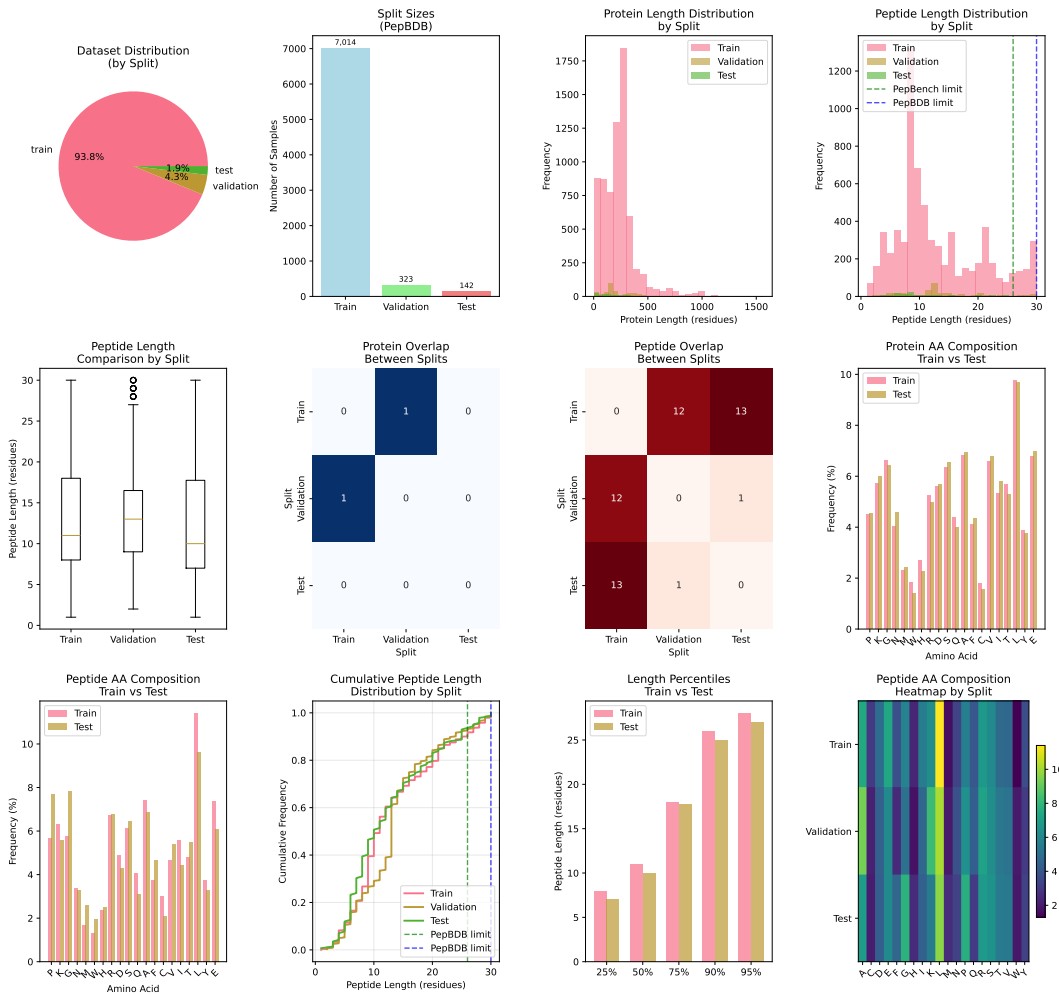

Figure 8: The comprehensive analysis examines the PepBDB dataset .

## O.1 CLUSTERING TARGET ANALYSIS USING MMSEQS2

We performed clustering using `MMseqs2` based on three types of feature representations: composition-based, physicochemical-based, and a combined representation.

The composition-based representation achieved consistent separation across both datasets, indicating stable performance regardless of peptide length or source. In contrast, the physicochemical-based representation showed greater separation in the *PepBench* dataset, likely due to its shorter peptide sequences, which emphasize physicochemical diversity. The combined representation yielded the best overall clustering performance, effectively capturing both compositional and physicochemical characteristics.

The average separation scores for each dataset further support these findings. For *PepBDB*, the clustering achieved an average separation of 15.2 as shown in Figure 7, indicating good clustering quality. In Figure 8 For *PepBench*, the average separation was 22.8, reflecting excellent clustering quality.

As shown in Figure 9, these results confirm that both datasets exhibit well-separated train/test splits, demonstrating the effectiveness of our feature engineering and clustering strategy.

These results indicate that both datasets exhibit well-separated train/test splits, validating the effectiveness of our feature extraction and clustering approach.

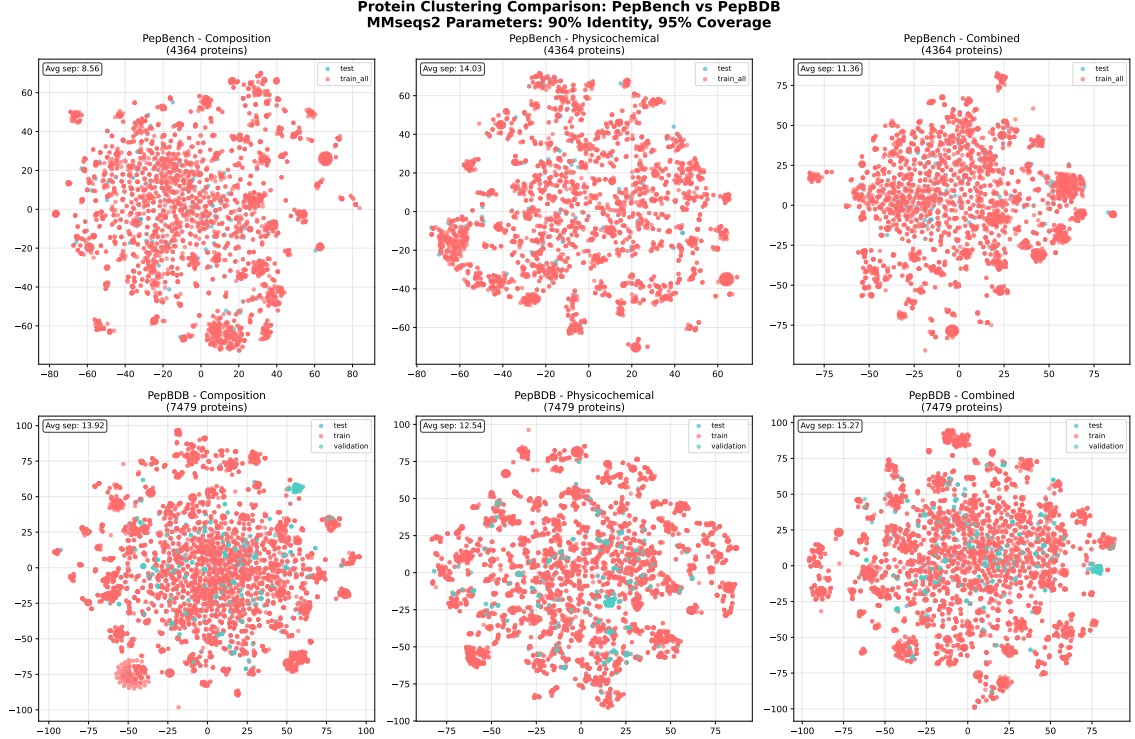

Figure 9: The Pepbench data experiment (up) and PepBDB data experiment (down) with MMseqs2 clustering for targets. Composition clustering: Catches sequences with similar AA recipes. Physicochemical clustering: Catches functionally similar sequences. Combined clustering: Catches both types of similarity. Our average separation scores ($>10.0$) across multiple feature representations (composition, physicochemical, combined) show that our dataset splits are well-separated and minimize the risk of data leakage.

## P   TRI-GUIDANCE LATENT DIFFUSION DETAILS

### P.1   VAE WITH SE(3)-EQUIVARIANT GRAPH ENCODING

Our autoencoder constructs a latent representation that functions as the interface for the diffusion process, ensuring preservation of geometric consistency via graph-based message passing. The PepTri encoder employs SE(3)-equivariant graph neural networks, which enforce rotational and translational symmetries while encoding both local residue-level interactions and global structural dependencies. This design enables accurate modeling of protein conformations within a symmetry-aware latent space, thereby facilitating downstream generative diffusion.

**Graph representation.**   We first represent peptides as molecular graphs where:

- **Nodes**: Each residue $i \in \{0, \ldots, L-1\}$ with features $h_i \in \mathbb{R}^d$ encoding amino acid type and positional information

- **Edges**: Two types of connections:
    - *Sequential edges*: Connect adjacent residues $(i, i+1)$ along the backbone
    - *Spatial edges*: Connect residues within cutoff distance $r_c = 10\text{Å}$

- **Edge features**: Distance-based radial basis functions (RBF) encoding 3D geometry

**Inputs and outputs.** Given sequence $S \in \{0, \dots, 19\}^L$ and coordinates $X \in \mathbb{R}^{L \times C \times 3}$ with mask $M \in \{0, 1\}^L$, we construct:

$$\mathcal{G} = (\mathcal{V}, \mathcal{E}_{\text{seq}} \cup \mathcal{E}_{\text{spatial}}), \tag{29}$$

$$h_i^{(0)} = \text{Embed}(S_i) + \text{PosEmbed}(i), \tag{30}$$

$$e_{ij} = \text{RBF}(\|X_i^{\text{C}\alpha} - X_j^{\text{C}\alpha}\|_2), \tag{31}$$

where $\text{C}\alpha$ denotes $\alpha$-carbon coordinates used for graph construction.

**SE(3)-equivariant message passing with enhancements.** We employ an *enhanced* Adaptive Multi-channel EGNN (AMEGNN) that performs $K$ layers of equivariant message passing. The enhancement includes explicit SE(3)-invariant geometric features beyond basic distances:

**Core equivariant operations** (preserved from baseline):

$$m_{ij}^{(\ell)} = \psi_m^{(\ell)}(h_i^{(\ell-1)}, h_j^{(\ell-1)}, d_{ij}^{(\ell-1)}, e_{ij}), \tag{32}$$

$$h_i^{(\ell)} = h_i^{(\ell-1)} + \psi_h^{(\ell)}\left(h_i^{(\ell-1)}, \sum_{j \in \mathcal{N}(i)} m_{ij}^{(\ell)}\right), \tag{33}$$

$$x_i^{(\ell)} = x_i^{(\ell-1)} + \sum_{j \in \mathcal{N}(i)} (x_i^{(\ell-1)} - x_j^{(\ell-1)}) \cdot \psi_x^{(\ell)}(m_{ij}^{(\ell)}), \tag{34}$$

where $d_{ij} = \|x_i - x_j\|$ and $\psi_x : \mathbb{R}^d \to \mathbb{R}$ ensures scalar outputs.

**SE(3)-invariant enhancements** (new in enhanced version):

- **Bond angles**: For each triplet $(i, j, k)$:

$$\zeta_{ijk} = \arccos\left(\frac{(x_i - x_j) \cdot (x_k - x_j)}{\|x_i - x_j\|\|x_k - x_j\|}\right) \tag{35}$$

- **Dihedral angles**: Four-body torsion angles $\tau_{ijkl}$ along the backbone
- **Global shape descriptors**:

$$R_g = \sqrt{\frac{1}{N}\sum_i \|x_i - \bar{x}\|^2} \quad \text{(radius of gyration)} \tag{36}$$

$$\lambda_1, \lambda_2, \lambda_3 = \text{eigenvalues}(\mathbf{S}) \quad \text{(principal moments)} \tag{37}$$

$$\Delta = \lambda_1 - \frac{1}{2}(\lambda_2 + \lambda_3) \quad \text{(asphericity)} \tag{38}$$

where $\mathbf{S}$ is the gyration tensor. These are all SE(3)-invariant.

The enhanced message function incorporates these invariants:

$$m_{ij}^{(\ell)} = \psi_m^{(\ell)}(h_i^{(\ell-1)}, h_j^{(\ell-1)}, d_{ij}^{(\ell-1)}, \zeta_{ijk}^{(\ell-1)}, \tau_{ijkl}^{(\ell-1)}, R_g, \Delta, e_{ij}) \tag{39}$$

**SE(3) guarantees.** The architecture maintains strict SE(3)-equivariance through:

1. **Invariant node features**: All $h_i$ updates use only SE(3)-invariant inputs (never raw coordinates)
2. **Equivariant coordinate updates**: Position changes use relative vectors $(x_i - x_j)$ scaled by invariant coefficients
3. **Invariant aggregation**: Summation over neighbors preserves equivariance
4. **No global reference frame**: All computations are relative or invariant

**Theorem**: For any rotation $\mathbf{R} \in \text{SO}(3)$ and translation $\mathbf{t} \in \mathbb{R}^3$:

$$h_i^{(\ell)}(\mathbf{R}X + \mathbf{t}) = h_i^{(\ell)}(X) \quad \text{(invariance)} \tag{40}$$

$$x_i^{(\ell)}(\mathbf{R}X + \mathbf{t}) = \mathbf{R}x_i^{(\ell)}(X) + \mathbf{t} \quad \text{(equivariance)} \tag{41}$$

**Multi-channel processing.** To handle all atoms (not just C$\alpha$), we extend to multi-channel coordinates:

- Each node processes $C$ atom channels in parallel

- Channel-specific attention weights $w_c \in [0, 1]$ indicate atom presence

- All operations maintain per-channel SE(3) properties

**Latent encoding.** After $K$ message passing layers, the encoder outputs disentangled latents:

$$z_h = \text{VAE}_h(h^{(K)}[M]) \in \mathbb{R}^{|M| \times d_h} \quad \text{(sequence features, invariant)}, \tag{42}$$

$$z_x = \text{VAE}_x(x^{(K)}[M]) \in \mathbb{R}^{|M| \times n_{\text{lat}} \times 3} \quad \text{(structure anchors, equivariant)}, \tag{43}$$

where $\text{VAE}_h$ and $\text{VAE}_x$ include reparameterization for variational learning,

**Graph-aware training objective.** The VAE is trained with geometric consistency:

$$\mathcal{L}_{\text{VAE}} = \mathcal{L}_{\text{recon}} + \beta \, \mathcal{L}_{\text{KL}} + \lambda_{\text{geom}} \mathcal{L}_{\text{geom}} + \lambda_{\text{graph}} \mathcal{L}_{\text{graph}}, \tag{44}$$

where all loss terms are SE(3)-invariant:

$$\mathcal{L}_{\text{recon}} = \text{CE}(S, \hat{S}) + \|X - \hat{X}\|_2^2, \tag{45}$$

$$\mathcal{L}_{\text{geom}} = \|D(\hat{X}) - D(X)\|_F^2, \tag{46}$$

$$\mathcal{L}_{\text{graph}} = \sum_{(i,j) \in \mathcal{E}} \left( \|x_i - x_j\|_2 - \|\hat{x}_i - \hat{x}_j\|_2 \right)^2. \tag{47}$$

**Advantages of SE(3)-enhanced graph representation.**

- **Theoretical guarantees**: Provable equivariance under rotations and translations

- **Richer features**: Explicit geometric descriptors complement learned representations

- **Physical interpretability**: Bond angles, dihedrals have direct structural meaning

- **Improved generalization**: SE(3) symmetry reduces sample complexity

### P.2 GAUSSIAN REVERSE TRANSITION

We follow the DDPM formulation with a cosine variance schedule and apply explicit physics guidance during sampling on the structural latent only (Huberman-Spiegelglas et al., 2024; Meng et al., 2023). Let $p \in \{H, X\}$ denote, respectively, the sequence and structure latents at step $t \in \{1, \dots, T\}$. We use a cosine schedule with small offset $s = 0.01$:

$$f_t = \cos^2\left( \frac{\pi}{2} \frac{(t/T)+s}{1+s} \right), \qquad \bar{\alpha}_t = \frac{f_t}{f_0}, \qquad \beta_t = 1 - \frac{\bar{\alpha}_t}{\bar{\alpha}_{t-1}}, \qquad \alpha_t = 1 - \beta_t.$$

**Forward diffusion (noising).** For either latent $p \in \{H, X\}$, sample $\epsilon \sim \mathcal{N}(0, \mathbf{I})$ and form

$$q(p_t \mid p_0) = \mathcal{N}\left( \sqrt{\bar{\alpha}_t} \, p_0, \, (1 - \bar{\alpha}_t) \, \mathbf{I} \right).$$

Equivalently, $p_t = \sqrt{\bar{\alpha}_t} \, p_0 + \sqrt{1 - \bar{\alpha}_t} \, \epsilon$. In practice we apply noising only to positions to be generated using a binary mask $M$ ($M = 1$ for generated positions),

$$p_t = M \odot \left( \sqrt{\bar{\alpha}_t} \, p_0 + \sqrt{1 - \bar{\alpha}_t} \, \epsilon \right) + (1 - M) \odot p_0,$$

and retain $\epsilon$ as the supervision target for the denoiser.

**Noise-prediction training loss.** Let $\hat{\epsilon}_\theta(H_t, X_t, t) = (\hat{\epsilon}_H, \hat{\epsilon}_X)$ be the predicted noises. We use masked mean-squared error losses:

$$\mathcal{L}_X = \frac{\left\|(\hat{\epsilon}_X - \epsilon_X) \odot M_X\right\|_2^2}{\|M_X\|_1},$$

$$\mathcal{L}_H = \frac{\left\|(\hat{\epsilon}_H - \epsilon_H) \odot M_H\right\|_2^2}{\|M_H\|_1},$$

where the mask restricts the reduction to generated entries. These losses are combined with evolutionary, mutual-information, and physics terms during training.

At sampling time, given a noise predictor $\hat{\epsilon}_\theta$ and letting

$$c_0(\alpha_t) = \frac{1}{\sqrt{\alpha_t}}, \qquad c_1(\alpha_t, \bar{\alpha}_t) = \frac{1 - \alpha_t}{\sqrt{1 - \bar{\alpha}_t}},$$

the Gaussian reverse transition without guidance is

$$p_{t-1} = c_0\Big(p_t - c_1\,\hat{\epsilon}_\theta(p_t, t)\Big) + \sigma_t\, z, \qquad z \sim \mathcal{N}(0, \mathbf{I}), \tag{48}$$

with variance

$$\sigma_t^2 = \frac{1 - \bar{\alpha}_{t-1}}{1 - \bar{\alpha}_t}\, \beta_t \quad \text{(cosine schedule)}.$$

**Physics guidance on structure.** During sampling we incorporate differentiable molecular mechanics as an energy $E_{\text{phys}}(X_t)$. In code, the predicted noise for $X$ is modified as

$$\hat{\epsilon}'_X(H_t, X_t, t) = \hat{\epsilon}_X(H_t, X_t, t) + \sqrt{1 - \bar{\alpha}_t}\, \nabla_{X_t} E_{\text{phys}}(X_t), \tag{49}$$

which corresponds to using guidance $= -\nabla_{X_t} E_{\text{phys}}$ inside the denoiser and the update rule equation 48. The structural reverse step is therefore

$$X_{t-1} = c_0\Big(X_t - c_1\,\hat{\epsilon}'_X(H_t, X_t, t)\Big) + \sigma_t z.$$

An optional weight $\lambda_{\text{phys}}$ can scale the guidance term (set to 1 in our implementation), and with the OpenMM (if enable).

Thus, all three signals (physics, evolutionary, mutual information) shape the denoiser $\hat{\epsilon}_\theta$ through the training loss, while the physics contributes an explicit gradient term during sampling for the structure latent.

## P.3   LATENT INPAINTING DIFFUSION

**Strengths.** Our latent inpainting diffusion brings several practical advantages:

- **Targeted controllability**: Noise and supervision are *masked* (equation 21), so only designed residues are modified while structural context is preserved via clamping (equation 27). This yields precise, locality-aware edits.

- **SE(3) consistency**: The explicit guidance uses energies over *internal* geometry (distances/angles), ensuring the correction in equation 25 is invariant to global rotations/translations and remains compatible with the equivariant backbone.

- **Physics-aware generation**: The composite energy penalizes $C\alpha$ bond-length/angle violations, steric clashes, and poor non-bonded interactions, leading to fewer post-relaxation artifacts and improved local geometry at sample time.

- **Stable guidance design**: Evolutionary/MI terms act only during training (shape $\varepsilon_{\theta\star}$), while sampling applies *only* physics gradients. This separation avoids double-counting objectives and keeps inference stable.

---

**Algorithm 1** Latent Inpainting Diffusion

---

1: **procedure** LIDPG($S, \mathbf{z}_0, M, T, \varepsilon_{\theta^\star}, \{\alpha_t\}_{t=1}^T$)
2:      Broadcast $M_X, M_H$ using Eq. equation 21
3:      Initialize noise: $\mathbf{z}_T \leftarrow M \odot \xi + (1 - M) \odot \mathbf{z}_0, \;\; \xi \sim \mathcal{N}(0, I)$
4:      **for** $t = T, T - 1, \dots, 1$ **do**
5:          Predict noise: $\hat{\varepsilon}_t \leftarrow \varepsilon_{\theta^\star}(\mathbf{z}_t, t)$
6:          Decode structure: $\widehat{X}_t \leftarrow \text{decode}(\mathbf{z}_{X,t})$
7:          Physics guidance: $G_t^{\text{phys}} \leftarrow -\lambda_{\text{phys}}(t) \, \nabla_{\mathbf{z}_{X,t}} E_{\text{phys}}(\widehat{X}_t, S; M)$
8:          Guided noise:

$$\tilde{\varepsilon}_{H,t} \leftarrow \hat{\varepsilon}_{H,t}, \quad \tilde{\varepsilon}_{X,t} \leftarrow \hat{\varepsilon}_{X,t} - \sqrt{1 - \bar{\alpha}_t} \, G_t^{\text{phys}}$$

9:          Sample masked noise $\xi_t \sim \mathcal{N}(0, I)$, optionally $\xi_t \leftarrow M \odot \xi_t$
10:        Reverse update $\mathbf{z}_{t-1}$ using Eq. equation 26
11:        Clamp context: $\mathbf{z}_{t-1} \leftarrow M \odot \mathbf{z}_{t-1} + (1 - M) \odot \mathbf{z}_0$
12:     **return** $(H_0, X_0) \leftarrow \text{decode}(\mathbf{z}_{H,0}, \mathbf{z}_{X,0})$

---

- **Noise and context control**: Optional masking of the per-step noise $\boldsymbol{\xi}$ confines stochasticity to redesigned sites, while context clamping guarantees exact preservation of the unmasked scaffold over the whole trajectory.

- **Late-stage refinement**: A simple annealing of $\lambda_{\text{phys}}(t)$ emphasizes physical validity near convergence without over-constraining early exploration.

- **Efficiency**: Guidance is computed on $C_\alpha$ (lightweight, numerically stable), with optional OpenMM as a gated add-on. Gradients are obtained by differentiating through a partial decode, reducing overhead.

- **Compatibility**: The reverse update (equation 26) supports both stochastic DDPM sampling ($\sigma_t > 0$) and deterministic DDIM-style sampling ($\sigma_t = 0$) without changing the architecture.

- **Interpretability and compositionality**: The energy is a transparent sum of physically meaningful terms with tunable weights, allowing principled trade-offs and easy integration of additional constraints (motifs, distance restraints).

**Limitations.**

- **Heuristic energies**: Electrostatics (distance-dependent dielectric), LJ radii, and secondary-structure proxies are coarse; weights $w_j$ require tuning and can interact non-linearly.

- **Training–sampling**: Only physics is applied at sampling; evolutionary and MI act implicitly via $\theta^\star$, which can drift under strong guidance (Ingraham et al., 2023).

- **Hyperparameter sensitivity**: Performance depends on $\lambda_{\text{phys}}(t)$, schedule, masking of $\boldsymbol{\xi}$, and decode quality; poor settings cause over-smoothing or instability (Ho et al., 2020; Dhariwal & Nichol, 2021).

- **Local minima/exploration**: Energy guidance can trap samples in local basins; masking $\boldsymbol{\xi}$ or using deterministic sampling reduces diversity (Jumper et al., 2021).

- **Scalability**: Long peptides increase pair/triplet costs (non-bonded terms), stressing memory/time without sparse approximations (Shaw et al., 2021).

P.4    PHYSICS-INFORMED STRUCTURAL GUIDANCE

**Motivation and scope.** Physics guidance addresses specific structural defects observed in purely data-driven generation:

- **Local geometry violations**: Incorrect bond lengths (e.g., C-N distances $> 1.5$Å)

- **Angular distortions**: Unrealistic bond angles (e.g., N-C$\alpha$-C deviating from $109°$)

- **Steric clashes**: Atom overlaps violating van der Waals radii
- **Unphysical conformations**: Structures with high internal strain energy
- **OpenMM**: Plays a late-stage and selective role, closely aligned with the reviewer's suggestion to "apply it only near the end of the trajectory.

We represent peptides in an all-atom format, $X \in \mathbb{R}^{L \times 14 \times 3}$ where $L$ is the sequence length. This fixed-size representation allocates channels 0-3 for backbone atoms (N, C$\alpha$, C, O) and channels 4-13 for sidechain atoms following standard PDB atom ordering for each amino acid type. Residues with fewer than 14 heavy atoms have unused channels padded with the mean position of that residue's existing atoms. An atom validity mask $M_{\text{atom}} \in \{0,1\}^{L \times 14}$ tracks real atoms (1) versus padding (0), while a separate design mask $M \in \{0,1\}^L$ indicates which residues to optimize. This uniform tensor representation enables efficient batched processing across all 20 standard amino acids—from glycine (4 atoms, channels 4-13 padded) to tryptophan (14 atoms, fully populated)—while preserving complete atomic detail for physics-based calculations.

**Comprehensive physics energy function.** We define a composite energy function that captures multiple aspects of molecular physics:

$$E_{\text{phys}}(\widehat{X}, S; M) = \sum_{i=1}^{7} \lambda_i E_i(\widehat{X}, S; M), \tag{50}$$

where each term addresses specific physical constraints.

**1. Bond length constraints.** Maintains ideal covalent bond distances:

$$E_{\text{bond}} = \sum_{(i,j) \in \mathcal{B}} k_b \left( \|x_i - x_j\| - d_{ij}^0 \right)^2 \tag{51}$$

where $\mathcal{B}$ is the set of covalent bonds, $d_{ij}^0$ is the ideal bond length for atom types $(i,j)$, and $k_b = 100$ kcal/mol / Å$^2$.

Typical values:

- N-C$\alpha$: $d^0 = 1.46$ Å
- C$\alpha$-C: $d^0 = 1.53$ Å
- C-N: $d^0 = 1.33$ Å(peptide bond)
- C=O: $d^0 = 1.23$ Å(carbonyl)

**2. Bond angle constraints.** Enforces ideal bond angles for triplets of bonded atoms:

$$E_{\text{angle}} = \sum_{(i,j,k) \in \mathcal{A}} k_a \left( \zeta_{ijk} - \zeta_{ijk}^0 \right)^2 \tag{52}$$

where $\zeta_{ijk} = \arccos\left( \frac{(x_i - x_j) \cdot (x_k - x_j)}{\|x_i - x_j\| \|x_k - x_j\|} \right)$ and $k_a = 50$ kcal/mol/rad$^2$.

Key angles:

- N-C$\alpha$-C: $\zeta^0 = 111.0$ (tetrahedral)
- C$\alpha$-C-N: $\zeta^0 = 116.2$
- C-N-C$\alpha$: $\zeta^0 = 121.7$ (peptide plane)

**3. van der Waals interactions.** Models non-bonded atomic interactions using the Lennard-Jones potential:

$$E_{\text{vdW}} = \sum_{\substack{i,j \\ |i-j|>2}} 4\epsilon_{ij} \left[ \left( \frac{\sigma_{ij}}{r_{ij}} \right)^{12} - \left( \frac{\sigma_{ij}}{r_{ij}} \right)^6 \right] \tag{53}$$

where $r_{ij} = \|x_i - x_j\|$, $\epsilon_{ij} = \sqrt{\epsilon_i \epsilon_j}$ (well depth), and $\sigma_{ij} = (\sigma_i + \sigma_j)/2$ (collision diameter).

Parameters by atom type:

| Atom | $\sigma$ (Å) | $\epsilon$ (kcal/mol) |
|------|-----------|------------------------|
| C | 1.70 | 0.110 |
| N | 1.55 | 0.170 |
| O | 1.52 | 0.210 |
| S | 1.80 | 0.250 |

**4. Electrostatic interactions.**  Coulombic interactions between charged residues:

$$E_{\text{elec}} = \sum_{\substack{i,j \\ |i-j|>4}} \frac{k_e q_i q_j}{\epsilon_r r_{ij}} \tag{54}$$

where $k_e = 332.0$ kcal·Å/mol·e$^2$, $\epsilon_r = 80$ (water dielectric), and charges $q_i$ are:

- Asp, Glu: $q = -1$
- Lys, Arg: $q = +1$
- His: $q = +0.5$ (at pH 7)

**5. Clash prevention.**  Hard sphere repulsion to prevent atomic overlaps:

$$E_{\text{clash}} = \sum_{\substack{i,j \\ r_{ij}<r_{\text{clash}}}} k_{\text{clash}} \left(r_{\text{clash}} - r_{ij}\right)^4 \tag{55}$$

where $r_{\text{clash}} = 0.8 \cdot (\sigma_i + \sigma_j)/2$ and $k_{\text{clash}} = 1000$ kcal/mol / Å$^4$.

**6. Secondary structure preferences.**  Encourages formation of regular secondary structures:

$$E_{\text{ss}} = -\sum_i \sum_{s \in \{\alpha,\beta\}} P_s(S_i) \cdot f_s(\phi_i, \psi_i) \tag{56}$$

where $P_s(S_i)$ is the propensity of residue $S_i$ for structure $s$, and $f_s(\phi, \psi)$ is a Gaussian centered at ideal Ramachandran angles:

- $\alpha$-helix: $(\phi, \psi) = (-60, -45)$
- $\beta$-sheet: $(\phi, \psi) = (-120, 120)$

**7. Hydrogen bonding.**  Promotes backbone hydrogen bonds:

$$E_{\text{hbond}} = \sum_{(i,j) \in \mathcal{H}} -\epsilon_h \cdot f_{\text{angle}}(\eta) \cdot f_{\text{dist}}(r_{ij}) \tag{57}$$

where:

$$f_{\text{dist}}(r) = \exp\left(-\frac{(r - r_h^0)^2}{2\sigma_r^2}\right), \quad r_h^0 = 2.8\text{Å} \tag{58}$$

$$f_{\text{angle}}(\eta) = \cos^4(\eta), \quad \eta = \angle(\text{N-H}\cdots\text{O}) \tag{59}$$

with $\epsilon_h = 2.0$ kcal/mol for backbone H-bonds.

**SE(3)-invariance of physics energy.**  All energy terms are constructed to be SE(3)-invariant:

- **Distances**: $\|x_i - x_j\|$ invariant under rotation and translation
- **Angles**: $\arccos(\mathbf{v}_1 \cdot \mathbf{v}_2/|\mathbf{v}_1||\mathbf{v}_2|)$ invariant
- **Dihedrals**: Four-body angles invariant
- **No absolute positions**: All computations use relative coordinates

Therefore: $E_{\text{phys}}(\mathbf{R}X + \mathbf{t}) = E_{\text{phys}}(X)$ for any $\mathbf{R} \in \text{SO}(3), \mathbf{t} \in \mathbb{R}^3$.

**Gradient computation and application.** During diffusion sampling at timestep $t$:

1. **Partial decoding**:

$$z_x^t \xrightarrow{\text{decoder}} \widehat{X}^t \in \mathbb{R}^{L \times C \times 3} \tag{60}$$

2. **Energy evaluation**:

$$E^t = E_{\text{phys}}(\widehat{X}^t, S; M) = \sum_i \lambda_i E_i(\widehat{X}^t, S; M) \tag{61}$$

3. **Gradient computation**:

$$\nabla_{z_x} E_{\text{phys}} = \frac{\partial E^t}{\partial \widehat{X}^t} \cdot \frac{\partial \widehat{X}^t}{\partial z_x^t} \tag{62}$$

4. **Guidance application**:

$$\tilde{\varepsilon}_x^t = \varepsilon_\theta(z_x^t, t) - \sqrt{1 - \bar{\alpha}_t} \cdot \lambda_{\text{phys}} \cdot \nabla_{z_x} E_{\text{phys}} \tag{63}$$

**Adaptive weighting and scheduling.** The physics guidance weight $\lambda_{\text{phys}}$ can be:

- **Time-dependent**: $\lambda_{\text{phys}}(t) = \lambda_0 \cdot (1 - t/T)$ (stronger near end)
- **Energy-dependent**: $\lambda_{\text{phys}}(E) = \lambda_0 \cdot \tanh(E/E_0)$ (adaptive to quality)
- **Component-specific**: Different weights for each energy term

**Computational optimizations.** To make physics guidance tractable:

- **Cutoff distances**: Only compute interactions within $r_{\text{cut}} = 10\text{Å}$
- **Neighbor lists**: Pre-compute interaction pairs
- **Approximations**: Use smooth approximations for discontinuous potentials
- **Gradient clipping**: $\|\nabla E\|_\infty \leq \tau$ to prevent instabilities

**Role and feasibility of using OpenMM with C$\alpha$-level coordinates.** In our implementation, OpenMM is used in a restricted and coarse-grained way: Backbone-only topology and C$\alpha$ coordinates. At the (clean) end of each training step, our Amber14 OpenMM wrapper constructs a minimal peptide model containing only backbone atoms (N–C$\alpha$–C–O per residue, no side chains) from the sequence and assigns standard Amber14 parameters to those backbone atoms. The positions we supply come from the model's C$\alpha$ outputs; for simplicity and efficiency we (i) evaluate the Amber energy/forces on this backbone-only system and (ii) project the resulting forces back onto the C$\alpha$ channel used by the diffusion model.

Single, weak, clamped loss on clean structures. This OpenMM term is evaluated once per batch on the clean structure, not at every diffusion timestep, and it is down-weighted by small energy/force scales and an overall loss weight. We also explicitly clamp both energies and force magnitudes to fixed bounds before using them in the loss.

As a result, the OpenMM contribution acts as a coarse, low-weight backbone regularizer rather than a full high-fidelity all-atom simulation. We do not claim to obtain exact physical forces for detailed side-chain conformations; instead, we use OpenMM only to inject a modest, physically motivated signal that is numerically stable and compatible with our C$\alpha$-level training setup.

## P.5 EVOLUTIONARY SEQUENCE GUIDANCE

Evolutionary guidance leverages billions of years of natural selection encoded in protein sequences to guide peptide generation toward biologically viable designs. This component addresses a critical limitation of purely physics-based approaches: while physics ensures structural validity, it doesn't guarantee biological function or evolutionary plausibility.

**Biological motivation.** Natural proteins have been optimized through evolution for stability, function, and interaction specificity. By incorporating evolutionary signals, we bias generation toward sequence patterns that have proven successful in nature. This is particularly important for:

- **Functional motifs**: Conserved patterns essential for biological activity
- **Fold stability**: Amino acid preferences that promote proper folding
- **Interaction interfaces**: Residue combinations favorable for binding

**BLOSUM-based evolutionary embeddings.** We initialize amino acid representations using the BLOSUM62 substitution matrix, which captures evolutionary relationships between amino acids:

$$\mathbf{e}_i^{\text{BLOSUM}} = \text{BLOSUM}_{i,:} \in \mathbb{R}^{20}, \quad i \in \{0, \dots, 19\} \tag{64}$$

where each row represents substitution scores for amino acid $i$. These embeddings encode:

- Physicochemical similarity (e.g., hydrophobic: I-L-V)
- Functional equivalence (e.g., charged: D-E, K-R)
- Conservation patterns (high self-scores for W, C, P)

The BLOSUM embeddings are projected into the latent space:

$$\mathbf{h}_{\text{evo}} = \mathbf{h} + \alpha \cdot \text{MLP}_{\text{evo}}(\mathbf{e}^{\text{BLOSUM}}[S]) \tag{65}$$

where $\alpha$ is a learnable weight balancing evolutionary and structural information.

**Self-supervised evolutionary fitness scoring.** We predict sequence viability using a self-supervised evolutionary fitness network:

$$f_{\text{fitness}}(\mathbf{z}_h) = \sigma\left(\text{MLP}_{\text{fit}}(\text{Pool}(\mathbf{h}_{\text{evo}}))\right) \in [0, 1] \tag{66}$$

where $\sigma$ is sigmoid activation. The fitness score estimates the probability that a sequence is evolutionarily viable, trained on:

- Natural sequences: $f_{\text{target}} \approx 0.8 - 1.0$
- Random sequences: $f_{\text{target}} \approx 0.0 - 0.2$
- Mutated sequences: $f_{\text{target}} \propto$ stability

**Position-specific conservation.** We model position-specific amino acid preferences through a conservation predictor:

$$p_{\text{cons}}(S_i|i, \mathbf{h}) = \text{softmax}(\mathbf{W}_{\text{cons}} \cdot \mathbf{h}_i + \mathbf{b}_{\text{pos}[i]}) \tag{67}$$

This captures:

- **Structural constraints**: Proline in turns, glycine in tight loops
- **Hydrophobic core**: Preference for I, L, V, F in buried positions
- **Surface preferences**: K, R, D, E in exposed regions

**Residue dependency attention modeling.** Correlated mutations reveal functional coupling between positions. We capture this through multi-head attention:

$$\mathbf{Q} = \mathbf{h}_{\text{evo}}\mathbf{W}_Q, \quad \mathbf{K} = \mathbf{h}_{\text{evo}}\mathbf{W}_K, \quad \mathbf{V} = \mathbf{h}_{\text{evo}}\mathbf{W}_V \tag{68}$$

$$\mathbf{A} = \text{softmax}\left(\frac{\mathbf{Q}\mathbf{K}^T}{\sqrt{d_k}}\right) \tag{69}$$

$$\mathbf{h}_{\text{coevo}} = \mathbf{h}_{\text{evo}} + \mathbf{A}\mathbf{V} \tag{70}$$

The attention weights $\mathbf{A}_{ij}$ identify co-evolving position pairs, such as:

- Salt bridges: (D/E)-(K/R) pairs
- Disulfide bonds: C-C pairs
- Hydrophobic clusters: coordinated I/L/V patterns

**Evolutionary energy function.**    The total evolutionary guidance combines multiple terms:

$$E_{\text{evo}}(\mathbf{z}_h) = -w_1 f_{\text{fitness}} + w_2 \mathcal{H}(p_{\text{cons}}) - w_3 \log p_{\text{coevo}} \tag{71}$$

where:

- $f_{\text{fitness}}$: Overall sequence viability (maximize)
- $\mathcal{H}(p_{\text{cons}})$: Conservation entropy (balance diversity)
- $p_{\text{coevo}}$: Residue dependency attention consistency score

**Gradient computation.**    During diffusion sampling at timestep $t$:

$$\nabla_{\mathbf{z}_h} E_{\text{evo}} = -w_1 \nabla_{\mathbf{z}_h} f_{\text{fitness}} + w_2 \nabla_{\mathbf{z}_h} \mathcal{H} - w_3 \nabla_{\mathbf{z}_h} \log p_{\text{coevo}} \tag{72}$$

This gradient is computed by:

1. Partially decode $\mathbf{z}_h^t \rightarrow \hat{S}^t$ (sequence probabilities)
2. Evaluate evolutionary scores
3. Backpropagate through the self-supervised evolutionary fitness and conservation networks

**Advantages of evolutionary guidance.**

- **Biological relevance**: Generated sequences resemble natural proteins
- **Functional bias**: Promotes sequences likely to fold and function
- **Diversity**: Conservation entropy prevents convergence to single solutions
- **Interpretability**: Attention weights reveal important interactions

### P.6    MUTUAL INFORMATION FOR SEQUENCE-STRUCTURE CONSISTENCY

**Theoretical motivation.**    In natural proteins, sequence fully determines structure (Anfinsen's principle). This deterministic relationship implies maximal mutual information: $I(S; X) = H(X)$ where $H(X)$ is the structure entropy. During generation, we must maintain this tight coupling to ensure:

- **Foldability**: Sequence can actually fold into the generated structure
- **Uniqueness**: Structure is the native fold for the sequence
- **Stability**: Sequence-structure pair is thermodynamically favorable

**Information-theoretic foundation.**    Mutual information quantifies the reduction in uncertainty about one variable given knowledge of another:

$$I(S; X) = H(S) + H(X) - H(S, X) \tag{73}$$

For peptide co-design, we decompose this into latent space:

$$I(\mathbf{z}_h; \mathbf{z}_x) = \mathbb{E}_{p(\mathbf{z}_h, \mathbf{z}_x)} \left[ \log \frac{p(\mathbf{z}_h, \mathbf{z}_x)}{p(\mathbf{z}_h) p(\mathbf{z}_x)} \right] \tag{74}$$

where $\mathbf{z}_h$ and $\mathbf{z}_x$ are sequence and structure latents respectively.

**MINE estimator.**    Since direct computation of MI is intractable for continuous high-dimensional variables, we employ the Mutual Information Neural Estimator (MINE):

$$I(\mathbf{z}_h; \mathbf{z}_x) \geq \mathbb{E}_{p(\mathbf{z}_h, \mathbf{z}_x)}[T_\theta(\mathbf{z}_h, \mathbf{z}_x)] - \log(\mathbb{E}_{p(\mathbf{z}_h)p(\mathbf{z}_x)}[e^{T_\theta(\mathbf{z}_h, \mathbf{z}_x)}]) \tag{75}$$

where $T_\theta : \mathbb{R}^{d_h} \times \mathbb{R}^{d_x} \rightarrow \mathbb{R}$ is a neural network (statistics network) that learns to distinguish between:

- Joint samples: $(\mathbf{z}_h, \mathbf{z}_x) \sim p(\mathbf{z}_h, \mathbf{z}_x)$ (matched pairs)
- Product samples: $\mathbf{z}_h \sim p(\mathbf{z}_h), \mathbf{z}_x \sim p(\mathbf{z}_x)$ (independent)

**Statistics network architecture.** The MINE statistics network processes sequence-structure pairs:

$$T_\theta(\mathbf{z}_h, \mathbf{z}_x) = \mathrm{MLP}_{\mathrm{final}}(\mathrm{concat}[\phi_h(\mathbf{z}_h), \phi_x(\mathbf{z}_x), \psi(\mathbf{z}_h \odot \mathbf{z}_x)]) \tag{76}$$

where:

- $\phi_h$: Sequence feature extractor (captures motifs, conservation)
- $\phi_x$: Structure feature extractor (captures geometry, contacts)
- $\psi$: Cross-modal interaction network
- $\odot$: Element-wise product for capturing correlations

**Training the MINE estimator.** The statistics network is trained to maximize the lower bound:

$$\mathcal{L}_{\mathrm{MINE}} = \mathbb{E}_{(\mathbf{z}_h, \mathbf{z}_x) \sim p_{\mathrm{joint}}}[T_\theta(\mathbf{z}_h, \mathbf{z}_x)] \tag{77}$$

$$- \mathbb{E}_{\mathbf{z}_h \sim p_h, \tilde{\mathbf{z}}_x \sim p_x}[\log(1 + e^{T_\theta(\mathbf{z}_h, \tilde{\mathbf{z}}_x)})] \tag{78}$$

where $\tilde{\mathbf{z}}_x$ are structure samples shuffled across the batch to break sequence-structure correspondence.

## Q    DETAILED METRICS CALCULATION

The pipeline calculates five critical metrics for evaluating peptide design quality. Each metric addresses different aspects of structural and functional prediction accuracy. This document provides detailed mathematical formulations and biological significance for each metric.

### Q.1    BINDING FREE ENERGY SUCCESS RATE

#### Q.1.1    DEFINITION

The **Binding Free Energy Success Rate** measures the percentage of predicted peptides that exhibit favorable binding thermodynamics. To evaluate the best performance of the model. We calculate the

#### Q.1.2    MATHEMATICAL FORMULATION

$$\mathrm{Success\ Rate} = \frac{1}{N} \sum_{i=1}^{N} \mathbb{I}(\Delta G_i < 0) \tag{79}$$

$$\mathrm{where}\ \mathbb{I}(x) = \begin{cases} 1 & \text{if } x \text{ is true} \\ 0 & \text{otherwise} \end{cases} \tag{80}$$

The binding free energy $\Delta G$ is calculated using PyRosetta interface energy:

$$\Delta G = E_{\mathrm{complex}} - E_{\mathrm{receptor}} - E_{\mathrm{peptide}} \tag{81}$$

#### Q.1.3    CALCULATION PROCESS

1. For each predicted peptide structure, perform energy minimization using PyRosetta FastRelax
2. Calculate interface energy between receptor and peptide chains
3. Determine if $\Delta G < 0$ or $\Delta G < -5$ (favorable binding following criteria)
4. Compute success rate across all predictions

### Q.1.4 SIGNIFICANCE IN PEPTIDE DESIGN

- **Thermodynamic Viability**: Ensures predicted peptides can actually bind to target proteins
- **Drug Development**: Critical for therapeutic peptide design
- **Functional Validation**: Confirms structural predictions have biological relevance
- **Design Optimization**: Guides model training toward energetically favorable conformations

**Report:** To evaluate the best performance of the model, we report the median of the minimum $\Delta G$ value for each target, as the median is more robust to outliers than the mean. To capture variability across all test targets, we report the standard deviation. In practice, some failed complexes may yield extreme values of REU.

### Q.2 DOCKQ SCORE

#### Q.2.1 DEFINITION

DockQ is a continuous, bounded measure of docking model quality that combines three CAPRI-style criteria—fraction of native contacts ($F_{\text{nat}}$), interface RMSD (iRMSD), and ligand RMSD (LRMSD)—into a single score in $[0, 1]$. Higher is better.

#### Q.2.2 MATHEMATICAL FORMULATION

DockQ normalizes iRMSD and LRMSD with saturating transforms and averages them with $F_{\text{nat}}$:

$$\text{DockQ} = \frac{1}{3}\left(F_{\text{nat}} + \frac{1}{1 + \left(\frac{\text{iRMSD}}{1.5}\right)^2} + \frac{1}{1 + \left(\frac{\text{LRMSD}}{8.5}\right)^2}\right). \tag{82}$$

The constants $1.5\,\text{Å}$ (for iRMSD) and $8.5\,\text{Å}$ (for LRMSD) follow the original DockQ calibration to CAPRI categories.

**Components.**

$$F_{\text{nat}} = \frac{\#\{\text{native contacts recovered}\}}{\#\{\text{native contacts}\}}, \tag{83}$$

$$\text{iRMSD} = \text{RMSD over interface } C_\alpha \text{ atoms (CAPRI definition)}, \tag{84}$$

$$\text{LRMSD} = \text{RMSD of ligand (peptide) } C_\alpha \text{ atoms after superposition on the receptor.} \tag{85}$$

**Report:** For each target, we report the mean DockQ value. To capture variability across all test targets, we report the standard deviation. Complexes that failed were assigned a score of 0. Backbone models were calculated using $C\alpha$ settings.

#### Q.2.3 CONTACT AND INTERFACE DEFINITIONS

**Native contacts** (for $F_{\text{nat}}$) are residue pairs (one from each partner) that have any heavy-atom distance $\leq 5.0\,\text{Å}$ in the reference complex. We count a contact as "recovered" if the same residue pair is within $5.0\,\text{Å}$ in the prediction.

**Interface residues** (for iRMSD) follow CAPRI practice: residues whose any heavy atom in the reference complex lies within a chosen cutoff (typically $10\,\text{Å}$) of any atom in the binding partner. iRMSD is then computed as the RMSD over the interface $C_\alpha$ atoms after the standard superposition (as in CAPRI).

#### Q.2.4 SIGNIFICANCE IN PEPTIDE DESIGN

- **Interface nativeness:** DockQ summarizes how well a predicted peptide–protein interface matches the reference geometry (contacts and pose).
- **Comparability:** The bounded transforms make iRMSD/LRMSD commensurate with $F_{\text{nat}}$, enabling a single score.

- **Community alignment:** DockQ correlates with CAPRI quality classes and is widely used to compare docking methods.

**Quality thresholds** (commonly used):

- DockQ $\geq 0.80$: High quality (near-native)
- DockQ $\geq 0.49$: Medium quality
- DockQ $\geq 0.23$: Acceptable quality
- DockQ $< 0.23$: Incorrect

## Q.3 GLOBAL DISTANCE TEST TOTAL SCORE (GDT_TS)

### Q.3.1 DEFINITION

GDT_TS measures the percentage of residues that can be superimposed within multiple distance thresholds after optimal structural alignment. It is superior to TM-score for short peptides.

### Q.3.2 MATHEMATICAL FORMULATION

$$\text{GDT\_TS} = \frac{1}{4}\left(\text{GDT}_1 + \text{GDT}_2 + \text{GDT}_4 + \text{GDT}_8\right) \tag{86}$$

$$\text{GDT}_d = \frac{1}{N}\sum_{i=1}^{N}\mathbb{I}\left(||r_i^{\text{ref}} - T(r_i^{\text{pred}})||_2 \leq d\right) \tag{87}$$

where:

- $d \in \{1, 2, 4, 8\}$ are distance thresholds in Angstroms
- $T$ represents optimal superposition transformation (Kabsch algorithm)
- $r_i^{\text{ref}}, r_i^{\text{pred}}$ are reference and predicted coordinates for residue $i$
- $N$ is the number of residues

### Q.3.3 INDIVIDUAL THRESHOLD INTERPRETATION

$$\text{GDT}_1 : \text{Ultra-high precision (crystallographic quality)} \tag{88}$$
$$\text{GDT}_2 : \text{High precision (functional accuracy)} \tag{89}$$
$$\text{GDT}_4 : \text{Good structure (correct fold)} \tag{90}$$
$$\text{GDT}_8 : \text{Acceptable structure (gross topology)} \tag{91}$$

### Q.3.4 SIGNIFICANCE IN PEPTIDE DESIGN

- **Length Independence**: No normalization bias for short peptides (unlike TM-score)
- **Multi-scale Assessment**: Captures both precision and overall fold quality
- **Better Discrimination**: More sensitive quality assessment for peptides $< 50$ residues
- **Functional Relevance**: Higher GDT_TS correlates with binding site accuracy

### Q.3.5 QUALITY BENCHMARKS FOR PEPTIDES

- GDT_TS $> 0.7$: Excellent model (publication-worthy)
- GDT_TS $> 0.5$: Good model (functionally relevant)
- GDT_TS $> 0.3$: Acceptable model (some utility)
- GDT_TS $< 0.3$: Poor model (requires improvement)

**Report:** For each target, we report the **mean of the maximum $GDT\_TS$ value**.

## Q.4 CONTACT F1 SCORE

### Q.4.1 DEFINITION

**Contact_F1** measures the accuracy of predicting inter-residue contacts using the harmonic mean of precision and recall.

## Q.5 MATHEMATICAL FORMULATION

$$\text{Contact\_F1} = \frac{2 \times \text{Precision} \times \text{Recall}}{\text{Precision} + \text{Recall}} \tag{92}$$

$$\text{Precision} = \frac{\text{TP}}{\text{TP} + \text{FP}} \tag{93}$$

$$\text{Recall} = \frac{\text{TP}}{\text{TP} + \text{FN}} \tag{94}$$

where:

$$\text{TP} = |\{(i,j) : C_{ij}^{\text{ref}} = 1 \land C_{ij}^{\text{pred}} = 1\}| \tag{95}$$

$$\text{FP} = |\{(i,j) : C_{ij}^{\text{ref}} = 0 \land C_{ij}^{\text{pred}} = 1\}| \tag{96}$$

$$\text{FN} = |\{(i,j) : C_{ij}^{\text{ref}} = 1 \land C_{ij}^{\text{pred}} = 0\}| \tag{97}$$

### Q.5.1 CONTACT MAP DEFINITION

Contacts are defined with sequence separation constraint:

$$C_{ij} = \mathbb{I}\left(||r_i - r_j||_2 \leq 8.0 \text{ Å} \land |i - j| \geq 2\right) \tag{98}$$

### Q.5.2 SIGNIFICANCE IN PEPTIDE DESIGN

- **Local Interaction Accuracy**: Measures spatial relationship prediction quality
- **Functional Prediction**: Contacts determine binding specificity and affinity
- **Binding Site Assessment**: Critical for peptide-protein interaction prediction
- **Design Validation**: High Contact_F1 indicates reliable interaction patterns

### Q.5.3 PERFORMANCE INTERPRETATION:

- Contact_F1 > 0.6: Excellent contact prediction (highly reliable)
- Contact_F1 > 0.5: Good contact accuracy (useful for drug design)
- Contact_F1 > 0.4: Acceptable prediction (some functional value)
- Contact_F1 < 0.4: Poor contact accuracy (unreliable for design)

**Report:** We report the mean with standard deviation of the maximum $GDT\_TS$ value for each target.

## Q.6 LOCAL RMSD

### Q.6.1 DEFINITION

**Local_RMSD** measures regional structural accuracy using sliding window analysis, providing more detailed assessment than global RMSD.

### Q.6.2 MATHEMATICAL FORMULATION

$$\text{Local\_RMSD}_w = \sqrt{\frac{1}{w} \sum_{k=1}^{w} ||r_{i+k}^{\text{ref}} - T_w(r_{i+k}^{\text{pred}})||_2^2} \tag{99}$$

$$\text{Local\_RMSD\_Mean} = \frac{1}{N-w+1} \sum_{i=1}^{N-w+1} \text{Local\_RMSD}_w(i) \tag{100}$$

where:

- $w = 5$ is the window size (5 consecutive residues)
- $T_w$ is an optimal superposition transformation for window $w$
- $N$ is total number of residues
- Window $i$ spans residues $[i, i+w-1]$

### Q.6.3 ADDITIONAL STATISTICS

$$\text{Local\_RMSD\_Min} = \min_{i=1}^{N-w+1} \text{Local\_RMSD}_w(i) \tag{101}$$

$$\text{Local\_RMSD\_Max} = \max_{i=1}^{N-w+1} \text{Local\_RMSD}_w(i) \tag{102}$$

$$\text{Local\_RMSD\_Std} = \sqrt{\frac{1}{N-w+1} \sum_{i=1}^{N-w+1} (\text{Local\_RMSD}_w(i) - \text{Mean})^2} \tag{103}$$

### Q.6.4 SIGNIFICANCE IN PEPTIDE DESIGN

- **Regional Quality Assessment**: Identifies well-predicted vs poorly-predicted regions
- **Functional Region Analysis**: Key binding regions may be accurate despite poor global structure
- **Design Optimization**: Guides focused improvement of specific peptide regions
- **Flexibility Analysis**: Shows structural variation along peptide sequence

### Q.6.5 QUALITY INTERPRETATION

- Local_RMSD $< 2.0$ Å: Excellent regional precision
- Local_RMSD $< 3.0$ Å: Good regional structure
- Local_RMSD $< 5.0$ Å: Acceptable regional quality
- Local_RMSD $> 5.0$ Å: Poor regional structure

**Report:** we report the mean and the standard deviation of the $Local\ RMSD(Mean)$ value for each target.

### Q.7 C$\alpha$ CLASH METRICS (PRIMARY GEOMETRY METRIC)

### Q.7.1 DEFINITION

We use **CA_Clash_in** and **CA_Clash_out** as our primary geometry metrics, following the UniMoMo definition. They quantify steric crowding at the C$\alpha$ level:

- **CA_Clash_in**: fraction of ligand residues whose C$\alpha$ atom is too close to another ligand C$\alpha$ (internal clashes).
- **CA_Clash_out**: fraction of ligand residues whose C$\alpha$ atom is too close to any receptor C$\alpha$ (interface clashes).

A clash is defined by a C$\alpha$–C$\alpha$ distance below a fixed threshold $d_{\text{clash}} = 3.6574$ Å.

### Q.7.2 Mathematical Formulation

Let $\{\mathbf{x}_i^{\text{lig}}\}_{i=1}^{N_{\text{lig}}}$ be ligand C$\alpha$ coordinates (in sequence order), and $\{\mathbf{y}_j^{\text{rec}}\}_{j=1}^{N_{\text{rec}}}$ receptor C$\alpha$ coordinates.

**Internal clashes (CA_Clash_in).** We ignore self-pairs and immediate sequence neighbors ($i, i \pm 1$). A ligand residue $i$ is in clash if

$$\exists\, k \notin \{i-1, i, i+1\} : \ \|\mathbf{x}_i^{\text{lig}} - \mathbf{x}_k^{\text{lig}}\|_2 < d_{\text{clash}}.$$

Then

$$\text{CA\_Clash\_in} = \frac{\#\{\text{clashing ligand residues}\}}{N_{\text{lig}}} \times 100\ (\%).$$

**Interface clashes (CA_Clash_out).** A ligand residue $i$ is in clash with the receptor if

$$\exists\, j : \ \|\mathbf{x}_i^{\text{lig}} - \mathbf{y}_j^{\text{rec}}\|_2 < d_{\text{clash}}.$$

Then

$$\text{CA\_Clash\_out} = \frac{\#\{\text{ligand residues clashing with receptor}\}}{N_{\text{lig}}} \times 100\ (\%).$$

### Q.7.3 Significance in Peptide Design

- **CA_Clash_in**: detects over-packed or self-colliding peptide backbones.
- **CA_Clash_out**: measures whether the peptide backbone fits into the binding pocket without penetrating the receptor.
- **Model comparison**: these two percentages are directly comparable across methods (e.g., UniMoMo vs ours).

**Report:** we report the mean and standard deviation of CA_Clash_in and CA_Clash_out across all designs for each target.

## Q.8 Backbone Bond Length Outliers

### Q.8.1 Definition

**BondLength_Outlier** fractions measure how often backbone bonds (N–CA, CA–C, C–O, C–N) deviate from ideal lengths by more than a threshold $T$.

For each bond $k$:

$$\Delta_k = |d_k - d_k^{\text{ideal}}|, \quad I_k^{(T)} = \begin{cases} 1, & \Delta_k > T \\ 0, & \text{otherwise} \end{cases}$$

$$\text{Outlier\_Frac}^{(T)} = \frac{1}{M} \sum_{k=1}^{M} I_k^{(T)}, \quad T \in \{0.10, 0.20, 0.50\}\ \text{Å}.$$

We report these fractions globally and for the ligand backbone only, as mean ± std across designs.

## Q.9 Sequence Diversity

### Q.9.1 Mathematical Formulation

$$\text{Similarity}_{ij} = \frac{\text{BLOSUM62}(S_i, S_j)}{\sqrt{\text{BLOSUM62}(S_i, S_i) \times \text{BLOSUM62}(S_j, S_j)}} \tag{104}$$

$$\text{Distance}_{ij} = 1 - \text{Similarity}_{ij} \tag{105}$$

$$\text{Sequence Diversity} = \frac{\text{Unique Clusters}}{N} \quad (\text{threshold} = 0.4) \tag{106}$$

where, $N$ is the number of generated complexes.

### Q.9.2 SIGNIFICANCE

Measures amino acid sequence space exploration. Higher diversity indicates better functional exploration and prevents mode collapse.

### Q.9.3 INTERPRETATION

$$> 0.6 : \text{Excellent exploration} \tag{107}$$
$$> 0.4 : \text{Good diversity} \tag{108}$$
$$< 0.3 : \text{Mode collapse concern} \tag{109}$$

## Q.10 SEQUENCE VALIDITY RATE

Given $N$ generated sequences and assay pH (default $pH = 7.4$),

$$\text{Valid count} = \sum_{i=1}^{n} \mathbf{1}\left[\text{NetChargeOK}_i \wedge \text{GRAVY\_OK}_i \wedge \text{Instability\_OK}_i \wedge \text{pI\_OK}_i\right]$$

$$\text{Valid rate} = \frac{\text{Valid count}}{N}.$$

The four criteria are

$$\begin{aligned} \text{NetChargeOK}_i &\iff \text{net\_charge}_i(pH) \in [-2,\ 4], \\ \text{GRAVY\_OK}_i &\iff \text{GRAVY}_i \in [-1.0,\ 0.5], \\ \text{Instability\_OK}_i &\iff \text{instability\_index}_i < 40, \\ \text{pI\_OK}_i &\iff \left| \text{pI}_i - pH \right| > 0.5. \end{aligned}$$

## Q.11 STRUCTURAL DIVERSITY

### Q.11.1 MATHEMATICAL FORMULATION

$$\text{RMSD}_{ij} = \sqrt{\frac{1}{L} \sum_{k=1}^{L} ||x_i^{(k)} - x_j^{(k)}||_2^2} \tag{110}$$

$$\text{Structural Diversity} = \frac{\text{Unique Structure Clusters}}{N} \quad (\text{threshold} = 4.0 \text{ Å}) \tag{111}$$

where $x_i^{(k)}$ are C$\alpha$ coordinates of residue $k$ in structure $i$.

### Q.11.2 SIGNIFICANCE

Quantifies conformational space coverage. Critical for binding versatility and allosteric mechanisms.

### Q.11.3 INTERPRETATION

$$> 0.5 : \text{Excellent conformational exploration} \tag{112}$$
$$> 0.3 : \text{Good structural variation} \tag{113}$$
$$< 0.2 : \text{Limited conformational coverage} \tag{114}$$

## Q.12 Consistency

### Q.12.1 Mathematical Formulation

Given sequence clusters $\mathbf{C}_{\text{seq}}$ and structural clusters $\mathbf{C}_{\text{struct}}$:

$$\mathbf{T}_{ij} = |\{k : \mathbf{C}_{\text{seq}}[k] = i \wedge \mathbf{C}_{\text{struct}}[k] = j\}| \tag{115}$$

$$\chi^2 = \sum_{i,j} \frac{(\mathbf{T}_{ij} - E_{ij})^2}{E_{ij}} \tag{116}$$

$$\text{Consistency} = \sqrt{\frac{\chi^2}{N \times (\min(n_{\text{seq}}, n_{\text{struct}}) - 1)}} \tag{117}$$

where $E_{ij}$ are expected frequencies under independence assumption.

### Q.12.2 Significance

Measures correlation between sequence and structural clustering using Cramér's V. Tests biological constraint preservation (similar sequences $\rightarrow$ similar structures).

## Q.13 TM-score

$$\text{TM-score} = \max_{\mathcal{A}} \; \frac{1}{L_{\text{ref}}} \sum_{(i,j) \in \mathcal{A}} \frac{1}{1 + \left( \frac{\| r_i^{\text{ref}} - T_{\mathcal{A}}^\star(r_j^{\text{pred}}) \|}{d_0(L_{\text{ref}})} \right)^2} \tag{118}$$

$$d_0(L_{\text{ref}}) = \max\left(0.5, \; 1.24 \left(L_{\text{ref}} - 15\right)^{1/3} - 1.8\right) \tag{119}$$

where:

- $r_i^{\text{ref}}, r_j^{\text{pred}}$ are $C_\alpha$ coordinates of the reference and predicted peptides.
- $\mathcal{A}$ is a residue–residue alignment (index pairs $(i, j)$); we choose the alignment that maximizes TM-score.
- $T_{\mathcal{A}}^\star$ is the optimal rigid transform via Kabsch on aligned $C_\alpha$ pairs.
- $L_{\text{ref}}$ is the reference sequence length; $d_0(L_{\text{ref}})$ is the length-dependent scale (with a 0.5 Å floor for very short peptides).

### Q.13.1 Discussion: TM-score in Peptide Design Context

TM-score is a global fold-similarity metric for full proteins, and it breaks down for short, flexible peptides and interface design. It doesn't reflect what actually matters in peptide design (pose at the binding site, contacts, and energetics). TM-score isn't great for ranking peptide binders, but we still squeeze value out of it in a few very specific, low-stakes roles.

## Q.14 Sliding-AAR (Amino Acid Recovery)

### Q.14.1 Definition

**Sliding-AAR** measures the maximum sequence identity between generated and reference peptides across all possible alignments, accounting for potential positional shifts in the generated sequence.

## Q.15 Mathematical Formulation

$$\text{Sliding-AAR} = \max k \in K \left( \text{Sliding-AAR}(S_{\text{cand}}^{(k)}, S_{\text{ref}}) \right) \tag{120}$$

$$\text{Sliding-AAR}(S_1, S_2) = \frac{1}{L} \sum_{i=1}^{L} \mathbb{I}(S_1[i] = S_2[i]) \tag{121}$$

where:

$$K = \{-(L_{\text{ref}} - 1), ..., 0, ..., L_{\text{cand}} - 1\} \tag{122}$$

$$S_{\text{cand}}^{(k)}[i] = \begin{cases} \text{pad} & \text{if } i - k < 0 \text{ or } i - k \geq L_{\text{cand}} \\ S_{\text{cand}}[i - k] & \text{otherwise} \end{cases} \tag{123}$$

### Q.15.1 ALIGNMENT PROCESS

The algorithm evaluates all possible alignments:

$$\text{Alignments} = \{S_{\text{cand}}[k : k + L_{\text{ref}}] : 0 \leq k \leq L_{\text{cand}} + L_{\text{ref}} - 2\} \tag{124}$$

where the candidate sequence is padded with $(L_{\text{ref}} - 1)$ special tokens on each side.

### Q.15.2 DISCUSSION: SLIDING-AAR IN PEPTIDE DESIGN CONTEXT

Sliding-AAR poorly suits peptide design because peptides require exact positioning for function. Their short length (5-30 residues) means single-position shifts often destroy binding activity, making alignment flexibility counterproductive. The metric's fundamental flaw is the sequence-structure disconnect: high Sliding-AAR doesn't ensure functional similarity. Two peptides with 80% sliding similarity may have entirely different structures and no binding activity. This creates dangerous false positives where designed peptides score well but fail functionally. Structure-based metrics (Contact F1, Local RMSD, binding energy) directly assess molecular recognition requirements and better predict function. Sliding-AAR should remain supplementary, useful only for motif identification or initial diversity screening, never as a primary validation metric for functional peptide design.

