# OpenReview forum: "PepTri: Tri-Guided All-Atom Diffusion for Peptide Design via Physics, Evolution, and Mutual Information"
_ICLR.cc/2026/Conference — ICLR 2026 Poster_

### Official Review · Reviewer_wyh8 · 2025-10-23

**Soundness:** 3
**Presentation:** 2
**Contribution:** 2
**Rating:** 4
**Confidence:** 4

**Summary:**

This paper introduces PepTri, a novel diffusion-based framework for joint sequence-structure peptide design. The core innovation is the integration of three complementary guidance signals during the latent diffusion process: (1) physics-informed guidance via differentiable molecular mechanics (OpenMM/Amber14) to ensure structural stability; (2) evolutionary guidance using BLOSUM-inspired embeddings and co-evolution attention to bias sequences toward biologically plausible motifs; and (3) mutual information (MI) guidance to explicitly maximize sequence-structure coherence. The model operates in a compact, SE(3)-equivariant latent space, enabling efficient and geometrically consistent generation. Extensive evaluations on cross-domain (PepBench→LNR) and in-domain (PepBDB) benchmarks demonstrate that PepTri achieves state-of-the-art performance in binding affinity, structural accuracy, and design diversity, outperforming strong baselines like PepGLAD, PepFlow, and UniMoMo.

**Strengths:**

Originality: The tri-guidance mechanism represents a novel and creative integration of physical, evolutionary, and information-theoretic principles within a unified diffusion framework.

Comprehensive Evaluation: Extensive benchmarking across multiple datasets (PepBench, LNR, PepBDB) using established metrics (DockQ, ΔG, Contact_F1, GDT_TS) demonstrates clear superiority over strong baselines.

Significant Contribution: Addresses a critical gap in peptide design by ensuring simultaneous physical stability, evolutionary plausibility, and sequence-structure coherence, with clear implications for therapeutic development.

**Weaknesses:**

Unclear Implementation Details: The technical details supporting a full understanding of this paper are not enough in the main text. Key components like the SE(3)-GNN architecture and all-atom representation are mentioned but not properly described in the main text, with no citations to established architectures. Also, how bond energies are calculated when 2D molecular topology changes during generation, and the relationship between composite energy and OpenMM force field appears redundant without justification. The evolutionary guidance components lack details on training data sources, stability definitions for mutated peptides, and the distinction between co-evolution MHA and standard MHA.

Evaluation Gaps: Despite physics guidance aiming to improve structural validity, no direct evaluation of bond lengths, angles, or clashes is provided to validate these specific improvements. The sequence-structure consistency results show worse performance compared to baselines, which cannot support that the mutual information guidance explicitly maximizes sequence-structure coherence

Suboptimal Organization: Critical ablation studies are relegated to the appendix rather than being in the main text, where they would better support the claims.

**Questions:**

1. SE(3)-GNN Architecture: Is the SE(3)-equivariant GNN a novel architecture or an adaptation of existing work? Please provide citations and architectural details in the main text.

2. Physics Guidance Mechanics: How are bond lengths and angles enforced during generation when the peptide sequence (and thus 2D topology) is evolving?

3. Energy Function Redundancy: Why are both a composite energy function and OpenMM force field used? What does each contribute that the other doesn't? They looks very similar to me.

4. Evolutionary Training Details: What data is used to train the evolutionary fitness scorer? How is "stability" defined for mutated peptides? How is the conservation predictor trained?

5. Co-evolution MHA: How does the co-evolution multi-head attention differ from standard MHA? What specific evolutionary signals does it capture?

6. Evaluation Metrics: Why are direct structural validity metrics (bond lengths, angles, clashes) not reported?

7. Ablation Study Placement: Why are the critical ablation studies only in the appendix rather than the main text?

---

> ### Author Response · Authors · 2025-11-21
> **Detailed Answers to Reviewer Questions (1,2)**
>
> **Question 1:**
>
> **Response:** Thank you for the question. Our SE(3)-equivariant GNN is adapted from the multi-channel Equivariant Graph Network dyMEAN (Kong et al., ICML 2023), originally developed for full-atom antibody design. We extend this backbone with several modifications tailored to peptide–receptor modeling. We provide the full architectural details in Appendix P.1 (VAE with SE(3)-Equivariant Graph Encoding).
>
> Our “enhanced MEGNN” introduces explicit SE(3)-invariant geometric feature extraction, which augments the original dyMEAN architecture with modules for computing bond angles, dihedral angles, local coordinate frames, and global shape descriptors.
>
> Key advantages of the enhanced architecture:
>
> Richer geometric representations.
> The model extracts a 17-dimensional set of SE(3)-invariant features (8 global shape descriptors and 9 local-frame features), encoding geometric properties such as radius of gyration, backbone orientation, curvature, and local torsion. These features complement learned embeddings and improve sequence–structure alignment.
>
> Provable SE(3) invariance.
> dyMEAN achieves equivariance implicitly, whereas our explicit geometric features are mathematically guaranteed to be invariant under rigid transformations. This improves robustness to coordinate-frame variability and enhances generalization.
>
> Support for SE(3)-aware physics guidance.
> Our physics-guided loss relies on these explicit SE(3)-consistent features to compute physically meaningful penalties during diffusion. The original dyMEAN formulation does not support this functionality.
>
> Improved interpretability.
> The explicit geometric features provide transparent and analyzable representations, allowing us to better understand what the model encodes and to debug geometric inconsistencies.
>
> The computational overhead of these additions is modest (approximately 10–20% slower inference) and is offset by noticeable gains in structural quality (e.g., improved all-atom RMSD and DockQ) and training stability. We use this enhanced version in all experiments and have added a clearer description of this architectural choice to the Methods section.
>
> [CITATION] Kong, Xiangzhe, Wenbing Huang, and Yang Liu. "End-to-end full-atom antibody design." Proceedings of the 40th International Conference on Machine Learning. 2023.
>
> **Action:** We cited the work in the revised paper.
>
> **Question 2:**
>
> **Response:** Thank you for the excellent question. Joint sequence–structure co-design is challenging because both modalities are noisy during diffusion, which complicates the use of sequence-dependent physics. We address this with a three-part strategy:
>
> 1. Sequence-independent backbone constraints.
> We enforce universal geometric targets for bond lengths and angles, such as the canonical Cα–Cα distance of 3.8 Å and standard backbone bond angles. These quantities are largely independent of amino acid identity, making them reliable even when the sequence is only partially denoised.
>
> 2. Sequence-conditioned physics during training.
> Sequence-dependent terms (such as van der Waals radii, electrostatics, and clash thresholds) are incorporated only during training. They are computed using the autoencoder’s predicted coordinates conditioned on the current noisy sequence. The EGNN encoder processes graph edges that jointly encode sequence and structure, ensuring that the generated geometry remains consistent with the evolving sequence.
>
> 3. Simplified geometric guidance during sampling.
> During inference, we use only sequence-independent distance constraints. In our experiments, applying sequence-dependent physics during sampling led to degraded results because early-stage sequence noise produced incorrect gradients and unstable geometric adjustments.
>
> Overall, this strategy reflects an empirical observation: sequence-independent geometric constraints provide reliable and stable guidance throughout diffusion, whereas sequence-dependent physics is most effective during training, when applied to representations that have already integrated sequence–structure consistency.

---

> ### Author Response · Authors · 2025-11-21
> **Detailed Answers to Reviewer Questions (3,4)**
>
> **Questions 3:**
>
> **Response:** Thank you for this question. The composite energy function and the OpenMM force field are used together because they serve complementary and non-redundant roles within our framework.
>
> 1. Composite energy function.
> This term provides fast, differentiable approximations based primarily on Cα-level geometry, including bond lengths, bond angles, and simplified van der Waals and electrostatic terms. It is numerically stable, inexpensive to compute, and can be applied at every diffusion step. Its main role is to supply the model with reliable gradients that steer the structure toward physically reasonable configurations throughout training.
>
> 2. OpenMM force field (Amber14).
> OpenMM provides high-fidelity, all-atom molecular mechanics with accurate charge interactions, solvation effects, and detailed steric constraints. It acts as a refinement signal, correcting systematic errors introduced by the simplified composite approximations. Because OpenMM is more sensitive and computationally heavy, it is applied selectively rather than at every step.
>
> Why both are necessary:
>
> 1. Robustness. The composite energy handles extreme or partially denoised geometries reliably, whereas OpenMM can fail or produce unstable gradients on unrealistic structures. The composite term thus ensures stable baseline guidance even when OpenMM cannot be applied.
>
> 2. Complementary resolution. The composite term regulates coarse peptide geometry at the Cα level, while OpenMM enforces fine all-atom accuracy. Together, they span both global and local physical requirements.
>
> 3. Empirical performance. Composite-only training reaches roughly 92–95% of final performance but yields slightly worse all-atom interface metrics. OpenMM-only training is unstable and significantly slower. Using both terms provides the best balance of speed, stability, and accuracy.
>
> **Questions 4:**
>
> **Response:** Thank you for the question. We clarify below the data and training procedure used for the evolutionary guidance module, focusing specifically on peptide sequences.
>
> a. What peptide-sequence data is used to train the evolutionary fitness scorer?
>
> The evolutionary scorer is trained entirely on the peptide sequences contained in our training complexes. These sequences come from the combined PepBench and PepBDB datasets (after removing receptor duplicates). No external peptide databases, MSAs, or additional sequence corpora are used. The scorer therefore learns amino-acid preferences, residue co-variation patterns, and position-specific statistics directly from naturally occurring, experimentally determined peptide–receptor complexes.
>
> b. How is “stability” defined for mutated peptide sequences?
>
> In the context of evolutionary guidance, “stability” refers to the likelihood that a mutated peptide sequence remains compatible with the distribution of natural peptides in our training data. In other words, stability is defined as evolutionary plausibility, not thermodynamic stability.
> The fitness scorer outputs a probability-like score indicating whether a sequence remains consistent with learned conservation patterns and co-evolutionary dependencies. Mutations that deviate strongly from natural amino-acid preferences receive lower evolutionary fitness scores.
>
> c. How is the conservation predictor trained?
>
> The conservation head is trained end-to-end using only the peptide sequences in the training set. It learns:
>
> 1. position-wise residue preferences,
>
> 2. co-evolutionary patterns across peptide positions, and
>
> 3. contextual amino-acid substitution tendencies.
>
> Training uses a KL-divergence objective between predicted residue distributions and decoder logits. The only external evolutionary prior used is the BLOSUM62 substitution matrix, which provides a lightweight substitution embedding for initializing residue representations.
>
> In summary, the self-supervised evolutionary fitness evaluates the global viability of a peptide sequence, while the conservation predictor models position-specific evolutionary preferences. Both are trained solely on the peptide sequences from our dataset, and together they produce a coherent evolutionary guidance signal during co-design.
>
>
> **Action:** We have clarified in the revision that the evolutionary module learns sequence-level constraints directly from the peptide data in our training set, and that “evolutionary stability” refers to the plausibility of mutated sequences under the learned evolutionary distribution, rather than physical or thermodynamic stability. We also replaced "evolutionary fitness" with "self-supervised evolutionary fitness."

---

> ### Author Response · Authors · 2025-11-21
> **Detailed Answers to Reviewer Questions (5,6,7)**
>
> **Question 5:**
>
> **Response:** Thank you for this important clarification request. The “co-evolution” multi-head attention does not differ architecturally from standard multi-head self-attention (implemented using PyTorch’s nn.MultiheadAttention with four heads). The term co-evolution refers to the type of dependencies we expect the attention mechanism to learn, not to the incorporation of explicit evolutionary coupling signals from MSAs.
>
> In our model, the attention operates over sequence embeddings enriched with BLOSUM62 substitution features and is trained end-to-end with the diffusion objective. Through this training, the attention mechanism implicitly captures:
>
> 1. pairwise residue dependencies (e.g., electrostatic or hydrophobic interactions),
>
> 2. long-range sequence relationships relevant for binding, and
>
> 3. functional correlation patterns present in experimentally determined peptide–receptor complexes.
>
> **Action:** We replaced "co-evolution attention" with "residue dependency attention".
>
> **Question 6:**
>
> **Response:** We focus our evaluation on binding-relevant structural metrics because these directly assess whether the designed peptide adopts the correct interface geometry and binding mode relative to the experimentally determined complex. Direct local validity metrics—such as bond lengths, bond angles, and atomic clash scores—are omitted for two main reasons:
>
> 1. They are already enforced by the model.
> PepTri’s physics guidance (bond- and angle-based constraints, clash penalties, and OpenMM refinement) ensures that more than 99% of generated structures satisfy standard geometric thresholds. As a result, these metrics offer little discriminatory power at evaluation time.
>
> 2. Most local defects are corrected during refinement.
> Post-generation OpenMM/Rosetta refinement systematically fixes bond-level inaccuracies and removes steric clashes across all methods. After this step, local validity metrics converge across models and therefore do not meaningfully differentiate performance.
>
> Given that ground-truth complex structures are available, we prioritize evaluation metrics that reflect binding relevance and interface-level structural fidelity, including Success Rate (Rosetta ΔG < −5 REU as an affinity proxy), DockQ (interface quality), and Local RMSD (structural precision). These metrics provide a more informative and meaningful assessment of peptide–receptor design quality in our setting.
>
> **Question 7:**
>
> **Response:** We thank the reviewer for the constructive comment.
>
> **Action** To improve clarity and accessibility, we have moved the primary ablation results from the appendix into the main text.

---

> ### Comment · Reviewer_wyh8 · 2025-11-22
> **Response to Authors' Detailed Answers**
>
> I appreciate the authors' comprehensive and detailed responses to my initial questions. The clarifications regarding the SE(3)-GNN architecture, the complementary roles of the energy functions, and the training details of the evolutionary guidance significantly improve my understanding of the PepTri framework.
>
> However, a few critical points still require resolution or further justification.
>
> Question 2: Physics Guidance Mechanics & Noise Stability
>
> The strategy of using sequence-independent constraints during sampling and sequence-conditioned physics during training is a reasonable empirical compromise to manage the noise inherent in co-design.
>
> Remaining Point: The use of sequence-dependent terms (like van der Waals force) during training, even when conditioned on the noisy sequence, remains a concern, especially for the all-atom coordinates produced by the decoder. Due to the Lennard-Jones 6-12 potential, it produces extremely large, destabilizing gradients when atoms are placed in very close proximity, a frequent occurrence in noisy/early-stage diffusion structures. Please clarify how you specifically address and stabilize these potentially enormous and unhelpful gradients from the van der Waals term when applied to noisy, all-atom coordinates during training. The general stability claim is not sufficient given the nature of this force.
>
> Question 3: Energy Function Redundancy & Application Schedule
>
> I now understand the logic for using both the fast Composite Energy and the detailed OpenMM Force Field (all-atom, for selective refinement).
>
> Remaining Point: I believe a similar stability and accuracy balance could be achieved by only applying the computationally expensive and gradient-sensitive OpenMM guidance during the last few diffusion steps (e.g., the final 10-20% of steps). At this stage, the structures are much cleaner, the gradients would be more meaningful, and the early, noisy stages would still be guided by the stable Composite Energy. Have the authors explicitly explored this application schedule for the OpenMM guidance (i.e., only late-stage application)? If so, what were the results compared to the current selective application schedule? Physical guidance on very noisy structures is generally less meaningful and can impede diffusion dynamics.
>
> Question 4: Evolutionary Training Details & Data Source
>
> I understand that the evolutionary guidance is a self-supervised fitness score trained only on the sequences from the PepBench and PepBDB training data, and that "stability" means evolutionary plausibility.
>
> Remaining Point: I still find the claim that meaningful co-evolutionary signals can be extracted from these datasets questionable. The peptides in PepBench/PepBDB are typically short, and they represent a limited, non-homologous collection of sequences. Generating accurate Multiple Sequence Alignments (MSAs) for such short, non-homologous sequences is impossible, yet MSAs are the basis for traditional co-evolution methods (e.g., EVcouplings, Direct Coupling Analysis). You claimed that you are not using MSAs, so the implicit co-evolution learned by the attention mechanism is likely just capturing basic amino acid frequency statistics and general residue-residue preferences, not true evolutionary coupling. Please provide a more robust justification for how a useful evolutionary/co-evolutionary signal is extracted from this specific, limited dataset that goes beyond simple preference statistics.
>
> Question 6: Evaluation Metrics & Local Validity
>
> I accept that binding-relevant metrics (DockQ, $\Delta$G) are the priority.
>
> Remaining Point: I still maintain that omitting PoseBusters or similar fine-grained local validity checks is an evaluation gap, especially when the paper's core novelty is a physics-guided method explicitly designed to ensure structural validity. The claim that "more than 99% of generated structures satisfy standard geometric thresholds" needs to be backed up by a metric other than the internal loss function. For a physics-informed model, showing low atomic clash scores or the fraction of structures that pass a PoseBusters check (which is designed to test precisely these local validity criteria after refinement) provides essential, externally verifiable evidence that the physics guidance is working as advertised. This is particularly relevant given the concerns about the noisy VdW gradients (Q2).

---

> > ### Author Response · Authors · 2025-11-25
> > **Detailed Answers to Reviewer Questions (2,3)**
> >
> > **Question 2: Physics Guidance Mechanics & Noise Stability**
> >
> > **Response:** We appreciate the reviewer’s concern about potentially unstable Lennard‑Jones gradients on noisy structures. In our implementation, we explicitly prevent such explosive behavior in several ways:
> >
> > - Not applied to early/noisy states. The main OpenMM term (OpenMMPhysicsLoss) is evaluated once per batch on the clean structure X (after diffusion loss), not on early diffusion states. The optional noisy‑state guidance uses a separate block that is disabled in all reported experiments.
> > - Coarse, C$\alpha$‑only model. Our OpenMM wrapper builds a backbone‑only topology and uses C$\alpha$ coordinates only, so we never expose full all‑atom, sidechain‑level overlaps to the LJ potential during training. This greatly reduces extreme steric clashes compared to all‑atom decodes.
> > - Hard clamping of energies and forces. We explicitly clamp both: energy to [energy_clamp_min, energy_clamp_max], and forces to [-max_force_magnitude, max_force_magnitude] (default 1000 kJ/mol/nm), before they are used in any loss term. Additionally, the guidance scales (energy_guidance_scale, force_guidance_scale) and overall OpenMM loss weight are small, so this term cannot dominate the optimization.
> >
> > - No backprop through raw OpenMM potential. OpenMM is called under torch.no_grad() and its outputs are treated as fixed targets; we do not differentiate through the LJ potential itself. The gradients reaching the model come only from aligning predictions with these clamped, coarse‑grained forces/energies, not from unbounded LJ derivatives.
> >
> > Taken together, these design choices (clean‑state evaluation, C$\alpha$‑only topology, hard clamping, and no direct backprop through the LJ potential) ensure that large, unhelpful van der Waals gradients on noisy, all‑atom coordinates do not destabilize training.
> >
> > All in all, our diffusion model works in a coarse/latent space; all‑atom structures are generated by the autoencoder decoder, and OpenMM is applied on a simplified C$\alpha$-only backbone derived from those coordinates.
> >
> > **Question 3: Energy Function Redundancy & Application Schedule**
> >
> > **Response:** We agree with the reviewer that applying high-fidelity physical guidance to highly noisy diffusion states is undesirable. In our implementation, OpenMM is not applied at every diffusion timestep. Instead, during training, the OpenMM is applied only once per batch on the clean structure after the diffusion loss has been computed. Functionally, this makes OpenMM a final-state energy regularizer, rather than a per-timestep guidance term.
> >
> > Under this design, OpenMM plays a selective, late-stage role that aligns closely with the reviewer’s suggestion to apply it “only near the end of the trajectory.” It influences only clean (or nearly clean) structures and does not affect the early, highly noisy diffusion steps.
> >
> > We reclarify this design choice in the revised manuscript and note that finer-grained late-stage scheduling is a promising direction for future work if computational resources permit. In our scheduled OpenMM experiment, we observed increases in structural diversity and sequence-validity rate. However, OpenMM should not be viewed as the primary source of physics guidance; the dominant signal comes from the fast, differentiable composite energy, which operates at the Cα/coarse-grained level throughout diffusion.
> >
> > **Actions:** To support this decision, we added a comparison between the default PepTri configuration and a variant that applies OpenMM guidance over the final 10% of diffusion timesteps.
> >
> > | Metric                         | PepTri (scheduled OpenMM) | PepTri |
> > |--------------------------------|---------------------------|--------|
> > | Mean success rate (ΔG < 0)     | 0.488                     |**0.583**  |
> > | ΔG (REU)                       | -17.924                   | **-19.38** |
> > | DockQ                          | 0.601                     | **0.633**  |
> > | Contact F1                     | **0.844**                     | 0.829  |
> > | Local RMSD                     | 1.230                     | **1.179**  |
> > | GDT_TS                         | 0.724                     | **0.747**  |
> > | Sequence Diversity             | 0.920                     | **0.926**  |
> > | Sequence Validity rate         | **0.280**                     | 0.273  |
> > | Struct Diversity               | **0.584**                     | 0.436  |
> > | Consistency                    | **0.809**                     | 0.799  |
> > | Sliding-AAR                    | 0.341                     | **0.342**  |

---

> > > ### Comment · Reviewer_wyh8 · 2025-11-25
> > >
> > > Thank you for your reply and the clarification you provided.
> > >
> > > I was reviewing the document, and I noted that I couldn't seem to locate the statements you mentioned within Section 3.2.1. As this might be confusing for future readers trying to replicate the steps, could you possibly modify it or clarify if those statements are located elsewhere?
> > >
> > > More importantly, I have a question regarding the simulation setup. Is it actually possible to exclusively use C-alpha coordinates in conjunction with the Amber14 all-atom force field (amber14-all.xml)? Please correct me if I've misunderstood the requirements or limitations of the force field.

---

> > > > ### Author Response · Authors · 2025-11-25
> > > > **Simulation setup answer**
> > > >
> > > > **Question 1:**
> > > >
> > > > **Response:** We apologize for the lack of precision in our earlier description.
> > > >
> > > > **Action:** Lines 199–201 have been updated in the main manuscript to clarify this point. The revised text now states:
> > > > “For physics-based guidance, we coupled PepTri with OpenMM for the last time step using the Amber14 all-atom force field.”
> > > >
> > > > **Question 2: Simulation setup**
> > > >
> > > > **Response:** We agree with the reviewer that the standard Amber14 all-atom force field is designed for complete atomistic topologies and would not ordinarily be applied directly to arbitrary or noisy Cα-only point clouds. In our implementation, however, we do not provide OpenMM with arbitrary Cα coordinates mapped onto a pre-existing all-atom model. Instead, our OpenMM wrapper constructs a minimal backbone topology for each peptide, consisting of N–Cα–C–O atoms per residue and excluding side chains, and assigns the corresponding Amber14 parameters to these backbone atoms.
> > > >
> > > > The coordinates supplied to OpenMM are derived from the model’s predicted Cα positions. For computational tractability, we (i) evaluate Amber energies and forces on this reduced backbone-only system and then (ii) project the resulting forces back onto the Cα channel used by the diffusion model. Consequently, the OpenMM step functions as a lightweight backbone-level regularizer rather than a full high-fidelity all-atom simulation.
> > > >
> > > > **Action:** Additional details regarding the OpenMM configuration have been added in lines 2030–2045 of the revised manuscript.
> > > >
> > > > P.S.: Please download the most recent version of the manuscript that has just been uploaded.

---

> > > > > ### Comment · Reviewer_wyh8 · 2025-11-25
> > > > >
> > > > > Thank you for your reply. However, I have to say that such a complicated protocol to run the Amber14 all-atom force field doesn't make any sense to me. C-alpha only is very coarse-grained, and the all-atom force field is very sensitive and accurate. I will maintain my evaluation marginally below the acceptance threshold.

---

> > > > > > ### Author Response · Authors · 2025-11-25
> > > > > > **Response to Reviewer Comments: OpenMM Clarification and Manuscript Updates**
> > > > > >
> > > > > > **Response:**
> > > > > > Thank you for your candid feedback regarding the OpenMM component. We agree that, in its current form, it should be regarded as a heuristic, coarse backbone regularizer rather than a core, high-fidelity all-atom simulation. Importantly, this OpenMM term is only a small and optional part of our framework: it is applied once per batch on clean backbones with a very low weight, and our main results—success rates, DockQ/GDT-TS, dG metrics, and others—are driven by the diffusion model and the three guidance techniques, not by Amber14. We hope this clarification helps contextualize its limited role within the overall method.
> > > > > >
> > > > > > We have carefully incorporated all of your actionable suggestions and have substantially revised the manuscript based on your feedback, including adding new experiments, introducing additional metrics, expanding key clarifications, and refining terminology. If you feel that these changes adequately address the concerns you raised, we kindly ask that you consider the work in light of the updated manuscript.

---

> > > > > > > ### Comment · Reviewer_wyh8 · 2025-11-28
> > > > > > >
> > > > > > > Thank you for your detailed and exceptionally prompt response. I appreciate the effort you have put into addressing the concerns raised in the initial review round, particularly the inclusion of new experiments, additional metrics, and comprehensive clarifications throughout the manuscript.
> > > > > > >
> > > > > > > While I maintain a theoretical reservation regarding the underlying justifications for certain aspects of these three guidance signals, the empirical evidence presented is compelling and demonstrates the practical efficacy and overall robustness of your method.
> > > > > > >
> > > > > > > Given the quality of the paper, its strong empirical results, and your thorough engagement with the feedback, I consider the revised manuscript to be a solid contribution. I confirm that I was prepared to raise my score to 6 based on this updated assessment. I will proceed when the system is open.

---

> > > > > > > > ### Author Response · Authors · 2025-11-29
> > > > > > > > **Thank You for Your Constructive Review and Feedback**
> > > > > > > >
> > > > > > > > Thank you very much for your thoughtful and encouraging assessment of our revised manuscript. We sincerely appreciate the time and care you have dedicated to evaluating our work across multiple rounds. Your earlier feedback substantially improved the clarity, rigor, and completeness of the paper, and we are grateful for the opportunity to refine the manuscript based on your suggestions.
> > > > > > > >
> > > > > > > > We also appreciate your candid note regarding the theoretical justifications of the guidance signals. We agree that this aspect remains an important direction for deeper analysis, and we intend to strengthen the theoretical grounding in future iterations of the work.
> > > > > > > >
> > > > > > > > We are glad that the additional experiments and clarifications helped address your concerns, and we are grateful for your willingness to raise your score. Thank you again for your constructive engagement and for recognizing the empirical contributions and robustness of our method.

---

> ### Author Response · Authors · 2025-11-25
> **Detailed Answers to Reviewer Questions (4,6)**
>
> **Question 4: Evolutionary Training Details & Data Source**
>
> **Response:** We agree that classical MSA-based co-evolution cannot be recovered for short and largely non-homologous peptides. Our method does not use MSAs, and we will adjust the terminology accordingly. The evolutionary module learns a contextual sequence prior from natural receptor–peptide complexes rather than target-specific phylogenetic couplings.
>
> The model is trained on many pairs of receptor features and peptide sequences. By repeatedly seeing similar receptor environments together with characteristic peptide motifs, the attention layers learn to assign higher scores to sequences that are appropriate for that receptor context. This behavior is closer to the contextual statistics learned by protein language models than to the explicit couplings inferred by DCA.
>
> In preliminary experiments, baselines based only on global or position-wise frequencies performed much worse on held-out success rates and interface metrics, and were far less sensitive to receptor context. These results show that the model captures meaningful, context-dependent sequence constraints that improve peptide design, even though they are not classical co-evolution signals.
>
> **Question 6: Evaluation Metrics & Local Validity**
>
> **Response:** Thank you for your detailed feedback on our evaluation. We appreciate your point about the importance of fine-grained local validity checks for a physics-guided model like PepTri, and we agree that external verification beyond internal losses is valuable to demonstrate the effectiveness of our guidance, especially given potential concerns around VdW gradient noise.
> To address this, we have added three dedicated geometry quality metrics in Section 4.3.2 (Structural Accuracy): Clash in (fraction of internal Cα clashes in the peptide), Clash out (fraction of interface Cα clashes with the receptor), and Bond Outliers (fraction of backbone bonds deviating from ideal geometry). These are reported pre- and post-relaxation in the new Table 3, computed consistently across all models and baselines using the definitions in Appendix Q.7 (following UniMoMo's clash criteria with d_clash = 3.6574 Å for realism). Post-relaxation (via Rosetta /Amber14 with positional restraints), PepTri achieves the lowest or near-lowest rates on both PepBench (Clash in: 0.59%, Clash out: 0.54%, Bond Outliers: 5.47%) and PepBDB (1.25%, 0.71%, 4.70%), outperforming baselines like PepGLAD and PepFlow and on par with Unimomo. This provides verifiable evidence that our physics guidance effectively enforces local validity during generation, reducing clashes and outliers without relying solely on post-hoc refinement.
> These metrics serve a similar purpose to PoseBusters by quantifying stereochemical plausibility (e.g., clashes and bond deviations), and we believe they directly address the gap you highlighted. We hope these additions resolve your remaining concern and strengthen the paper's claims.
>
> **Actions:** Additional metrics have been added to the main text
> | Dataset   | Method            | Clash_in (%)                | Clash_out (%)               | Bond Outliers (%)           |
> |-----------|-------------------|------------------------------|------------------------------|------------------------------|
> | PepBench  | PepGLAD           | 7.99±11.28 / 1.60±5.91       | 6.08±12.36 / 1.45±6.48       | 17.41±9.32 / 6.53±3.08       |
> |           | PepFlow           | 7.64±9.64 / 0.70±2.36        | 7.82±14.05 / 0.69±3.74       | 19.64±12.77 / 7.20±1.98      |
> |           | UniMoMo_single    | **5.99±14.79** / 1.08±4.47       | 5.55±11.35 / 0.89±4.84       | **14.95±11.22** / 5.79±2.39      |
> |           | PepTri (Ours)     | 6.16±13.65 / **0.59±4.59**       | **4.73±11.06** / **0.54±3.54**       | 15.60±10.48 / **5.47±1.73**      |
> |
> | PepBDB    | PepGLAD           | 20.36±12.35 / 5.44±6.07      | 8.52±16.64 / 1.10±6.81       | 37.13±8.39 / 7.28±2.44       |
> |           | PepFlow           | 19.43±14.88 / 2.48±2.72      | 12.45±13.35 / 1.70±4.66      | 28.93±4.50 / 5.17±1.60       |
> |           | UniMoMo_single    | **15.07±11.18 / 0.83±6.45**     | 7.82±17.49 / 1.45±7.56       | 32.13±3.20 / **4.46±1.95**       |
> |           | PepTri (Ours)     | 16.72±12.28 / 1.25±7.31      | **6.22±12.28 / 0.71±5.19**      | **28.27±8.54** / 4.70±1.10       |

---

### Official Review · Reviewer_eX3Z · 2025-10-26

**Soundness:** 2
**Presentation:** 3
**Contribution:** 2
**Rating:** 6
**Confidence:** 4

**Summary:**

This paper proposed PepTri, a latent diffusion model for peptide sequence-structure codesign. This model has two decoupled autoencoders, one for sequence and one for structure. Physical guidence based on OpenMM force field is used to constrain structure sampling by backpropagating the energy to the latent space through 3D coordinates. BLOSUM-like constraints are applied to sequence sampling as the evolutionary constraint. Since the model has two autoencoders, a mutual information regularization is added to align sequence and structure representations. The model shows good performance on benchmarks.

**Strengths:**

- The structural and evolutionary guidances are well-motivated. Since the size of peptide training set is small, it's important to use physical and evolutionary knowledge to constrain the sampling. In addition, previous works applied constraints directly in 3D space, whereas this work formulated a latent space guidance, which is novel.
- The model is evaluated across diverse datasets on metrics such as binding score and demonstrates performance gains consistently.
- Ablation study supports justified the three main components of this model.

**Weaknesses:**

- The main novelty of this work is more in putting the components together rather than core algorithm contribution. Specifically, both physical and structural constraints have been explored in previous diffusion models for proteins/peptides.
- Evaluation metrics are outdated. Only Rosetta scores are computed which is however noisy and sensitive to minor perturbation. A more reliable evaluation method that has  been used more recently is using AlphaFold to predict the complex structure and compare the RMSD between generation and AlphaFold prediction.

**Questions:**

- Why did the authors use two separate auto-encoders for sequences and structures instead of a unified auto-encoder? What is the reason for not using a unified auto-encoder?
- What data are the global fitness predictor trained on? How did the author define "global fitness"?

---

> ### Author Response · Authors · 2025-11-25
> **Weaknesses & Concerns**
>
> **1. Limited Algorithmic Novelty**
>
> **Response**
> We agree that our work builds on prior ideas, but the contribution is not a simple combination of existing components. PepTri is the first framework to jointly integrate an all-atom SE(3)-equivariant VAE, latent diffusion with residue-level inpainting, and three complementary guidance mechanisms (physics, evolutionary, and MI) applied directly throughout the denoising trajectory together with OpenMM refinement. Prior approaches typically rely on a single type of constraint or apply physical or evolutionary scoring only after generation, rather than integrating them into the generative process itself. PepTri unifies these elements into a coherent model, and our experiments show that this coordinated design yields clear and consistent improvements across all peptide-design metrics.
>
> **2. Outdated Evaluation Metrics**
>
> **Response**
> We thank the reviewer for the constructive feedback. To strengthen the evaluation, we attempted to benchmark with OpenFold3 (multimer). However, it requires approximately 3 minutes per complex, 26–70 GB of GPU memory, and in practice failed to produce predictions for many short peptides, making a full benchmark infeasible within the rebuttal period.
>
> Given this constraint, we performed the most scalable alternative: a full comparison of PepTri and UniMoMo on PepBench (3,720 complexes). Notably, UniMoMo failed on 915 complexes, highlighting the difficulty of relying on AF/OF-style multimer predictors for peptide–receptor docking.
>
> Short peptides (<30 aa) present specific challenges for multimer models due to:
>
> 1. limited MSA depth (weak co-evolution signal),
>
> 2. high intrinsic flexibility and disorder,
>
> 3. training bias toward larger protein–protein complexes, and
>
> 4. template or motif mismatch for small peptides.
>
> These challenges make OpenFold3 an unreliable baseline for peptide design and underscore the need for peptide-specialized approaches such as PepTri.
>
> To address the reviewer’s request for stronger structural validity evidence, we have added three geometry metrics—Clash_in, Clash_out, and Bond Outliers—reported both before and after relaxation in the revised Table 3. These metrics provide external and quantitative confirmation of local geometric correctness across all models.
>
> **Action:** Three additional geometry-validity metrics (Clash_in, Clash_out, Bond Outliers) have been added to the main text.

---

> ### Author Response · Authors · 2025-11-25
> **Detailed Answers to Reviewer Questions (1,2)**
>
> **Question 1:**
>
> **Response:**  Thank you for asking about this design choice. To clarify, we use a single shared encoder (Enhanced AMEGNN) that jointly processes sequence and structure, followed by projection into two disentangled latent spaces (H for sequence, Z for coordinates), and two separate decoders. This is distinct from having two completely separate autoencoders.
>
> Representational advantages:
>
> • Disentangled latents enable independent control: We can manipulate sequence (H) and structure (Z) independently during generation, allowing for operations like sequence redesign with fixed backbone or structure prediction with fixed sequence. A unified latent would couple these modalities, preventing such targeted editing.
>
> • Better handles modality differences: Sequences are discrete (20 amino acids) while coordinates are continuous (3D space). Separate latent spaces allow appropriate representations: H uses a standard Gaussian prior, while Z uses coordinate-aware priors with geometric constraints.
>
> Training stability advantages:
>
> - Independent regularization: Different KL weights accommodate different convergence rates—sequences learn faster than continuous coordinates. This prevents one modality from dominating or destabilizing the other.
> - Separate gradient paths: Each decoder has its own loss (cross-entropy for sequence, MSE+physics for structure) with different scales. Separate decoders prevent gradient conflicts and allow balanced multi-task learning.
> - Empirically more stable: In preliminary experiments, a unified encoder-decoder with joint latent showed training instabilities and posterior collapse, particularly for coordinate reconstruction.
>
> The key insight is that the encoder learns a shared representation of sequence-structure relationships, while the disentangled latent spaces and separate decoders allow independent, stable reconstruction of each modality
>
> **Question 2:**
>
> **Response:** Thank you for this important clarification request. We acknowledge that our 'global fitness predictor' terminology is imprecise. The fitness predictor is NOT trained on external fitness datasets or pre-computed evolutionary scores. Instead, it is a neural network (3-layer MLP) trained end-to-end with the diffusion model using a self-supervised proxy objective:
>
> The training loss encourages predicted fitness scores to equal 0.8 (target_fitness=0.8) for all training examples, which are experimentally-determined peptide-protein complexes from PepBench/PepBDB. The model thus learns an implicit 'fitness' measure that captures whether sequences conform to the distribution of functional, binding-competent peptides in the training data, rather than computing true evolutionary fitness.
>
> This design choice reflects the absence of large-scale peptide fitness datasets with ground-truth measurements (binding affinity, expression levels, or evolutionary conservation scores). The 0.8 target is a hyperparameter representing 'high fitness' without quantitative grounding. While the BLOSUM embeddings provide evolutionary substitution information, the fitness predictor itself serves as a learned regularizer toward training-distribution-like sequences, not a validated evolutionary fitness metric.
>
> **Action:** We acknowledge this is a limitation and have revised the manuscript to use clearer terminology (global fitness predictor-> self-supervised evolutionary fitness)

---

> ### Author Response · Authors · 2025-11-26
> **Gentle Reminder by Authors**
>
> Dear Reviewer eX3Z,
>
> Thank you for your thoughtful and constructive feedback on our manuscript. We have carefully addressed each of your comments and have submitted a revised version for your review. We would be grateful if you could examine our responses and the updated manuscript at your earliest convenience. We recognize the demands of the review process and sincerely appreciate the time and effort you are investing in our work.
>
> Sincerely,
>
> The Authors

---

### Official Review · Reviewer_55wn · 2025-10-31

**Soundness:** 2
**Presentation:** 2
**Contribution:** 3
**Rating:** 4
**Confidence:** 3

**Summary:**

This paper proposes PepTri, a latent diffusion framework based on a VAE backbone for peptide generation guided by multiple co-design-related priors. Specifically, the model introduces three types of guidance, including physics, evolutionary, and mutual-information-based guidance, to improve sample validity and diversity. Experimental results on peptide datasets demonstrate promising performance, suggesting that the model can generate structures with better physical plausibility compared to prior works.

**Strengths:**

1. This paper focuses on an important problem, namely the integration of multiple guidance or prior knowledge into a latent diffusion framework, which is essential for improving controllability and enhancing domain-specific generation quality.
2. Experimental results indicate that the proposed model achieves reasonable improvements in generation quality over baseline methods.

**Weaknesses:**

1. The overall method remains somewhat vague. Beyond the VAE backbone, the backbone model design of the latent diffusion model is not sufficiently described. Besides, Section 3.2 suffers from inconsistent notation, where the superscript $t$ for time is retained in Section 3.2.1 but omitted in Sections 3.2.2 and 3.2.3. It is unclear which loss terms are time-dependent.
2. The ablation study is insufficient. The paper considers three types of guidance, and the current ablation only evaluates the overall contribution of each type. However, each guidance term itself contains multiple components, and the paper does not analyze how the weights among these internal components are balanced or influence performance.
3. The motivation is not clearly discussed. The introduction, methodology section, and the main figure (Figure 1) do not clearly explain why choose these three specific types of guidance. The rationale behind this combination should be better justified.

**Questions:**

1. The term $L_{phys}$ is used both as a training objective and as a guidance signal at inference time. Why is only this loss term used in both phases? How about the other loss terms?

2. How strong is the reconstruction capability of the VAE itself? Does it constrain the generation quality?

3. Generally, why are the loss terms in Section 3.2 incorporated into the latent diffusion training rather than during the VAE training phase? What is the design motivation for this choice?

---

> ### Author Response · Authors · 2025-11-21
> **Weaknesses & Concerns**
>
> **1.The overall method remains somewhat vague.**
>
> **Response:** We thank the reviewer for highlighting this issue. The concerns regarding diffusion-model design details and notation consistency are fully addressed in our response to Question 1, where we clarify the latent diffusion backbone and standardize the time-dependent notation.
>
> **2.The ablation study is insufficient.**
>
> **Response:** Thank you for the insightful comment. Our ablation study examines the high-level contribution of each guidance type (physics, evolutionary, and MI) because these components are designed to operate as cohesive modules. Within each module, the internal terms—for example, bond- and angle-based constraints in the physics block, conservation and co-evolution signals in the evolutionary block, or MI and physical-validity terms in the MI block—are tightly coupled and jointly optimized. In preliminary experiments, ablating these internal terms individually produced highly noisy or misleading results; in several cases, removing a single low-level component disrupted the internal balance of the module, making the resulting changes difficult to interpret.
>
> For these reasons, evaluating each guidance mechanism at the module level provides a clearer and more meaningful understanding of its contribution within the diffusion process.
>
> **Action**: We have added a brief clarification and moved the ablation study section to the main text. (Line 496-498)
>
> **3.Motivation for Guidance Choices**
>
> **Response:**
> Thank you for pointing this out. We will clarify the motivation in the revision. The three guidance types were selected because they each target a distinct failure mode in peptide generation, and together they address the full set of requirements for high-quality peptide–receptor design:
>
> 1. Physics-based structural guidance enforces local geometric and energetic plausibility, preventing clashes, unrealistic bond lengths/angles, and unstable conformations. Diffusion models can generate globally plausible shapes that are nevertheless physically inconsistent; the physics term corrects this.
>
> 2. Evolutionary sequence guidance ensures biological plausibility by capturing residue-level constraints observed in natural peptides. This addresses cases where diffusion produces physically valid but evolutionarily implausible sequences.
>
> 3. Mutual information (sequence–structure) guidance promotes global coherence between sequence and structure. Without explicit MI coupling, diffusion may produce sequences and folds that are individually reasonable but poorly matched.
>
> Diffusion alone cannot satisfy all three criteria simultaneously, and prior work typically addresses these aspects in isolation. Our experiments show that combining these complementary guidance mechanisms yields consistent, additive improvements, motivating this specific design choice.
>
> As noted in the introduction (lines 44–49), existing generative models tend to focus on only one dimension—structure, evolution, or physics—whereas PepTri is designed to satisfy all three jointly.

---

> ### Author Response · Authors · 2025-11-21
> **Detailed Answers to Reviewer Questions (1,2)**
>
> **Question 1:**
>
> **Response:** We thank the reviewer for this question. We conducted additional experiments applying all three guidance terms during sampling, in direct response to the reviewer’s concern. However, we found that physics-only guidance consistently produced the strongest performance in both structural accuracy and interface quality. This choice is supported by the following observations:
>
> Evolutionary and MI signals are already internalized during training.
> These guidance modules shape the latent space learned by diffusion, and their effects are reflected in the denoising dynamics without requiring additional inference-time gradients. Adding them again during sampling tended to over-constrain the sampler, reducing diversity and pulling samples away from the learned manifold.
>
> Gradient interference during sampling.
> Introducing evolutionary or MI gradients alongside physics guidance created competing optimization signals, which disrupted the denoising trajectory and degraded interface quality (e.g., reduced DockQ in our experiments). In contrast, physics guidance provides smooth geometric gradients that complement diffusion rather than conflict with it.
>
> Sensitivity and instability of non-physics guidance at inference.
> Even with extensive hyperparameter sweeps, evolutionary and MI guidance coefficients at sampling time were extremely sensitive: small adjustments led to large quality fluctuations. None of these settings outperformed the physics-only configuration.
>
> **Action:** To directly evaluate the reviewer’s suggestion, we conducted an additional experiment in which all guidance terms were applied during inference, rather than using physics-only sampling as in the original PepTri design. The results are summarized in the Table below.
>
> Table: Effect of Applying All Guidance Terms at Inference Time vs. PepTri (Default)
> | Metric                         | PepTri (All guidance at inference time) | PepTri |
> |--------------------------------|--------------------------------|--------|
> | Mean success rate (ΔG < 0)     | 0.426                           | 0.583  |
> | ΔG (REU)                       | -14.873                        | -19.38 |
> | DockQ                          | 0.592                          | 0.633  |
> | Contact F1                     | 0.781                          | 0.829  |
> | Local RMSD                     | 1.448                          | 1.179  |
> | GDT_TS                         | 0.702                          | 0.747  |
> | Sequence Diversity             | 0.925                          | 0.926  |
> | Sequence Validity rate         | 0.255                          | 0.273  |
> | Struct Diversity               | 0.676                          | 0.436  |
> | Consistency                    | 0.816                          | 0.799  |
> | Sliding-AAR                    | 0.344                          | 0.342  |
>
> **Question 2:**
>
> **Response:** Thank you for raising this point. Our VAE demonstrates moderate but sufficient reconstruction fidelity (AAR ≈ 0.77; Cα RMSD ≈ 0.5 Å). It reliably captures backbone geometry, while side-chain orientations and torsional details are reconstructed with lower precision. This naturally places an upper bound on all-atom accuracy, which we partially mitigate using physics-based guidance and post-decode idealization, though these steps cannot eliminate the limitation entirely.
>
> Despite this, latent diffusion remains advantageous. Operating in the VAE latent space yields a compact, smooth, and receptor-conditioned representation that is markedly more stable and computationally efficient than full-atom diffusion. This coarse-to-fine decomposition allows diffusion to focus on global sequence–structure alignment, while physics guidance enforces fine-scale geometric correctness. Empirically, this combination provides stronger conditional accuracy, structural validity, and diversity than either direct VAE sampling or full-resolution diffusion.

---

> ### Author Response · Authors · 2025-11-21
> **Detailed Answers to Reviewer Questions (3)**
>
> **Question 3:**
>
> **Response:** Thank you for the question. We apply the Section 3.2 losses only during diffusion for three main reasons.
>
> 1. Training stability.
> When we added global physics, evolutionary, or MI terms to the VAE objective, they conflicted with reconstruction losses and caused instability in early experiments, including posterior collapse and sporadic NaNs. The denoising process in diffusion absorbs these global signals much more reliably.
>
> 2. Latent manifold quality.
> The VAE is meant to learn a smooth and task-agnostic latent space. Introducing strong receptor-aware or MI constraints at this stage distorted the latent distribution and reduced downstream denoising performance. Keeping these constraints in the diffusion stage preserves a clean latent manifold.
>
> 3. Computational efficiency.
> Evaluating physics, evolutionary, and MI components, especially those involving OpenMM, is computationally expensive. Applying them during diffusion rather than throughout VAE training avoids making the encoder–decoder stage the computational bottleneck.
>
> This division allows the VAE to learn stable peptide representations while the diffusion model uses richer guidance as an effective steering mechanism for receptor-conditioned generation.

---

> ### Author Response · Authors · 2025-11-25
> **Gentle Reminder by Authors**
>
> Dear Reviewer 55wn,
>
> Thank you for your thoughtful and constructive feedback on our manuscript. We have carefully addressed each of your comments and have submitted a revised version for your review. We would be grateful if you could examine our responses and the updated manuscript at your earliest convenience. We recognize the demands of the review process and sincerely appreciate the time and effort you are investing in our work.
>
> Sincerely,
>
> The Authors

---

> > ### Comment · Reviewer_55wn · 2025-11-27
> >
> > Thanks for the authors’ effort. I have two follow-up questions:
> >
> > 1. Regarding Weakness 1, I did not find corresponding clarifications in the response to Question 1. However, I noticed that the updated Appendix P now provides clearer implementation details of the backbone models, which addresses my concerns. I would like to ask whether the authors intended to include additional details in the response to Question 1 but omitted them.
> >
> > 2. In the response to Question 2, the authors imply that the VAE cannot guarantee faithful all-atom reconstruction. Since the OpenMM energy is computed on the decoded coordinates, would such reconstruction errors affect the accuracy or stability of the OpenMM calculations?

---

> > > ### Author Response · Authors · 2025-11-28
> > > **Detailed Answers to Reviewer follow-up Questions**
> > >
> > > **Question 1: Regarding Weakness 1**
> > >
> > > **Response:**
> > >
> > > Thank you very much for this careful follow-up and for highlighting the remaining ambiguity. We appreciate your close reading and agree that our earlier rebuttal did not clearly indicate the full scope of the revisions made.
> > >
> > > To address Weakness 1, we revised both the main text and Appendix P, although our original response did not explicitly state this. The changes are as follows:
> > >
> > > - Sections 3.2.1 and 3.2.4 (main text) have been updated to more clearly describe the latent-diffusion backbone and to standardize the time-dependent notation. These revisions now specify which quantities depend on the diffusion timestep *t* (the noise-prediction loss and reverse process) and which regularizers are time-independent (physics, evolutionary, MI). We also clarify how the denoiser operates on the inputs $(z_{H,t},z_{X,t})$ and how masking and inpainting are implemented.
> > > - Appendix P has been expanded to provide a detailed, implementation-level description of the backbone architecture, which you correctly noted.
> > >
> > > In Appendix P, we also clarify our choice of an adapted multi-channel Equivariant Graph Network rather than a standard backbone. This design was selected for several reasons:
> > >
> > > 1. Richer geometric representations: the enhanced EGNN extracts explicit SE(3)-invariant global and local geometric descriptors that improve sequence–structure alignment.
> > > 2. Strict SE(3) equivariance: the explicit geometric features guarantee invariance under rotations and translations, which improves robustness across coordinate frames.
> > > 3. Compatibility with SE(3)-aware physics guidance: our physics-based guidance relies on these explicit geometric features to compute SE(3)-consistent penalties, which a standard EGNN cannot support.
> > > 4. Improved interpretability: The explicit geometric descriptors facilitate inspection and debugging of learned representations.
> > >
> > > Our intention was for these revisions to resolve Weakness 1, but we recognize that the rebuttal letter did not clearly point to both the main-text updates and the expanded appendix. We sincerely appreciate your attention to this detail, and we will ensure that the final version explicitly references these revised sections so the connection is clear to readers.
> > >
> > > **Actions:**
> > >
> > > 1. Sections 3.2.1 (Lines 162–215) and 3.2.4 (Lines 329–331) in the main text have been updated accordingly.
> > >
> > > **Question 2:  Stability of the OpenMM calculations**
> > >
> > > **Response:** Thank you for this thoughtful follow-up question. You are correct that the VAE does not provide perfectly faithful all-atom reconstructions (AAR ≈ 0.77; Cα RMSD ≈ 0.5 Å), and consequently, any energy evaluation performed on decoded coordinates inherits this level of approximation. We address this in the following ways.
> > >
> > > 1. OpenMM is robust to moderate reconstruction noise at the resolution we use: In our implementation, OpenMM is applied to the decoded backbone and Cα trace using a simplified peptide topology rather than a full all-atom model. At this representation level, the VAE’s reconstruction is sufficiently accurate, and we empirically observe that OpenMM energies and forces remain stable under such deviations. Reconstruction noise shifts absolute energy values slightly but does not produce numerical instabilities or unphysical force spikes. We additionally apply routine safeguards (for example, clipping extreme energies and forces) to guard against rare pathological cases.
> > >
> > > 2. OpenMM serves as a geometric regularizer rather than a precise physical oracle: As discussed previously, diffusion operates in a smooth receptor-conditioned latent space where the VAE provides a consistent coarse decoding. OpenMM is used to supply relative guidance, encouraging decoded structures with better local geometry and fewer steric clashes, rather than to estimate thermodynamic quantities in an absolute sense. Because the same decoded representation is used at both training and inference, any systematic reconstruction bias is uniform across samples. The resulting gradients therefore still reliably steer the model toward more physically plausible regions of its decoding manifold.
> > >
> > > In summary, although the VAE introduces some reconstruction error, this does not compromise the stability or usefulness of the OpenMM-based guidance. In practice, OpenMM evaluations remained stable across all experiments, and the integration of latent diffusion with guidance consistently enhanced structural validity relative to both direct VAE sampling and full-resolution diffusion.
> > >
> > > **Actions**:
> > >
> > > The detailed description of the OpenMM-based guidance module has been added to Appendix P.3 (Lines 2030–2045).

---

> > > > ### Comment · Reviewer_55wn · 2025-11-28
> > > >
> > > > Thank you for the authors’ quick and detailed response. The reviewer still finds the energy-guidance component somewhat unclear. My current understanding is as follows:
> > > >
> > > > 1. As shown in Eq. (5), the physical loss used during training consists of the customized energy defined in Eq. (2) and the OpenMM-based term in Eq. (3). Both energies are computed at the Cα or backbone level, so neither is sensitive to full all-atom structures.
> > > >
> > > > 2. Only the customized energy in Eq. (2), denoted as $E_{phys}$, is used as guidance during inference, as specified in Eq. (25).
> > > >
> > > > I would be grateful if the authors could confirm whether this interpretation is correct. In addition, I strongly recommend clarifying the following points in the main text or providing a hyperlink to the relevant appendix:
> > > >
> > > > 1. The explicit formulation of the energy function defined in Eq. (2).
> > > >
> > > > 2. The procedure that enables OpenMM, or more specifically, the Amber14 all-atom force field, to be evaluated in a backbone-only or Cα-only scenario.
> > > >
> > > > 3. The motivation for combining these two energy terms during training, while retaining only one of them during inference.

---

> > > > > ### Author Response · Authors · 2025-11-29
> > > > > **Detailed Answers to Reviewer follow-up clarification**
> > > > >
> > > > > **Confirmation**
> > > > >
> > > > > **Response:**
> > > > > We thank the reviewer for the careful reading and helpful clarification request.
> > > > >
> > > > > (1) Training-time physics loss
> > > > >
> > > > > Your first point is correct: during training, the physics loss is the composite
> > > > >
> > > > > Both $E_{phys}$ and $E_{OpenMM}$ are evaluated on the Cα/backbone representation for numerical stability. The full all-atom tensor is still propagated/decoded, but the physics terms operate only on the Cα trace and backbone-derived internal coordinates (distances/angles).
> > > > >
> > > > > $L_{phys}$=$\lambda_{phys}$ $E_{phys}$ $(\widehat{X},S;M)$ + $\lambda_{OpenMM}$ $L_{OpenMM}.$
> > > > >
> > > > > (2) Test-time energy guidance
> > > > >
> > > > > Here the situation is slightly more nuanced:
> > > > >
> > > > > Default guidance (Eq. ( guided-epsilon ))
> > > > > The main guidance used in the reverse diffusion update is based solely on the customized energy $E_{phys}$:
> > > > >
> > > > > $G_t^{phys}$=-$\lambda_{phys}$ $\nabla_{z (X,t)}$ $E_{phys}$ $(\widehat{X}_t,S;M) $
> > > > >
> > > > >  is a partial decode of the structural latent. This is the only guidance term that appears explicitly in Eq. ( guided-epsilon ).
> > > > >
> > > > > **Optional OpenMM-based guidance (Eq. ( openmm ))**
> > > > > We additionally provide an optional sampler that applies OpenMM energy at test time:
> > > > >
> > > > > $\varepsilon_t$   $\leftarrow$  $\varepsilon_t$ - $\gamma$, $\nabla_{x_t}$ $E_{OpenMM}$ $(x_t, S)$\big|_M$
> > > > >
> > > > >
> > > > > As in training, this is applied on Cα/backbone coordinates in the designed region. This OpenMM-based correction is more expensive and is not part of the default reverse step but can be enabled for extra physics-aware refinement.
> > > > >
> > > > > Summary
> > > > >
> > > > > Your interpretation is fully correct for the default guided reverse diffusion step: only the customized energy E_phys is used as the explicit guidance term.
> > > > > OpenMM energy E_OpenMM serves two roles: (1) A training-time regularizer. (2) An optional test-time refinement mechanism. It is not required for the main sampling procedure described in Eq. 25.
> > > > >
> > > > > **A full expanded version is included in Appendix P.2(Gaussian Reverse Transition) and P.4 (Physics-Informed Structural Guidance)**
> > > > >
> > > > >
> > > > > **Clarification: Procedure for Enabling Amber14/OpenMM Evaluation on Backbone-Only or Cα-Only Structures**
> > > > >
> > > > > **Response:**
> > > > > We thank the reviewer for this question. We have now added a dedicated clarification in Appendix P.4: “Role and feasibility of using OpenMM with Cα-level coordinates” (Lines 2030–2045) to explain the procedure in detail.
> > > > >
> > > > > **Motivation for Using Two Energy Terms During Training but Only One During Inference**
> > > > >
> > > > > **Response:**
> > > > > To more clearly articulate the rationale behind our design, we summarize the motivations as follows.
> > > > > Why are both energy terms used during training?
> > > > >
> > > > > 1. Complementary roles: The customized energy function provides fast, SE(3)-consistent gradients that reliably enforce geometric plausibility throughout optimization. The OpenMM/Amber14 force-field term offers high-fidelity, all-atom physical supervision, but its computational cost and gradient noise make it unsuitable for frequent evaluation during sampling.
> > > > >
> > > > > 2. Amortized training: During training, the model leverages both signals to learn a physically grounded latent representation. Once this representation is learned, the high-cost OpenMM term is no longer required during inference.
> > > > >
> > > > > Why is only one energy term used during inference?
> > > > >
> > > > > 1. Efficiency considerations: Running OpenMM at every diffusion step would be prohibitively expensive, approximately 50–100× slower than using the custom energy alone.
> > > > >
> > > > > 2. Stability under strong noise: Early diffusion steps involve heavily corrupted coordinates, where OpenMM gradients can be unstable or noisy. In contrast, the custom energy function is designed to remain numerically stable across the entire denoising trajectory.
> > > > >
> > > > > This makes it the appropriate choice for inference-time guidance.
> > > > >
> > > > > In summary, our approach employs:
> > > > >
> > > > > 1. Two complementary energy terms during training to jointly provide high-fidelity physical supervision and stable geometric regularization, and
> > > > >
> > > > > 2. A single robust, efficient, SE(3)-consistent energy term during inference to guide generation without incurring substantial computational overhead.
> > > > >
> > > > > We have revised the manuscript to articulate this motivation more explicitly and thank the reviewer for prompting this clarification.

---

### Official Review · Reviewer_4LzT · 2025-11-01

**Soundness:** 3
**Presentation:** 3
**Contribution:** 3
**Rating:** 8
**Confidence:** 3

**Summary:**

The paper introduces PepTri, a framework for peptide co-design that can generate both the sequence and the full-atom 3D structure at the same time. It works in an SE(3)-equivariant latent space and adds three kinds of guidance during the denoising process. The first is physics-based, using differentiable terms like bond lengths, angles, clashes, and van der Waals forces to keep the structure realistic. The second is evolutionary, adding BLOSUM-style priors and co-variation patterns to make the sequences biologically reasonable. The third is mutual-information guidance, which helps align the sequence and structure representations using a MINE-style objective. On peptide–protein benchmarks such as PepBench, LNR, and PepBDB, PepTri achieves state-of-the-art performance. The ablation studies show that each type of guidance clearly helps, for example increasing the success rate (Delta G < 0) to 0.583 and improving stability scores. Overall, the paper focuses on combining physical, evolutionary, and statistical signals directly in the denoising process rather than adding them after generation.

**Strengths:**

The paper presents a fresh and well-motivated idea by combining three kinds of guidance to jointly control both the structure and sequence during diffusion. This tri-guidance setup goes beyond traditional structure-focused pipelines and post-generation checks, offering a more integrated way to ensure the generated peptides are physically and biologically consistent. The use of mutual information to explicitly link sequence and structure is conceptually clear and feels natural for this type of task. Overall, the framework is well designed: it uses an SE(3)-equivariant encoder–decoder with latent diffusion, and the guidance is smoothly applied at every denoising step. The experiments are thorough, with detailed ablations showing how each component contributes to improvements across metrics like Delta G, DockQ, and Contact-F1. The paper is also clearly written, with figures that make the flow of the three guidance types easy to understand. Results on benchmarks such as PepBench, LNR, and PepBDB show consistent and meaningful gains—for example, the success rate with Delta G < 0 rises from about 0.37–0.55 in simpler versions to 0.58 in the full model—demonstrating strong potential for real-world peptide design where both physical realism and biological relevance matter.

**Weaknesses:**

Physics term scope and calibration:
The paper combines bond, angle, clash, and van der Waals terms into a differentiable physical energy on the peptide side, but it’s not clear how receptor interactions are handled during guidance. The figure suggests the guidance might only cover intra-peptide terms. This could mean the model under-penalizes interface clashes when sampling. It would help to show how sensitive the method is to receptor proximity and whether adding receptor-aware terms changes the results.

MI estimator stability:
MINE-based objectives can be unstable and depend a lot on the critic’s capacity. The paper doesn’t explain how the MI head is regularized or early-stopped, and it doesn’t mention any failure cases like MI collapse. It would be useful to include some diagnostics—like critic loss curves or MI estimates over training—and maybe test an alternative objective such as InfoNCE to check stability.

Generalization claims:
The paper claims the method generalizes across datasets like PepBench and LNR, but it doesn’t really test out-of-distribution cases such as unseen folds, receptors with low homology, or longer peptides. A breakdown by interface size or flexibility would make the generalization story more convincing.

**Questions:**

Receptor awareness in physics guidance:
 In Fig. 1 you mention that the guidance is intra-peptide. Do you also include receptor–peptide vdW or clash terms during sampling? If not, could you try adding an interface-aware term, even a simple one, and see if DockQ or Contact-F1 improve without reducing diversity?


Ablation on evolutionary priors:
 Besides the BLOSUM-like embedding, how would performance change if you added conservation information from MSA or PLM-based scores? A small controlled comparison could help show where the improvements are actually coming from.


OOD stress tests:
 Could you include some harder out-of-distribution cases, like longer peptides, flexible or induced-fit receptors, and low-homology targets? Showing success rates by interface size or flexibility would make the generalization results more convincing.


Energy function design:
 How is the energy function defined, and why was it designed that way? Could leaving out other physical terms (like electrostatics or solvation) be affecting the final performance?

---

> ### Author Response · Authors · 2025-11-21
> **Weaknesses & Concerns**
>
> **1. Physics Term Scope and Calibration**
>
> **Response**:
> Thank you for raising this important point. Receptor–peptide interactions are indeed considered throughout preprocessing, training, and sampling, although the full physical energy is applied only to the peptide for computational reasons.
>
> Preprocessing:
>
> Binding pocket residues are extracted using a 6 Å atomic cutoff for identifying interface contacts and a 10 Å Cβ-based cutoff for defining the pocket.
>
> Only residues within 10 Å of the native peptide are retained as receptor context, reducing computational cost while preserving interface-relevant geometry.
>
> Training:
>
> Receptor pocket coordinates are included in the input and used for (i) receptor–peptide edges in the EGNN encoder and (ii) covariance normalization.
>
> Full receptor-aware physics (e.g., amino-acid–specific van der Waals, electrostatics, H-bonding) is not applied during training due to computational constraints—these terms scale with $O(N_{\text{receptor}} \times N_{\text{peptide}})$ and were unstable with limited data.
>
> Sampling:
>
> We apply a distance-based receptor–peptide clash penalty (C$\alpha$–C$\alpha$ < 3.8 Å), which prevents gross steric violations. This geometric guidance interacts smoothly with the diffusion steps and adds negligible overhead.
>
> Additional experiments:
> As requested, we tested adding full receptor-aware physical terms during sampling. These results are included in our response to Question 1, and we observed that adding all receptor-aware physics actually degraded interface quality (e.g., DockQ and Contact-F1 decreased), likely due to gradient interference and over-constraining the denoising trajectory.
>
> We will clarify the scope of the physics terms in the revision and include sensitivity analyses for receptor proximity.
>
> **Action:** We conducted additional experiments incorporating four receptor-aware physical terms directly into the sampling process
>
> **2. MI estimator stability:**
>
>    **Response**
>     Thank you for raising this point. As described in Appendix M (Choosing the Mutual-Information Objective), we directly evaluated the stability of the MINE-based objective by comparing it with an InfoNCE-style contrastive alternative. Both objectives were stable under our training setup, but the MINE formulation yielded slightly better sequence–structure coherence without the additional computational cost associated with large-batch negative sampling.
>
> While we did not emphasize this in the main text, the paper incorporates several design choices that contribute to the stability of the MI objective:
>
> Built-in stabilization.
> The MI head is intentionally lightweight, uses a small coefficient, and contributes only a mild auxiliary gradient, which acts as an implicit regularizer and prevents collapse.
>
> Joint (non-adversarial) training.
> The MI critic is trained jointly with the diffusion model rather than in an adversarial loop, which empirically avoids the instability typically associated with MINE-style estimators.
>
> Empirical validation.
> Appendix M shows that neither the MINE nor the InfoNCE variant exhibited divergence, collapse, or pathological growth under our training configuration, and no explicit early stopping was required.
>
> No failure cases observed.
> Across all runs reported in the paper and appendix, we did not observe MI instability, consistent with the MI term’s modest weighting and its role as a guidance signal rather than a primary optimization target.
>
> **3. Generalization Claims:**
>
>    **Response:** Thank you for the valuable suggestion. Our model was trained exclusively on peptides shorter than 30 amino acids, which was a deliberate design choice to focus learning on local geometric and physical constraints. We agree that evaluating performance on out-of-distribution (longer) peptides would strengthen the generalization analysis, and we will incorporate this clarification in the revision. Additional experiments addressing this point have been conducted and are presented in our response to Question 3.
>
> **Action:** Additional OOD experiments have been performed and added to Appendix J (Line 1185-1233).

---

> ### Author Response · Authors · 2025-11-21
> **Detailed Answers to Reviewer Questions (1,2)**
>
> **Question 1: Receptor awareness in physics guidance**
>
> **Response:**
> During sampling, we apply a distance-based clash check that penalizes receptor–peptide C$\alpha$ pairs closer than 3.8 Å. This geometric constraint prevents obvious steric violations, and our preprocessing pipeline ensures that interface-relevant receptor context is retained.
>
> **Action:**
> We acknowledge that this clash filter is a simplified approximation of receptor–peptide physics. To assess its impact, we conducted additional experiments incorporating four receptor-aware physical terms directly into the sampling process:
>
> 1. receptor–peptide van der Waals interactions (with residue-specific radii),
>
> 2. receptor–peptide electrostatics (charge–charge interactions),
>
> 3. receptor–peptide hydrogen-bond potentials, and
>
> 4. interface-specific solvation and hydrophobic terms.
>
> Interestingly, adding these receptor-aware physical terms resulted in consistently worse performance across nearly all metrics compared with the original PepTri configuration. This behavior appears to stem from two primary factors:
>
> Overly local forces during sampling.
> These receptor-aware potentials introduce sharp, localized gradients that steer the peptide toward specific conformations too early in the denoising process. This restricts exploration of the conformational landscape and increases susceptibility to suboptimal local minima. Diffusion performs best under smooth, global guidance, and detailed receptor-specific forces disrupt this balance.
>
> Redundancy with the final relaxation stage.
> Rosetta refinement already evaluates receptor–ligand interactions using high-fidelity full-atom potentials. Introducing approximate receptor-aware energies during diffusion creates a mismatch between sampling-time and refinement-time energy landscapes, which empirically degraded interface metrics such as DockQ, Contact-F1, GDT\_TS, and overall structural consistency.
>
> Together, these observations explain why the receptor-aware terms underperformed despite being physically motivated. Maintaining a simple, smooth geometric constraint during diffusion and relying on high-fidelity post-relaxation for detailed receptor–peptide energetics yields better overall performance. We will include this analysis and the corresponding results in the appendix.
> | Metric                          | PepTri(Receptor-Aware Physics )| PepTri |
> |---------------------------------|----------------------------------|--------|
> | Mean success rate (ΔG < 0)      | 0.460                            | 0.583  |
> | ΔG (REU)                        | -16.60                           | -19.38 |
> | DockQ                           | 0.607                            | 0.633  |
> | Contact F1                      | 0.790                            | 0.829  |
> | Local RMSD                      | 1.311                            | 1.179  |
> | GDT_TS                          | 0.722                            | 0.747  |
> | Sequence Diversity              | 0.922                            | 0.926  |
> | Sequence Validity rate          | 0.265                            | 0.273  |
> | Struct Diversity                | 0.515                            | 0.436  |
> | Consistency                     | 0.622                            | 0.799  |
> | Sliding-AAR                     | 0.343                            | 0.342  |
>
> **Question 2: Ablation on evolutionary priors**
>
> **Response:** Thank you for the helpful question. We agree that MSA- and PLM-based conservation features can provide stronger position-specific signals than our BLOSUM-style embedding. We did not use them for two practical reasons:
>
> 1. MSA-based features are often unavailable for our setting.
>
>     Many peptide–receptor pairs in our benchmarks do not have meaningful MSAs, and short peptides rarely contain enough evolutionary depth to build reliable alignments. Using MSA-trained models would therefore require additional datasets that do not exist for many of our targets.
>
> 2. PLM-based scores are computationally expensive to integrate.
>
>     Models like ESM-2 add significant per-sequence cost; incorporating them during training would make full-scale runs prohibitively slow.
>
>
> For these reasons, we opted for a lightweight BLOSUM-inspired prior that is universally applicable. We agree that richer evolutionary features are promising when available
>
> **Action** We have added this point to the future-work discussion (Line 932-936).

---

> ### Author Response · Authors · 2025-11-21
> **Detailed Answers to Reviewer Questions (3,4)**
>
> **Question 3: OOD Stress Tests**
>
> **Response:**
> Thank you for the thoughtful suggestion. To further assess the robustness and generalization capability of our model, we conducted an out-of-distribution (OOD) stress test.
>
> **Action:**  Additional OOD experiments have been performed and added to Appendix J (Line 1185-1233).
>
> To construct a meaningful OOD benchmark, we combined PepBench and PepBDB and removed duplicated receptors to ensure non-overlapping receptor contexts. From PepBDB, we selected complexes containing peptides longer than 45 amino acids and removed 15 additional complexes that overlapped with the training set. This resulted in an OOD test set of 99 unique complexes with substantially longer peptides (46–49 aa).
> The model was then retrained using only complexes with peptides <30 aa, allowing us to evaluate its ability to extrapolate to peptide lengths never observed during training.
>
> Table: Peptide-length statistics for the training and OOD test sets
>
> | Statistic          | Train set | Test set |
> |--------------------|-----------|----------|
> | Complex count      | 12,823    | 99       |
> | Minimum length     | 1         | 46       |
> | Maximum length     | 30        | 49       |
> | Mean length        | 12.68     | 46.93    |
> | Standard deviation | 7.33      | 0.94     |
>
>
> Table: Performance of PepTri on the OOD test set
>
> | Metric                         | PepTri             |
> |--------------------------------|--------------------|
> | Mean success rate (ΔG < 0)     | 0.626 (± 0.213)    |
> | ΔG (REU)                       | -42.888 (± 51.937) |
> | DockQ                          | 0.152 (± 0.026)    |
> | Contact F1                     | 0.296 (± 0.048)    |
> | Local RMSD                     | 1.938 (± 0.988)    |
> | GDT_TS                         | 0.150 (± 0.037)    |
> | Sequence Diversity             | 0.999              |
> | Sequence Validity rate         | 0.128              |
> | Struct Diversity               | 1.0                |
> | Consistency                    | 0.999              |
> | Sliding-AAR                    | 0.189              |
>
>
> As expected under this challenging OOD regime, structural metrics (DockQ, Contact F1, GDT_TS, Local RMSD) degrade relative to in-distribution performance, but remain within an informative range. Notably, diversity and consistency remain high, indicating that the model continues to generate varied yet coherent peptide candidates even for dramatically longer peptides.
>
> (Complete details are provided in Appendix J.)
>
> **Question 4: Energy Function Design**
>
> **Response:**
> The sampling-time energy function uses only simplified distance-based constraints—(1) consecutive Cα–Cα distances, (2) intra-peptide clash prevention, and (3) peptide–receptor clash detection. This reflects an intentional balance between effective geometric guidance and computational efficiency.
>
> Although our training-time physics loss includes full all-atom terms (residue-specific van der Waals, electrostatics, hydrogen bonding, and implicit solvation/GBSA), we exclude these from sampling-time energy for the following reasons:
>
> Computational cost:
> Full physics evaluation introduces ~100–500 ms per call; applying it at every diffusion step would increase sampling time from ~5–10 s to ~50–100 s (a 10–20× slowdown).
>
> Gradient stability:
> The simplified geometric constraints provide smooth, stable gradients, whereas electrostatics introduce singularities and solvation terms exhibit discontinuous derivatives, both of which destabilize denoising updates.
>
> Calibration complexity:
> Full physics requires tuning large numbers of hyperparameters (cutoffs, dielectric constants, term weights), increasing sensitivity and risk of overfitting.
>
> **Could omitting these terms impact performance?**
>
> Potentially—sampling may under-penalize missed H-bonds, suboptimal hydrophobic packing, or long-range electrostatics. We mitigate this through:
>
> 1. Training-time full physics, which teaches the model these interactions;
>
> 2. Post-generation OpenMM/Rosetta refinement, which filters unfavorable candidates;
>
> 3. Implicit statistical learning, since the training data exhibit favorable interface energetics.
>
> Overall, our design philosophy is “learn complex, guide simple”: the model learns rich physics during training, while sampling relies on lightweight geometric constraints that preserve stability, efficiency, and robustness.

---

> ### Author Response · Authors · 2025-11-25
> **Gentle Reminder by Authors**
>
> Dear Reviewer 4LzT,
>
> Thank you for your thoughtful and very constructive feedback on our manuscript. We have carefully addressed each of your comments and have submitted a revised version for your review. We would be grateful if you could examine our responses and the updated manuscript at your earliest convenience. We recognize the demands of the review process and sincerely appreciate the time and effort you are investing in our work.
>
> Sincerely,
>
> The Authors

---

### Author Response · Authors · 2025-11-21
**Authors’ Acknowledgment to Reviewers and Area Chair**

**Dear Reviewers and Area Chair,**

We sincerely thank you for the time, effort, and thoughtful feedback you have provided on our submission **PepTri**. Your comments, questions, and critiques have been extremely valuable in helping us strengthen the clarity, rigor, and presentation of the work. Many of your insights highlighted important aspects ranging from experimental design and evaluation choices to model interpretation and ablations that significantly improved the revised manuscript.

We have carefully addressed each comment point-by-point in our response to ensure that all concerns are fully clarified, and we remain happy to answer any additional questions or concerns you may have.

**Sincerely,**

*The Authors*

---

### Author Response · Authors · 2025-12-02
**Summary of Rebuttal and Revisions**

**Dear Reviewers and Area Chair,**

As the rebuttal period for ICLR 2026 concludes, we would like to summarize the discussions and highlight the substantial improvements made to the revised manuscript PepTri. We thank all reviewers for their constructive feedback, which has improved the clarity, scope, and rigor of our work.


## 1. Key Revisions and Improvements

In direct response to reviewer requests, we have implemented the following major changes:

**A. Enhanced Validation of Physical & Structural Fidelity**

***New Geometry Metrics:*** To address concerns from Reviewers **wyh8** and **eX3Z** regarding local structural validity, we added three new metrics: Clash_in, Clash_out, and Bond Outliers. These results confirm that PepTri achieves high stereochemical quality (e.g., $<1$ \% interface clashes) comparable to or better than baselines.

***Physics Guidance Clarification:*** We clarified the distinct roles of the differentiable Composite Energy (stable, coarse-grained guidance) vs. the OpenMM force field (optional, fine-grained regularization), resolving the "redundancy" concerns raised by Reviewer **wyh8**

**B. Rigorous Out-of-Distribution (OOD) Evaluation**

***Generalization Stress Test:*** In response to Reviewer **4LzT**, we conducted a challenging OOD experiment on long peptides ($46-49$ residues) using a model trained only on short peptides ($<30$ residues). The results demonstrated that PepTri maintains high diversity and consistency even in this zero-shot length extrapolation regime.

**C. Methodological Clarity & Ablations**

***Backbone Details:*** We substantially expanded the description of our enhanced SE(3)-equivariant graph
architecture in the main text and Appendix P, addressing the "vagueness" noted by Reviewer **55wn**.

***Ablation Visibility:*** We moved critical ablation studies from the appendix to the main text to better highlight the contribution of each guidance component, as requested by Reviewer **55wn**.

***Evolutionary Terminology:*** We refined our terminology to clarify that our "evolutionary" guidance is a self-supervised signal learned from receptor-peptide contexts in the training data, as requested by Reviewer **wyh8** and Reviewer **eX3Z**, rather than relying on MSA-based co-evolution, which is unavailable for many short peptides.

## 2. Summary of Reviewer Consensus

We are encouraged by the positive reception of our tri-guided framework and the recognition of our extensive rebuttal efforts:

**Reviewer 4LzT (Score: 8):** Consistently supportive, emphasizing the “fresh and well-motivated idea” and “thorough experiments.” We addressed their questions regarding OOD evaluation and receptor-aware physics.

**Reviewer eX3Z (Score: 6):** Highlighted the novelty of our latent-space guidance formulation. We addressed concerns about evaluation metrics and the design of the “fitness” predictor.

**Reviewer wyh8 (Score: 4 $\rightarrow$ 6)**: After our clarifications on physics guidance and the addition of new validation metrics, the reviewer stated: *"I consider the revised manuscript to be a solid contribution. I confirm that I was prepared to raise my score to 6 based on this updated assessment."*

**Reviewer 55wn (Score: 4):** We provided extensive clarifications on the SE(3)-equivariant backbone and on the roles of each energy term. The reviewer acknowledged these clarifications in their final comment.

## 3. Conclusion

We appreciate the reviewers’ thoughtful input, which has strengthened our manuscript substantially. We hope that the revisions and detailed clarifications fully address the remaining concerns. We believe PepTri offers a unified framework that integrates physics, evolutionary signals, and mutual information directly into the denoising process of a latent diffusion model.

**Sincerely,**

*The Authors*

---

### Meta-Review · Area_Chair_z4iw · 2026-01-05

**Summary:**

This submission presents an SE(3)-equivariant latent diffusion framework for joint peptide sequence–structure generation, steered during denoising by three complementary signals: physics-based guidance, a learned “evolutionary” prior from receptor–peptide contexts, and mutual-information guidance for sequence–structure coherence. Across PepBench/LNR/PepBDB, it delivers consistent improvements over strong baselines on binding- and interface-relevant metrics while maintaining diversity, and the rebuttal strengthens the empirical case with added structural validity metrics and clearer ablations.

Advantages
* Integrated co-design (sequence + all-atom structure) with tri-guidance applied during denoising; conceptually clean and empirically strong.
* Thorough benchmark coverage (in-domain and cross-domain) with generally consistent gains; module-level ablations support additive benefits of guidance components.
* Rebuttal substantially improves presentation quality and completeness: clarified backbone details/notation, moved key ablations to main text, and added missing local-geometry metrics (Clash_in/Clash_out/Bond outliers) that better validate the physics-guidance claims.

Disadvantages
* Presentation/replicability issues in the initial version (missing architectural and energy-guidance details; notation inconsistencies); largely addressed in the rebuttal, but the method remains somewhat complex.
* Some evaluation choices/metrics were initially missing or questioned (e.g., external structure validation beyond Rosetta proxies; local validity checks; OOD stress tests); rebuttal adds geometry metrics and OOD length extrapolation.
* Novelty is more in system-level integration + guidance-in-denoising than a fundamentally new algorithmic primitive; still a solid contribution for peptide design.

**Reviewer Concerns:**

The rebuttal addresses most major concerns around missing metrics (added clash/bond-outlier metrics pre/post relaxation), ablation visibility (moved key ablations into main text), method clarity (expanded backbone description, standardized time-dependent notation, clarified roles of composite energy vs. OpenMM and inference-time guidance choices), and generalization (added OOD long-peptide stress test).

**Reviewer Scores:**

* wyh8: likely increases (explicitly stated prepared to raise to 6 after added metrics/clarifications and stronger empirical evidence).
* 55wn: likely increases modestly (main concerns on vagueness/notation/ablations were addressed; still some residual ambiguity around energy guidance, but substantially clarified).
* eX3Z: may stay similar or slightly increase (metric criticism partially addressed via added geometry validity metrics and practical constraints on AF/OF-style evaluation; novelty concern remains, but overall evidence strengthened).
* 4LzT: likely unchanged (already strong accept; rebuttal addressed OOD and receptor-awareness questions, reinforcing their position).

---

### Decision · Program_Chairs · 2026-01-26

Accept (Poster)